# Genome-Wide Identification of *Arachis hypogaea LEC1s*, *FUS3s*, and *WRIs* and Co-Overexpression of *AhLEC1b*, *AhFUS3b*, *AhWRI1a* and *AhWRI1d* Increased Oil Content in *Arabidopsis* Seeds

**DOI:** 10.3390/plants14182910

**Published:** 2025-09-19

**Authors:** Xiangzhen Yin, Jianxin Zhao, Lijuan Pan, Enqi Wang, Na Chen, Jing Xu, Xiao Jiang, Xuhong Zhao, Junqing Ma, Shouhui Li, Hongfeng Xie, Zhen Yang, Shanlin Yu, Xiaoyuan Chi

**Affiliations:** 1Peanut Quality Breeding Laboratory, Shandong Peanut Research Institute, Qingdao 266100, China; yinxiangzhen1985@163.com (X.Y.); panlijuan_2008@aliyun.com (L.P.); chenna7948@163.com (N.C.); xu_jing_cool@yeah.net (J.X.); 15066223517@163.com (X.J.); zhaoxh24@163.com (X.Z.); mjq2024@yeah.net (J.M.); sige6353@163.com (H.X.); woozhyang@hotmail.com (Z.Y.); shanlinyu2012@163.com (S.Y.); 2College of Biological Engineering, Qingdao University of Science and Technology, Qingdao 266042, China; zhaojianxin0315@163.com (J.Z.); muyang166@163.com (E.W.); 3Faculty of Agriculture, Jilin Agricultural University, Changchun 130118, China; 17860986786@163.com

**Keywords:** peanut (*Arachis hypogaea*), transcription factors, co-overexpression, fatty acid synthesis

## Abstract

Peanut (*Arachis hypogaea*) is an important oil and economic crop widely cultivated worldwide. Increasing the oil yield is a major objective for oilseed crop improvement. Plant LEAFY COTYLEDON1s (LEC1s), FUSCA3s (FUS3s), and WRINKLED1s (WRI1s) are known master regulators of seed development and oil biosynthesis. While previous studies in peanut have primarily focused on two *AhLEC1s* and one *AhWRI1* genes, this study identified a broader set of regulators, including two *AhLEC1s*, two *AhFUS3s*, nine *AhWRI1s*, two *AhWRI2s,* and four *AhWRI3s* from the variety HY917. The analyses of phylogenetic trees, gene structures, conserved domains, sequence alignment and identity, and collinearity revealed that they were highly similar to their homologs in other plants. Expression profiling demonstrated that two *AhLEC1s*, two *AhFUS3s*, and three *AhWRI1s* (*AhWRI1a/b/c*) were specifically expressed in developing seeds, suggesting critical roles in seed development, whereas *AhWRI1d*, *AhWRI1f*, and *AhWRI1g* showed high expression in root nodules, pointing to potential functions in symbiosis and nodulation. Furthermore, co-overexpression of *AhLEC1b*, *AhFUS3b*, *AhWRI1a*, and *AhWRI1d* in *Arabidopsis* significantly enhanced seed oil content and thousand-seed weight, but also led to reduced germination rate, plant height, and silique length. The findings allow for the extensive evaluation of *AhLEC1s*, *AhFUS3s,* and *AhWRIs* gene families, establishing a useful foundation for future research into their multiple roles in peanut development.

## 1. Introduction

Seed storage reserves in many plants primarily constitute starch, vegetable oil, and storage proteins. Plants synthesize and accumulate vegetable oils in seeds not only as major carbon and energy reservoirs to support germination and seedling development, but also as important resources for the human diet and renewable industrial feedstock [1,2,3]. The continuous growth of the global population has led to a rapid increase in food demand, posing a serious challenge to global food security and thus incentivizing tremendous research efforts focusing on enhancing plant oil biosynthesis [1,4].

In plants, seed oil accumulation is precisely regulated by intricate multilevel regulatory networks, with transcriptional regulation playing a central role in controlling oil biosynthesis [5]. Several key transcription factors, such as LEAFY COTYLEDON1 (LEC1), LEAFY COTYLEDON1-LIKE (L1L), FUSCA3 (FUS3), and WRINKLED1 (WRI1), have been identified as critical regulators of seed oil accumulation [6,7,8,9].

AtLEC1 and AtL1L are two HAP3 subunits of CCAAT-binding transcription factor [10,11,12]. Induced expression of *AtLEC1* leads to a broad upregulation of genes involved in fatty acid (FA) metabolism, significantly promoting the accumulation of oil and major FA species [7]. Overexpression of *ZmLEC1* enhances the oil content in transgenic maize, *Arabidopsis thaliana*, and *Camelina sativa* [13,14]. Similarly, overexpression of oilseed crop *LEC1* homologs, including *BnLEC1* and *BnL1L* in *Brassica napus*, *GmLEC1* (*Glyma.07G268100*) in *Glycine max*, and *AhNF-YB1* and *AhNF-YB10* in *Arachis hypogaea*, also significantly increases total seed oil content in transgenic seeds [15,16,17]. In addition, *LEC1* activation induces ectopic embryogenesis in *Arabidopsis* vegetative tissues [7,10], and OsNF-YB7, a rice homolog LEC1, functions as a key regulator of seed maturation but also inhibits embryo greening during seed development [18,19,20].

FUS3, a member of the B3-domain transcription factor family [21], serves as a master regulator of seed development, establishing and maintaining embryonic identity. It also modulates hormonal responses during late embryogenesis and germination [22,23,24]. FUS3 promotes seed oil accumulation by upregulating genes involved in photosynthesis, FA biosynthesis, and triacylglycerol assembly [8,25,26,27], and boosts the accumulation of seed storage proteins, such as 2S and 12S [24,28,29,30].

The WRI1/2/3/4 proteins belong to the APETALA2-ethylene-responsive element-binding protein (AP2/EREBP) family [31,32,33]. WRI1 can directly target many genes of glycolysis and fatty acid biosynthesis [34,35]. Mutations in *AtWRI1* result in a wrinkled seed phenotype and significantly reduced FA levels in *Arabidopsis* seeds [9,31,36], whereas overexpression of *AtWRI1* or its homologs markedly increases oil content in both seeds and vegetative tissues [37,38,39,40,41]. Overexpression of *AtWRI3* or *AtWRI4*, but not *AtWRI2*, can rescue the low oil accumulation and wrinkled phenotype of the *wri1-4* mutant [33]. Deficiencies in *AtWRI2*, *AtWRI3,* or *AtWRI4* do not affect seed oil accumulation [33].

Peanut (*A*. *hypogaea*), one of the major oil crops in the world [42], is an allotetraploid (AABB) crop that possibly originated from the hybridization of two diploid progenitors, *Arachis duranensis* (AA) and *Arachis ipaensis* (BB) [43,44]. Functional analyses in polyploid oilseeds like peanut are complicated by gene redundancy from genome duplication. However, comprehensive analyses of the copy number, evolutionary relationships, gene structures, conserved domains, physicochemical properties, chromosome distributions, gene synteny, subcellular localization, *cis*-element distributions in promoters, and spatiotemporal expression patterns will allow for an extensive evaluation of redundant genes and build a useful foundation with which to dissect their functions in biochemical and physiological processes. Now these comprehensive analyses of *AhLEC1s*, *AhFUS3s*, and *AhWRIs* in *A. hypogaea* have not yet been performed. Although the roles of individual transcription factors LEC1, FUS3, and WRI1 have been explored in plants, there are currently no academic reports of these genes stacking to enhance seed oil content. Therefore, we try simultaneously to alter the expression of these genes and test empirically whether combinations provide a better effect.

In this study, we cloned and identified two *AhLEC1s*, two *AhFUS3s*, nine *AhWRI1s*, two *AhWRI2s,* and four *AhWRI3s* from the cultivated peanut variety HY917. We analyzed their evolutionary relationships, exon/intron gene structures, conserved domains, physicochemical properties, chromosome distributions, gene synteny, subcellular localization, and *cis*-element distributions in their promoters. Based on published transcriptome data, *AhLEC1a/b*, *AhFUS3a/b*, and *AhWRI1a/b/c* were specifically expressed in developing seeds. In contrast, *AhWRI1d*, *AhWRI1f*, and *AhWRI1g* were highly expressed in roots and nodules. *AhWRI1d* and *AhWRI1e* were upregulated under water-deficit conditions both with and without ABA, whereas *AhWRI3c* and *AhWRI3d* were repressed under drought or low-temperature treatments. Co-overexpression of *AhLEC1b*, *AhFUS3b*, *AhWRI1a*, and *AhWRI1d* in *Arabidopsis* increased the mature seed oil content and thousand-seed weight, but decreased the germination rate, plant height, and silique length. These findings provide valuable information for the peanut master regulators AhLEC1s, AhFUS3s, and AhWRIs, suggesting that they may play multiple important roles during peanut development.

## 2. Results

### 2.1. Cloning and Identification of AhLEC1s, AhFUS3s, and AhWRIs in A. hypogaea

To identify candidate members of LEC1s, FUS3s, and WRIs in *A. hypogaea*, protein sequences of *A. thaliana* homologs, including AtLEC1, AtL1L, AtFUS3, AtWRI1, AtWRI2, AtWRI3, and AtWRI4, were retrieved from The Arabidopsis Information Resource (TAIR, http://www.arabidopsis.org/; accessed on 10 May 2023) [45] and used as queries for BLAST searches against the following peanut genomic resources: Peanut Genome Resource (PGR, http://peanutgr.fafu.edu.cn/; accessed on 15 May 2023) [46], PeanutBase (https://www.peanutbase.org/; Tifrunner.gnm2.ann2.PVFB (https://data.legumeinfo.org/Arachis/hypogaea/annotations/Tifrunner.gnm2.ann2.PVFB/) and BaileyII.gnm1.ann1.PQM7 (https://data.legumeinfo.org/Arachis/hypogaea/annotations/BaileyII.gnm1.ann1.PQM7/); accessed on 15 May 2023) [47,48], *Arachis hypogaeaLine8* v1.3 (https://phytozome-next.jgi.doe.gov/info/AhypogaeaLine8_v1_3; accessed on 10 October 2024) [49,50], and the allotetraploid wild peanut *Arachis monticola* genome (http://gigadb.org/dataset/100453; accessed on 18 May 2023) [51] (Table 1 and Appendix A). Based on the retrieved nucleotide sequences, primers were designed (Appendix A) to clone and identify the coding sequences (CDSs) of putative *AhLEC1s*, *AhFUS3s,* and *AhWRIs* from the cultivated peanut variety HY917 (Appendix A). In addition, homologs of LEC1s, FUS3s, and WRIs were also identified in nine additional plant species (Appendix A). All putative CDS, genomic DNA, and protein sequences used in this study were obtained from their respective genome databases and are presented in Appendix A.

Two putative LEC1s were detected in each of the four *A. hypogaea* genome annotations and in *A. monticola*, whereas no LEC1 homologs were found in *A. duranensis* and *A. ipaensis* (Table 1). From HY917, two *AhLEC1* genes were successfully cloned and annotated as *AhLEC1a* and *AhLEC1b* (Table 1 and Appendix A). Their CDSs contained two exons, similarly to most *LEC1s* in other plants (Figure 1A and Appendix A). Both AhLEC1a and AhLEC1b, along with their homologs, contained a conserved CBFD_NFYB_HMF domain (pfam00808; Figure 1A). Phylogenetic analysis showed that both AhLEC1s were more closely related to AtL1L than to AtLEC1 (Figure 1A). Physicochemical property analysis demonstrated the molecular weight (MW) and theoretical isoelectric point (pI) of AhLEC1a and AhLEC1b are 25.29–25.44 kDa and 6.11–6.17, respectively (Appendix A). Their aliphatic index and the grand average of hydropathicity (GRAVY) were predicted to be 60.88–61.16 and −0.735–−0.728 (Appendix A), respectively, indicating high hydrophilicity, while their instability index was shown to be 36.10–37.40 (Appendix A), suggesting high stability.

Two putative *FUS3s* were identified in each of the four *A. hypogaea* genomes annotations and in the *A. monticola* genome. These were cloned from HY917 and named *AhFUS3a* and *AhFUS3b* (Table 1 and Appendix A). In contrast, only one putative *FUS3* was detected in *A. duranensis* and *A. ipaensis,* respectively (Table 1). All of these genes have six exons and five introns, and their encoded proteins contain a B3 DNA-binding domain (cd10017), similar to those in other plant species (Figure 1B and Appendix A). Phylogenetic analysis showed that both AhFUS3s are more closely related to their homologs from leguminous species (*G. max*, *Glycine soja*, and *Medicago truncatula*), than to those of other plants (Figure 1B).

Physicochemical property analysis indicated the molecular weight and theoretical pI of AhFUS3a and AhFUS3b are 42.24–42.31 kDa and 5.58, respectively (Appendix A). Their aliphatic index and GRAVY were predicted to be 69.63–69.66 and −0.500–−0.491 (Appendix A), respectively, indicating high hydrophilicity. Their instability index was shown to be 54.17–55.65 (Appendix A), suggesting high instability.

Compared with the four *A. hypogaea* genomes, nine putative *AhWRI1s* and two putative *AhWRI2s* in *A. hypogaea* were successfully cloned from HY917 and designated as *AhWRI1a* to *AhWRI1i* and *AhWRI2a* to *AhWRI2b*, respectively (Table 1 and Appendix A). Phylogenetic analysis indicated that AtWRI3 and AtWRI4 are more closely related to each other than to AtWRI1 or AtWRI2 (Figure 1C and Appendix A). Accordingly, four genes in *A. hypogaea* that showed close homology to both *AtWRI3* and *AtWRI4* were annotated as *AhWRI3a*, *AhWRI3b*, *AhWRI3c*, and *AhWRI3d* (Table 1). In addition, genome surveys identified six *WRI1s*, two *WRI2s*, and two *WRI3s* in *A. monticola*, four *WRI1s*, one *WRI2*, and two *WRI3s* in *A. duranensis*, and four *WRI1s*, one *WRI2*, and one *WRI3* in *A. ipaensis* (Table 1). Gene structure analysis revealed that all nine *AhWRI1s* and four *AhWRI3s* contained eight exons, whereas their homologs in other plant species exhibited variable exon numbers, ranging from six to eight exons (Figure 1C and Appendix A). Both *AhWRI2s* consistently contained nine exons, in contrast to the putative *WRI2s* in other plants, which range from seven to nine exons (Figure 1C and Appendix A). Notably, all *AhWRI1s*, *AhWRI2s*, and *AhWRI3s* were found to carry a 9 bp micro-exon (Figure 1C) consistent with previous reports for *AtWRI1* [32]. Domain analysis showed that All AhWRI1s and AhWRI3s proteins contain two AP2 domains (smart00380 and cl00033 for AP2 and AP2 superfamily, respectively; Figure 1C and Appendix A), similarly to their homologs in other plants [31,32,33]. In contrast, both AhWRI2s and other plant WRI2s possess only one AP2 domain (smart00380; Figure 1C and Appendix A). It is worth noting that three putative WRI1s (EVM0034741.1, Araip.N6C0B.1, and Aradu.EN58B.1) and four WRI3s (Araip.S3MIZ.1, Aradu.P6UBG.1, EVM0007256.1, and Aradu.A4KG4.1) from the previously reported genomes of *A. monticola*, *A. duranensis*, and *A. ipaensis* were found to lack one or both AP2 domains (Figure 1C). Similarly, one putative WRI2 in *A. monticola* (EVM0007664.1) had also lacked the AP2 domain (Figure 1C). The genes in these reported genomes may be partial and thus there is need for additional cloning to identify the full length. Physicochemical property analysis indicated the molecular weight and theoretical pI of AhWRIs were 40.71–47.04 kDa and 6.37–9.45, respectively (Appendix A). Their aliphatic index and GRAVY were predicted to be 55.24–64.45 and −1.019–−0.735 (Appendix A), respectively, indicating high hydrophilicity. Their instability index was shown to be 41.41–65.85 (Appendix A), suggesting high instability.

### 2.2. Protein Alignment and Identity Analysis

Protein alignment showed that the central B (CBFD_NFYB_HMF) domains of AhLEC1a and AhLEC1b demonstrated strong similarity to those of other plant LEC1s, as seen in the conserved central domain for both proteins (Figure 2A) as well as the full-length proteins (Appendix A). In contrast, the amino-terminal A or carboxyl-terminal C domains exhibited low sequence similarity among AhLEC1s and other LEC1 homologs in (Appendix A). Analyzing the identities of conserved domains (Appendix A), 83.3–100.0% similarity was demonstrated among AhLEC1a, AhLEC1b, AtLEC1, and AtL1L; however, AhLEC1a and AhLEC1b showed 51.3–51.6% similarity with AtLEC1 and 59.6–60.6% similarity with AtL1L based on full-length proteins, while that between AhLEC1a and AhLEC1b was 95.6% for full-length proteins. The putative DNA-binding region, putative subunit association regions, and secondary structures (α-helices and coils) were observed in the highly conserved B domains as other NF-YBs or HAP3s (Figure 2A) [10,12,56,57]. Furthermore, AhLEC1a and AhLEC1b contain amino acid residues conserved in AtLEC1 and AtL1L B domains but divergent in other AtNF-YBs (Figure 2A) [11,12]. Notably, both proteins contained a conserved residue Asp (D) within the central B domain, which is essential for AtLEC1 function [12] (Figure 2A).

Protein alignment of FUS3 homologs indicated that two AhFUS3s, like other FUS3s, contain a conserved B3 domain, flanked by non-conserved N- and C-terminal region (Figure 2B and Appendix A). Full-length FUS3 proteins from *A. hypogaea*, *A. monticola*, *A. duranensis*, and *A. ipaensis* share 60.1–60.5% identity with AtFUS3, while their B3 domains share 75.9–76.7% identity. Among FUS3s of *A. hypogaea*, *A. monticola*, *A. duranensis*, and *A. ipaensis* sequence identity reached 97.1–100% for full-length proteins and 97.4–100% for conserved domains (Appendix A). An aspartic acid residue (D) in the B3 domain, equivalent to D75 in HvFUS3 [58], was observed across all species (Figure 2B). The N-terminal regions of two AhFUS3s contain a putative B2 domain (Appendix A) with a potential nuclear localization signal (NLS; Appendix A). The three serine sites (55-SSS-57) in AtFUS3, phosphorylated by AKIN10 to promote degradation [59,60], were not conserved among other FUS3 homologs (Appendix A). Two putative PEST sequences, associated with protein turnover in response to ABA and GA signaling [61,62], were identified in the C-terminal domain of two AhFUS3s, while one or two putative PEST sequences were also detected in other plant FUS3s (Appendix A).

As in other WRIs in previous studies [33,40,63,64,65,66,67], all AhWRI1s and AhWRI3s contained two typical AP2/ERF DNA-binding domains, whereas two AhWRI2s possessed only one AP2 domain. All isoforms exhibited considerable sequence variation in their N- and C-terminal regions (Figure 2C,D and Appendix A). AhWRI1a, AhWRI1b, and AhWRI1c showed high mutual similarity (87.4–97.5% and 93.9–100.0% for the full-length proteins and conserved regions, respectively), lower identity with other AhWRI1s (40.5–43.7% and 75.5–81.6%, respectively; Appendix A). The remaining AhWRI1s (AhWRI1d–i) shared 55.5–99.5% (full-length) and 90.2–100.0% (conserved region) identity among themselves (Appendix A). AhWRI3s have higher identity (93.9–100.0%) with each other than with AhWRI1s (77.3–80.4%) in conserved regions (Appendix A). Both AhWRI2s have 69.6% and 100.0% identity with AtWRI2 for the full-length proteins and conserved regions, respectively (Appendix A).

The tripeptide VYL encoded by the conserved micro-exon was identified in all AhWRI1s, AhWRI2s, and AhWRI3s (Figure 2C,D and Appendix A) as previously reported in other WRIs [32,39,63,68,69,70,71]. The 14-3-3 proteins, which are phosphopeptide-binding proteins, were found to potentially bind to the first AP2 domain of AtWRI1 and increase its stability, while the 14-3-3 binding motif (AtWRI1^78-92^) in AtWRI1 was overlapped with that for BPM [72]. A putative motif for binding both 14-3-3 and BPM was also identified within the first AP2 domain of peanut and other plant WRI1 and WRI3 proteins (Figure 2D).

### 2.3. Chromosomal Distribution and Collinearity Analysis

To uncover the chromosomal location of *LEC1s*, *FUS3s*, *WRI1s*, *WRI2s*, and *WRI3s*, we visualized their physical positions on the corresponding chromosomes of *A. hypogaea*, *A. duranensis*, *A. ipaensis*, and *A. monticola* (Figure 3). In *A. hypogaea*, with 40 total chromosomes, two *AhLEC1s* were identified on chr01 and chr11; two *AhFUS3s* on chr06 and chr16; nine *AhWRI1s* on chr03, chr04, chr08, chr10, chr13, chr14, chr15, chr18, and chr20; two *AhWRI2s* on chr06 and chr16; and four *AhWRI3s* on chr01, chr11, chr09, and chr19 (Figure 3A). Clearly, for *A. duranensis* and *A. ipaensis*, it would instead be 20 total chromosomes. In *A. duranensis*, one putative *FUS3* was identified on Aradu.A06; four putative *WRI1s* on Aradu.A03, Aradu.A04, Aradu.A08, and Aradu.A10; one putative *AhWRI2* on Aradu.A06; and two putative *WRI3s* on Aradu.A01 and Aradu.A09 (Figure 3B). In *A. ipaensis*, one putative *FUS3* was identified on Araip.B06; four putative *WRI1s* on Araip.B03, Araip.B04, Araip.B05, and Araip.B10; one putative *AhWRI2* on Araip.B06; and one putative *WRI3* on Araip.B09 (Figure 3C). In *A. monticola*, two putative *LEC1s* were identified on A.mon-A01 and A.mon-B01; two putative *FUS3s* on A.mon-A06 and A.mon-B06; six putative *WRI1s* on A.mon-A04, A.mon-A08, A.mon-A10, A.mon-B03, A.mon-B03, and A.mon-B10; two putative *AhWRI2s*, in close proximity to each other, on A.mon-B06; and two putative *AhWRI3s* on A.mon-A01 and A.mon-B09 (Figure 3D).

We analyzed the collinear relationship of *AhLEC1s*, *AhFUS3s*, *AhWRI1s*, *AhWRI2s*, and *AhWRI3s* (Figure 4A). The results showed that *AhFUS3a*, *AhWRI1a*, *AhWRI2a*, *AhWRI3a*, and *AhWRI3c* had collinear relationships with *AhFUS3b*, *AhWRI1b*, *AhWRI2b*, *AhWRI3b* and *AhWRI3d*, respectively, and that *AhWRI1d*, *AhWRI1e*, *AhWRI1f*, *AhWRI1g*, *AhWRI1h*, and *AhWRI1i* shared the same collinear relationship. Conversely, *AhLEC1a* had no collinear relationship with *AhLEC1b*; *AhWRI1a* and *AhWRI1b* had no collinear relationships with the other seven *AhWRI1s*; *AhWRI1c* had no collinear relationships with the other eight *AhWRI1s*; and *AhWRI3a* and *AhWRI3b* had no collinear relationships with *AhWRI3c* and *AhWRI3d*, respectively.

To investigate the collinearity of *LEC1*, *FUS3*, *WRI1*, *WRI2*, and *WRI3* genes across species, we performed analysis among *A. thaliana*, *A. hypogaea*, *G. max*, *M. truncatula*, *Theobroma cacao*, and *Helianthus annuus* (Figure 4B). For *LEC1*-type genes, *AtL1L*, *AhLEC1a*/*AhLEC1b*, *Glyma.07G268100*/*Glyma.17G005600*, *Medtr4g133952*, and *Thecc.06G241100*/*Thecc.07G013400* shared conserved synteny. *Glyma.20G000600* was collinear with *Medtr1g039040*, and *Medtr5g095860* showed collinearity with *Thecc.06G241100* and *Thecc.07G013400*. *Thecc.06G241100* was also collinear with *HanXRQChr16g0515761*. Additionally, *Glyma.07G268100* and *Glyma.17G005600* were collinear with *Medtr2g026710*, which is phylogenetically closer to *AtNF-YB3* (*AT4G14540*) than to *AtLEC1* or *AtL1L*. All examined *FUS3* orthologs across the six species exhibited conserved collinearity. In contrast, *AtWRI1* did not show collinearity with any *WRI1* homologs from the other species. Multiple *AhWRI1* genes (*AhWRI1d–i*) shared synteny with *Glyma.09G240400*, *Glyma.18G256000*, *Glyma.18G125200*, and *Glyma.08G297000*. Other collinear relationships included *Glyma.15G221600* and *Glyma.08G227700* with *Medtr8g044040* and *Thecc.10G130200*; *Glyma.09G240400* and *Glyma.18G256000* with *Medtr7g009410* and *Medtr6g011490*; *Glyma.18G125200* with *Medtr8g468920*, *Medtr7g009410*, and *Medtr6g011490*; *Glyma.08G297000* with *Medtr8g468920* and *Medtr6g011490*; *Thecc.02G354400* and *Thecc.04G075600* with *Medtr8g468920*, *Medtr7g009410*, *Medtr6g011490*, and *HanXRQChr02g0056411*; *Thecc.04G075600* with *HanXRQChr09g0265501*. For *WRI2s*, *AtWRI2* exhibited conserved synteny with *AhWRI2a*/*AhWRI2b*, *Glyma.02G185200*/*Glyma.03G136100*/*Glyma.19G138000*, *Medtr7g091390*, and *Thecc.05G148300*. For *WRI3s*, *AtWRI3* was collinear with *AhWRI3c*/*AhWRI3d*, *Glyma.07G021000*/*Glyma.08G220800*, *Medtr4g007770*, *Thecc.06G047900*, and *HanXRQChr09g0274201*/*HanXRQChr16g0499481*. Additionally, *AhWRI3a*/*AhWRI3b* shared synteny with *Glyma.02G207100*/*Glyma.17G070800*, *Medtr4g130270*, and *Thecc.09G154700*. *AtWRI4* showed no collinearity with any *WRI3* homologs from the examined species.

### 2.4. Transcriptional Profiles of AhLEC1s, AhFUS3s, and AhWRIs

To investigate the expression patterns of *AhLEC1s*, *AhFUS3s*, *AhWRI1s*, *AhWRI2s*, and *AhWRI3s* across various tissues and under diverse abiotic stresses and phytohormone treatments, their transcriptional profiles were extracted from PGR (accessed on 11 June 2024; Figure 5A,B) [46], PeanutBase (accessed on 11 June 2024; Appendix A) [73,74], and our RNA-seq data from six seed developmental stages of four sister lines (P19-19, P19-61, P19-57, and HY917; Figure 5C and Appendix A).

According to PGR (Figure 5A), *AhLEC1a* and *AhLEC1b* were specifically expressed in embryos (Embryo-I/II/III/IV) and seed coats (Testa-I/II). *AhFUS3a* and *AhFUS3b* showed high specific expression in embryos (Embryo-I/III/IV), with lower expression in seed coat. *AhWRI1a*, *AhWRI1b*, and *AhWRI1c* were specifically expressed in embryos (Embryo-I/III/IV), while *AhWRI1f* and *AhWRI1g* were specifically highly expressed in root nodules. In contrast, *AhWRI1e*, *AhWRI1h*, and *AhWRI1i* were barely detectable across tissues. *AhWRI2a* and *AhWRI2b* were widely expressed in multiple tissues. *AhWRI3a* and *AhWRI3b* were barely detectable across tissues, whereas *AhWRI3c* and *AhWRI3d* were highly expressed in florescences. In addition, the expression of *AhWRI2a*, *AhWRI2b*, *AhWRI3c,* and *AhWRI3d* gradually increased during the development of pericarp.

According to PeanutBase (Figure 5B), *AhLEC1a*, *AhLEC1b*, *AhFUS3a*, *AhFUS3b*, *AhWRI1a*, *AhWRI1b*, and *AhWRI1c* were highly expressed in developing seeds (Pattee seed 5–8). While *AhWRI1d*, *AhWRI1f*, and *AhWRI1g* were specifically highly expressed in roots and nodules. *AhWRI2a* and *AhWRI2b* were broadly expressed across tissues, while *AhWRI3c* and *AhWRI3d* were barely detected in seeds. Conversely, *AhWRI1e*, *AhWRI1h*, *AhWRI1i*, *AhWRI3a*, and *AhWRI3b* were expressed at very low or nearly undetectable levels in all examined tissues.

Based on our RNA-seq data across the six seed development stages of four sister lines (Figure 5C and Appendix A), *AhLEC1a*, *AhLEC1b*, and *AhWRI1c* were highly expressed in early seed development stages; *AhFUS3a*, *AhFUS3b*, and *AhWRI1a* were most lowly expressed in the final stage; *AhWRI1h*, *AhWRI1i*, *AhWRI2a*, and *AhWRI2b* were lowly expressed across all stages; and *AhWRI1d*, *AhWRI1e*, *AhWRI1f*, *AhWRI1g*, *AhWRI3a*, *AhWRI3b*, *AhWRI3c*, and *AhWRI3d* were weakly expressed or undetectable throughout seed development.

Under abiotic stresses and phytohormone treatments, *AhWRI1d* and *AhWRI1e* were highly induced under water-deficit conditions, both with and without ABA. In contrast, *AhWRI1g* expression was repressed under the same conditions, and *AhWRI1i* was down-regulated under water-deficit without ABA (Appendix A). *AhWRI2a* was induced under water-deficit with ABA, while that of *AhWRI2b* was up-regulated under water-deficit both with and without ABA (Appendix A). Moreover, their expressions were induced under drought but repressed under abscisic acid, brassinolide, ethephon, or paclobutrazol treatments (Appendix A). Expressions of *AhWRI3c* and *AhWRI3d* were repressed under water-deficit conditions (with or without ABA; Appendix A) as well as under drought or low-temperature stress (Appendix A).

### 2.5. Putative Cis-Elements in the Promoters of AhLEC1s, AhFUS3s, and AhWRIs and miRNA Regulating AhLEC1s, AhFUS3s, and AhWRIs

To investigate the potential function and regulatory mechanisms of *AhLEC1s*, *AhFUS3s*, and *AhWRIs*, we analyzed putative *cis*-elements in their putative promoter regions. These regions, defined as the 2000 bp upstream of the start codon, were extracted from the *Arachis hypogaeaLine8* v1.3 genome (accessed on 10 October 2024; Appendix A) and analyzed using PlantCARE [76]. For comparison, putative promoter sequences of *AtLEC1*, *AtL1L*, *AtFUS3*, *AtWRI1*, *AtWRI2*, *AtWRI3*, and *AtWRI4*, were derived from TAIR (Figure 6 and Appendix A). The identified putative *cis*-elements were classified into fifteen functional categories based on their annotations, including MeJA responsiveness, meristem expression, abscisic acid responsiveness, light responsiveness, anaerobic induction, circadian control, defense and stress responsiveness, low-temperature responsiveness, gibberellin responsiveness, and drought inducibility (Figure 6 and Appendix A).

*AtLEC1*, *AtL1L*, and two *AhLEC1s* contained *cis*-acting elements associated with light responsiveness, meristem expression, anaerobic induction, and gibberellin responsiveness. Similarity was found in the distribution of *cis*-acting elements between *AtLEC1* and *AtL1L* and between two *AhLEC1s*. Additionally, *AtLEC1* and *AtL1L* contained *cis*-acting elements involved in abscisic acid responsiveness, auxin responsiveness, low-temperature responsiveness, drought inducibility, circadian control, and endosperm expression.

*AtFUS3* and two *AhFUS3s* contained *cis*-acting elements related to light responsiveness, MeJA-responsiveness, anaerobic induction, meristem expression, abscisic acid responsiveness, and defense and stress responsiveness. Two *AhFUS3s* demonstrated more similarity in their *cis*-element distribution than *AtFUS3*. They also contained elements involved in low-temperature responsiveness.

*AtWRI1* and nine *AhWRI1s* all contained light responsive *cis*-acting elements. *AhWRI1a* and *AhWRI1b*, *AhWRI1f* and *AhWRI1g*, and *AhWRI1h* and *AhWRI1i* demonstrated similarity in their *cis*-element distribution. *AtWRI2* and two *AhWRI2s* contained elements related to light responsiveness, MeJA responsiveness, low-temperature responsiveness, and anaerobic induction. Two *AhWRI2s* showed greater similarity to each other in their *cis*-element distribution than to *AtWRI2* and also contained elements involved in gibberellin responsiveness, defense and stress responsiveness, drought inducibility, and zein metabolism regulation. *AtWRI3*, *AtWRI4*, and four *AhWRI3s* contained *cis*-acting elements associated with light responsiveness, abscisic acid responsiveness, MeJA responsiveness and defense and stress responsiveness. Similar distribution patterns were observed between *AhWRI3a* and *AhWRI3b* and between *AhWRI3c* and *AhWRI3d*. All predicted *cis*-acting elements require functional validation to confirm their roles. We also predicted ahy-miRNAs targeting *AhLEC1s*, *AhFUS3s*, and *AhWRIs* using the psRNATarget website (www.zhaolab.org/psRNATarget/, accessed on 22 March 2025) [77]. The results showed that only two miRNAs (ahy-miR3509-5p and ahy-miR3520-5p) putatively target *AhWRI1s*, while no miRNAs were predicted to target *AhLEC1s*, *AhFUS3s*, *AhWRI2s*, and *AhWRI3s* (Appendix A). In contrast, eleven, three, three, twelve, five, one, and two miRNAs were predicted to target *AtLEC1*, *AtL1L*, *AtFUS3*, *AtWRI1*, *AtWRI2*, *AtWRI3*, and *AtWRI4*, respectively, in *A. thaliana* (Appendix A).

### 2.6. Subcellular Localization of AhLEC1s, AhFUS3s, and AhWRIs

All AhLEC1, AhFUS3, and AhWRI proteins were predicted to localize in the nucleus using ProtComp 9.0 in softberry (Appendix A), consistent with their homologs previously reported in other plants [20,39,59,78,79,80,81,82]. Analysis with cNLS Mapper [83] indicated that AhLEC1s lack putative NLSs, whereas all AhFUS3s shared one same putative NLS. Most AhWRIs were predicted to contain one putative NLS, except for AhWRI1a/b/c and AhWRI3c/d (Appendix A). It has been previously demonstrated that the NLS of AtWRI1 (AtWRI1^33–41^) is necessary and sufficient for its nuclear localization [84]. To experimentally validate their subcellular localization, we selected four highly expressed and/or stress-responsive isoforms: *AhLEC1b*, *AhFUS3b*, and *AhWRI1a* (highly expressed in seeds; Figure 5A,B) and *AhWRI1d* (upregulated under drought; Appendix A). Although all were predicted to be nuclear localized, we aimed to experimentally validate whether variations in their putative NLSs influence actual protein localization. Each was fused to GFP and expressed under the 35S promoter in *Arabidopsis* protoplasts, along with a nuclear mCherry marker. Confocal microscopy revealed that all four fusion proteins co-localized exclusively with the nuclear marker, confirming their nuclear localization (Figure 7), consistent with bioinformatic predictions.

### 2.7. Co-Overexpression of AhLEC1b, AhFUS3b, AhWRI1a, and AhWRI1d Increased Fatty Acids Accumulation and Thousand-Seed Weight, but Decreased Germination Rate, Plant Height, and Silique Length

*AhLEC1b*, *AhFUS3b*, and *AhWRI1a* were selected as representative isoforms due to their highest expression levels during seed development (Figure 5A,B), while *AhWRI1d* was chosen for its high expression in roots and nodules (Figure 5B) and upregulation under drought stress (Appendix A). Using these four selected genes, a tandem gene cassette was constructed with the following arrangement: *P_35S_::AhLEC1b::T_NOS_::P_MAS_::AhFUS3b::CaMV poly(A) signal::P_NOS_::AhWRI1a::CaMV poly(A) signal::P_35S_::AhWRI1d::T_PE9_*. This cassette was cloned into the pCAMBIA2300 vector to generate the co-overexpression construct pLFW (Figure 8A). The pLFW vector was then introduced into *A. thaliana* Col-0 via *Agrobacterium tumefaciens*-mediated transformation. Six independent transgenic lines were selected using kanamycin resistance and confirmed by PCR amplification with three pairs of primers specific to *AhLEC1b*, *AhFUS3b*, *AhWRI1a*, and *AhWRI1d* (Appendix A and Appendix A). qRT-PCR was also performed to detect their relative expression in transgenic lines. And the results showed that these four genes were highly expressed in transgenic lines, compared to in wild type (WT) plants with lowly expression or not detected (Appendix A).

To determine the effect of co-overexpression of these four genes on storage oil accumulation, four transgenic lines (LFW-OX#1, LFW-OX#3, LFW-OX#4, and LFW-OX#5) were used for detecting fatty acid profiling in via gas chromatography. Most measured fatty acid species exhibited increased levels in transgenic seeds (Figure 8B). Specifically, the contents of 16:0, 18:0, 18:1, 18:2, 18:3, 20:0, 20:1, 20:2, and total fatty acids were significantly elevated by 13.48–19.81%, 20.14–28.54%, 11.79–38.10%, 9.18–26.60%, 11.31–18.17%, 20.05–22.57%, 21.28–32.26%, 11.20–18.09%, and 14.22–26.18%, respectively, compared to the wild type. Furthermore, the levels of unsaturated fatty acids, saturated fatty acids, C18, and C20 fatty acids increased significantly by 10.16–27.29%, 12.54–22.93%, 9.75–26.62%, and 12.15–30.13%, respectively (Appendix A). The ratios of unsaturated to saturated fatty acids in LFW-OX#3 and C18 to C20 fatty acids in LFW-OX#1, LFW-OX#3, and LFW-OX#5 also increased significantly (Appendix A).

The thousand-seed weight of mature seeds from the four transgenic lines increased significantly by 8.74–19.20% (Figure 8C). Plant height, silique length, and seed germination were measured only in lines LFW-OX#3 and LFW-OX#5. The co-overexpression of *AhLEC1b*, *AhFUS3b*, *AhWRI1a*, and *AhWRI1d* in both lines resulted in reductions of 16.91–35.48% in plant height, 11.63–19.53% in silique length, and 9.48–11.62% in seed germination, respectively (Figure 8D–F).

## 3. Discussion

*A. hypogaea* is an allotetraploid (AABB) species that possibly originated from the hybridization of two diploid progenitors, namely *A. duranensis* (AA) and *A. ipaensis* (BB) [43,44]. In this study, two *AhLEC1s*, two *AhFUS3s*, nine *AhWRI1s*, two *AhWRI2s*, and four *AhWRI3s* were cloned and identified from *A. hypogaea*. Systematic analyses of phylogenetic trees, gene structures, conserved domains, physicochemical properties, chromosome locations, gene synteny, subcellular localization, and promoter *cis*-element distributions were performed. Expression patterns were further investigated using RNA-seq data. Co-overexpression of *AhLEC1b*, *AhFUS3b*, *AhWRI1a*, and *AhWRI1d* enhanced fatty acids accumulation and thousand-seed weight, but reduced germination rate, plant height, and silique length. These results provide an extensive evaluation of *AhLEC1s*, *AhFUS3s*, and *AhWRIs*, and establish a useful foundation for further function analysis of their roles in biochemical and physiological processes.

### 3.1. Gene Duplication and Functional Diversification of AhLEC1s in A. hypogaea

In *A. thaliana*, AtLEC1 and AtL1L are recognized as LEC1-type NF-YB transcription factors [11,12], with their orthologs subsequently identified in diverse plant species, including *H. annuus* [85], *T. cacao* [86], *Oryza sativa* [20,87], *B. napus* [7,15], and *G. max* [16,88]. In this study, two *AhLEC1* genes were identified in peanut (Table 1). Through comprehensive phylogenetic analysis, sequence alignment, identity matrix assessment, and gene synteny examination, both genes were shown to be more closely related to *AtL1L* than to *AtLEC1* (Figure 1A, Figure 2A and Figure 4, and Appendix A). These findings suggest that *AhLEC1a* and *AhLEC1b* are likely orthologous to *AtL1L*, similar to the previously characterized *PcL1L* [11], rather than to *AtLEC1*. Interestingly, no LEC1-type NF-YBs were detected in the two diploid progenitors, *A. duranensis* and *A. ipaensis* (Table 1 and Figure 3B,C), implying possible evolutionary loss in these lineages.

Structurally, NF-YB subunits consist of three domains: the amino-terminal A domain, central B domain and carboxyl-terminal C domain [10,89]. Sequence analysis revealed that the central B domains of AhLEC1a and AhLEC1b are highly conserved compared to those of LEC1-type NY-YBs from other plants, whereas the A and C domains show considerable sequence divergence (Figure 2A and Appendix A). The conserved B domain and the conserved residue Asp (D) (indicated by a red star in Figure 2A) are required for AtLEC1 function [12,90]. These observations collectively indicate that the central B domains may play key roles in the molecular function of AhLEC1a and AhLEC1b.

LEC1-type *NF-YBs* in plants are typically expressed specifically in seeds during embryogenesis, but are generally absent or present only at very low levels in vegetative tissues [13,16,17,20,85,91]. Functionally, they regulate multiple processes in seed development, including the accumulation of storage compounds [16,17,20,82,92], acquisition of desiccation tolerance [16,19], preparation for dormancy [19], and suppression of premature germination [17]. Consistent with these reports, transcriptome analyses in this study revealed that both *AhLEC1a* and *AhLEC1b* are specifically and highly expressed in peanut seeds (Figure 5A,B), with notably strong expression during early seed development (Figure 5C), corroborating a recent study [17].

*Cis*-acting element analysis revealed two *AhLEC1s* contain the putative motifs associated with light responsiveness, meristem expression, anaerobic induction, and gibberellin responsiveness (Figure 6 and Appendix A). However, their expression remained low or undetectable under abiotic stress or phytohormone treatments, similar to normal conditions (Appendix A). In *Arabidopsis*, repression of *AtLEC1* in vegetative tissues is mediated by distal upstream regulatory sequences, as shown by constitutive expression in the *lec1-d^tnp^* mutant with a 3256 kb deletion in the upstream region [93]. Similarly, deletion of distal promoter regions in *AhLEC1s* led to ectopic expression in vegetative tissues, resembling the *lec1-d^tnp^* phenotype [94,95], indicating that proximal promoter regions are necessary for seed-specific expression. Although AtLEC1 is activated by AtLEC2 via RY motifs in coordination with AtWRI1 [96], no such RY motifs were detected in the promoters of *AtL1L* or *AhLEC1s*. Furthermore, both PICKLE (PKL, a CHD3 chromatin remodeling factor) and VIVIPAROUS ABI3-LIKE (VAL) proteins act epigenetically to repress *AtLEC1* expression during vegetative development [97]. Thus, the mechanism enabling specific high expression of *AhLEC1s* in seeds requires further investigation.

Overall, the above imply strongly both AhLEC1s potentially play key roles in peanut seed development, similar to LEC1-type NF-YBs in other plants [10,12,85,88,98,99,100]. This is supported by functional studies showing that both genes participate in embryogenesis, embryo development, and reserve deposition in cotyledons, albeit with partial redundancy [17]. Specifically, expression of *AhNF-YB10* (*AhLEC1b*) via the *AtLEC1* promoter nearly fully rescued embryonic defects in *lec1-2* mutant, whereas *AhNF-YB1* (*AhLEC1a*) only partially complemented these defects [17]. Furthermore, overexpression of *AhLEC1a* reduced germination rates and seed longevity, a phenotype not observed with *AhLEC1b* [17], indicating functional divergence between the two paralogs.

### 3.2. Gene Duplication and Functional Diversification of AhFUS3s in A. hypogaea

In plants, at least one FUS3 has been identified as members of B3 transcription factor family in seed development and maturation [21,24,27,58,81,101]. In this study, two *AhFUS3s*, designated *AhFUS3a* and *AhFUS3b* (Table 1), were shown to contain one conserved B3 domain, similarly to other plants (Figure 1B, Figure 2B and Appendix A). Genomic analysis further revealed that single *FUS3* orthologs are present in the diploid progenitors *A. duranensis* and *A. ipaensis* (Table 1 and Figure 3B,C), indicating that the two *AhFUS3s* in tetraploid peanut originated from the respective ancestral genomes. Phylogenetically, *AhFUS3a* and *AhFUS3b* cluster closely with FUS3 proteins from other legume species, including *G. max*, *G. soja*, and *M. truncatula* (Figure 1B), suggesting that both AhFUS3s have conserved functions similar to their orthologs in other plants, especially in legume species.

The FUS3 protein contains one conserved B3 domain flanked by divergent N- and C-terminal regions, all of which contribute to its regulatory functions. The B3 domain mediates sequence-specific DNA-binding to RY motifs in target gene promoters [102]. For example, a D75N mutation in the B3 domain of HvFUS3 abolished its binding to RY elements in seed storage protein genes [58]. In this study, the B3 domains of two AhFUS3s showed high sequence identity to those of other FUS3s, and the aspartate residue equivalent to HvFUS3 D75 is strictly conserved (Figure 2 and Appendix A), indicating that both AhFUS3s are likely functional in RY motif recognition and binding.

The N-terminal region of FUS3 is involved in post-translational modifications and protein-protein interactions [58,59,103]. In *Arabidopsis*, AKIN10 interacts with and phosphorylates the N-terminus of AtFUS3 at three serine residues (55-SSS-57), enhancing its stability [59,60]. Although these serine residues are not fully conserved across species (Appendix A), the N-terminal region remains critical for interactions with regulatory partners, such as the E3 ubiquitin ligase AIP2 (which promotes FUS3 degradation) [103] HvBLZ2 in barley [58], and GmNDPK1 in soybean [27].

The C-terminal region of FUS3 plays a key role in hormone sensing and transcriptional activation. In *Arabidopsis*, ABA and GA signaling converge on a PEST sequence in the C-terminus to positively or negatively regulate AtFUS3 protein stability, respectively [61,62]. In this study, two putative PEST sequences were found in the C-terminal domain of both AhFUS3s, a feature shared with other plant FUS3 proteins (Appendix A). This region is also essential for transcriptional activation function [29,58,81], as truncation of the C-terminus in AtFUS3 results in only partial functional complementation [29,62]. These findings suggest that the C-terminal domains of both AhFUS3s are likely important for protein stability, activation capacity, and hormonal regulation.

Expression analyses indicate that *FUS3* genes are primarily associated with seed development. *AtFUS3* is highly expressed during mid-embryogenesis [21,104], and *GmFUS3* in soybean is specifically expressed in pods and seeds [27]. In this study, both *AhFUS3s* genes in peanut showed high specific expression in developing seeds (Figure 5), implying a principal role in seed maturation.

The regulation of *FUS3* is complex and involves multiple transcription factors. In *Arabidopsis*, *AtFUS3* is directly activated by AtLEC2, AtLEC1, and AtWRKY TFs through RY, CCAAT, and W-box elements, respectively [96,98,105,106,107], but repressed by AtTT2 and AtTT8 via MWAMC fragments and E-boxes, respectively, during seed development [108,109]. Promoter analysis of *AhFUS3a* and *AhFUS3b* identified two RY motifs and three CCAAT boxes, respectively (Appendix A), suggesting potential regulation by *LEC1* and *LEC2* homologs for their high seed-specific expression. LTR and ABRE elements were also detected in both promoters (Figure 6 and Appendix A). Although poplar *PeFUS3* is activated by PeABF2 binding to ABRE under osmotic stress [101], neither *AhFUS3* genes showed elevated expression in leaves under abiotic stress or hormone treatments (Appendix A). This indicates that these stress-related cis-elements may not drive significant transcription under the conditions tested or may require specific developmental or environmental contexts.

### 3.3. Gene Duplication and Functional Diversification of AhWRIs in A. hypogaea

In *Arabidopsis*, four WRI paralogs (WRI1-WRI4) belong to the AP2/EREBP family [31,32,33], and homologs have since been identified in numerous species [39,41,66,67,78,79,110,111,112,113,114,115,116,117]. In peanut, we identified fifteen *WRI* genes: nine *AhWRI1s*, two *AhWRI2s*, and four *AhWRI3s* (Table 1). Consistent with WRIs in other species [33,40,63,64,65,66,67,78], all AhWRI1s and AhWRI3s contain two AP2 domains, whereas two AhWRI2s owned only one (Figure 1C, Figure 2C,D and Appendix A). Both AP2 domains are essential for DNA-binding in *R*. *communis* RcWRI1 [41], and in AtWRI1, the first AP2 domain or the segment between the two AP2 domains mediates interaction with LEC2 [96], suggesting they play similar roles in AhWRI1 functions.

Several functional domains were identified in AhWRIs. All contained the conserved “VYL” motif within the first AP2 domain (Figure 2C,D and Appendix A), as with other previously reported *WRIs* [32,39,63,68,69,70,71]. Although alternative splicing can lead to VYL-less isoforms in some species [38,39,63,70] (Figure 2C,D and Appendix A), the VYL-containing variant represents the major and functional transcript in plants [63,70]. While the VYL motif is essential for the function of AtWRI1 and GmWRI1b [39,70,117], its requirement is not universal, as evidenced by functional variants in RcWRI1 [63] and OsWRI1-1 [117]. Therefore, the functional importance of the VYL motif in AhWRIs remains to be determined. The 14-3-3/BPM-binding motif (corresponding to AtWRI1^78–92^) is critical for protein stability, mediating either degradation via BPM-proteasome pathways or stabilization through 14-3-3 interaction [72,118]. This suggests that the putative 14-3-3/BPM-binding motifs in AhWRI1s and AhWRI3s may play similar roles in regulating their protein stability. A putative NLS in AtWRI1 (AtWRI1^33-41^) mediates its nuclear localization [84]. Most AhWRIs were predicted to contain one NLS, except for AhWRI1a/b/c and AhWRI3c/d AhWRI1a/b/c and AhWRI3c/d (Appendix A). Consistent with bioinformatic predictions (ProtComp 9.0, Appendix A), both AhWRI1a and AhWRI1d were experimentally confirmed to be nuclear localized (Figure 7).

The C-terminal acidic regions are required for transactivation in AtWRI1 and GhWRI1 [32,78], whereas in AtWRI4, the N-terminal region containing the first AP2 domain is sufficient for transcriptional activation, and the C-terminal region with the second AP2 domain is dispensable [80]. Although not required for transactivation, the C-terminal IDR3-PEST motif in AtWRI1 modulates its protein stability upon phosphorylation [119]. Additionally, AtMED15 interacts directly with the non-C-terminal acidic region of AtWRI1 to facilitate transcriptional activation [120]. Therefore, the functions of the C-terminal acidic regions in AhWRI1/3s and their contribution to transactivation activity require further investigation.

*WRI1s* are highly expressed in seeds and show co-expression with fatty acid biosynthetic genes across species [31,39,40,41,78,110,113,117]. Loss of WRI1 function reduces seed oil content [9,31,36,66,121], while its overexpression enhances oil accumulation in seeds and vegetative tissues [37,38]. In peanut, *AhWRI1a*, *AhWRI1b*, and *AhWRI1c* exhibit seed-specific expression (Figure 5A,B), consistent with a recent report on *AhWRI1a* (*Ahy_A08g040760*) [122]. Overexpression of the *AhWRI1a* (GG genotype) in *Arabidopsis* increased seed oil content and unsaturated FA levels [122], suggesting that *AhWRI1a*, *AhWRI1b*, and *AhWRI1c* may act redundantly in oil accumulation. Their promoters contain multiple RY and CCAAT elements (Appendix A), implying potential regulation by LEC2 and NF-Y transcription factors, similar to the activation of *AtWRI1* by AtLEC2 binding to RY motif [36,96], and *Elaeis guineensis EgWRI1-1* by EgNF-Y complexes binding specifically to CCAAT-boxes [123].

Beyond their role in oil biosynthesis, plant WRI1 homologs exhibit pleiotropic effects in various processes, including root nodulation in soybean [39], arbuscular mycorrhiza symbiosis in *M. truncatula* [79], auxin homeostasis [124], and cutin and wax biosynthesis [33,125]. For instance, GmWRI1a/b redundantly regulate nodulation and seed filling in soybean through plastidic glycolysis, lipid synthesis, and hormone signaling [39]. In peanut, *AhWRI1d*, *AhWRI1f*, and *AhWRI1g* were highly expressed in nodules (Figure 5A,B), suggesting that they may play key roles in root nodulation, despite showing less phylogenetic proximity to *GmWRI1a/b* [39] than *AhWRI1a/b/c* (Appendix A). In *M. truncatula*, MtWRI5s are involved in arbuscular mycorrhizal symbiosis via the regulation of genes involved in the biosynthesis of fatty acids and phosphate uptake in arbuscule-containing cells [79]. Phylogenetic analysis revealed close relationships between certain *AhWRI1s* and *MtWRI5* genes: *AhWRI1d/e* with *MedtrWRI5c*, *AhWRI1f/g* with *MedtrWRI5b*, and *AhWRI1h/i* with *MedtrWRI5a* (Appendix A). Collectively, these findings suggest that *AhWRI1d*, *AhWRI1f*, and *AhWRI1g* may function in carbon partitioning during both root nodulation and arbuscular mycorrhiza symbiosis in peanut.

AtWRI2 is broadly expressed, with elevated levels in drying siliques and mature seeds [33,80] (Appendix A). It is unable to complement the *wri1* mutant phenotype, and its loss does not affect seed oil content [33]. In contrast, *Persea americana* (avocado) *PaWRI2* is highly expressed during fruit development and enhances triacylglycerol accumulation in transient overexpression assays [67]. In peanut, both *AhWRI2a* and *AhWRI2b* showed broad tissue expression (Figure 5A–C), were induced by drought but repressed by ABA, brassinolide, ethephon, and paclobutrazol (Appendix A). Their promoters contain multiple *cis*-acting elements associated with MeJA, anaerobic induction, gibberellin response, defense and stress signaling, and drought, suggesting functional roles beyond fatty acid synthesis.

*AtWRI3* and *AtWRI4* are ubiquitously expressed, with highest levels in vegetative tissues and flowers [33] (Appendix A). Although capable of partially rescuing the *wri1* mutant, they are not essential for seed oil accumulation [33], but contribute to cutin biosynthesis in floral tissues, preventing organ fusion [33]. Furthermore, *AtWRI4* is salt-induced and regulates cuticular wax biosynthesis [80]. Homologs in other species exhibit diverse expression patterns: *CeWRI4* (*Cyperus esculentus*) is abundant in leaves and roots [65], *RcWRI3* is pollen-specific [41], and *MdWRI4* (*Malus* × *domestica*) is pericarp-specific and stress-responsive [64]. Heterologous expression of *CeWRI4* and *MdWRI4* in *Arabidopsis* improves stress tolerance by enhancing cuticular wax deposition [64,65]. In this study, *AhWRI3a*/*b* showed very low expression across tissues (Figure 5A–C), while *AhWRI3c/d* were more widely expressed (Figure 5A,B), particularly in florescences (Figure 5A), but not in seeds (Figure 5C). In addition, *AhWRI3c/d* were suppressed under drought and low-temperature conditions. This suggests possible roles in cuticular wax formation, florescence development, and stress responses, which merit further investigation.

### 3.4. Co-Overexpression of AhLEC1b, AhFUS3b, AhWRI1a, and AhWRI1d Alter Key Major Agronomic Traits

Previous studies have demonstrated that overexpression of individual regulators such as *LEC1*, *FUS3*, and *WRI1* homologs increases oil/FA content and often modifies FA composition in seeds or seedlings [37,38,97,102,126]. Specifically, transgenic expression of *AhNF-YB1/10* (*AhLEC1a/b*) and *AhWRI1a* (GG genotype) in *Arabidopsis* increased seed oil content by 6.2–29.1% and approximately 4.7%, respectively [17,122]. In our study, co-overexpression of *AhLEC1b*, *AhFUS3b*, *AhWRI1a*, and *AhWRI1d* in *Arabidopsis* significantly increased total FA content by 14.22–26.18% (Figure 8B), an effect comparable to that of *AhNF-YB1/10* expressions and greater than that achieved with *AhWRI1a* (GG).

Transgenic expression of *AhNF-YB1/10* elevated the contents of major FAs, with notable increases in C18:1n9c, C18:2n6c, C18:3n3, C20:1, and C22:1n9 [17]. Similarly, *AhWRI1a* (GG) overexpression raised unsaturated FA levels [122]. In our study, co-overexpression of *AhLEC1b*, *AhFUS3b*, *AhWRI1a*, and *AhWRI1d* increased both saturated and unsaturated FAs (16:0, 18:0, 18:1, 18:2, 18:3, 20:0, 20:1, and 20:2), though only line LFW-OX#3 significantly increased the ratio of unsaturated to saturated FAs (Figure 8B and Appendix A).

Thousand-seed weight in the four transgenic lines increased significantly by 8.74–19.20%, compared to the 3.31–27.81% increase in *AhNF-YB1*/*10*-overexpressing plants [17]. Although thousand-seed weight was not quantitatively reported, *AhWRI1a*-OE produced longer seeds [122]. Consistent with reports that *LEC1* or *WRI1* overexpression enhances seed weight [17,40,71,78,92,111,127,128]. Together, these findings suggest that *AhLEC1b*, *AhWRI1a*, and *AhWRI1d* may cooperatively regulate both FA composition and seed weight.

Overexpression of *LEC1* and *FUS3* homologs often lead to adverse agronomic traits, including decreased plant height, shorter siliques, and lower seed germination rates [13,17,24,129]. For example, overexpression of *ZmLEC1* driven by two embryo-preferred promoters (P*_OLE_* and P*_EAP1_*) decreased plant height and germination [13]. However, these negative effects can be mitigated by using suitable promoters [14,15,92]. Notably, both *AhLEC1s* were expressed under the truncated *NapA* promoter P211 showed no obvious defects throughout the life cycle, whereas *AtLEC1* promoter-driven *AhLEC1a* significantly reduced germination rate and seed longevity [17]. In contrast, *AhWRI1a* (GG) overexpression promoted larger rosette leaves, early flowering, larger pods, and longer seeds [122]. *FUS3* is known to influence plant architecture and germination by modulating ABA and GA levels [27,61,102]. In our study, co-overexpression of *AhLEC1b*, *AhFUS3b*, *AhWRI1a*, and *AhWRI1d* reduced plant height, silique length, and germination rate (Figure 8D–F), suggesting that *AhLEC1b* and *AhFUS3b* may function in a coordinated manner to regulate these traits.

These results demonstrate that co-overexpression of these regulatory genes markedly influences key agricultural traits, yet the molecular mechanisms underlying the trade-offs among FA composition, seed weight, plant architecture, and germination efficiency remain unclear and warrant further study. Previous studies highlight the importance of selecting appropriate promoters to maximize target gene expression while minimizing adverse phenotypic effects [14,15,17,92,127,130]. For instance, seed-specific co-expression of *AtWRI1*, *AtDGAT1*, and RNAi-*AtSDP1* significantly increase seed oil content without impairing seed vigor or plant growth [130]. Therefore, future work will utilize suitable seed-specific promoters to co-express these peanut genes, reducing the risk of undesirable traits. Additionally, the biological functions of *AhLEC1*, *AhFUS3*, and *AhWRI* genes require systematic characterization in peanut and/or *Arabidopsis*.

## 4. Materials and Methods

### 4.1. Plant Materials and Growth Conditions

At the Laixi Experimental Station of Shandong Peanut Research Institute, the roots, stems, leaves, root nodules, and developing pods of peanut cultivar HY917, grown under natural conditions, were collected and pooled for cloning members of *AhLEC1*, *AhFUS3* and *AhWRI* gene families. Additionally, developing seeds of four peanut sister lines (HY917, P19-19, P19-61, and P19-57) were sampled, immediately frozen in liquid nitrogen, and stored at −80 °C for subsequent analysis. These seeds were categorized into six stages based on kernel morphology according to Pattee et al. [75] with three biological replicates each containing more than 10 seeds (Appendix A).

All transgenic and wild-type (Col-0) *Arabidopsis* plants were cultivated concurrently under controlled environmental conditions: 16-h of light/8-h dark photoperiods, 23 °C day/21 °C night temperatures, and 60% relative humidity. The light intensity, measured at the mid-canopy level, was maintained at 160 µmol m^−2^ s^−1^.

### 4.2. BLASTing and Cloning of LEC1, FUS3, and WRI Family Members in A. hypogaea

Protein sequences of AtLEC1 (AT1G21970), AtL1L (AT5G47670), AtFUS3 (AT3G26790), AtWRI1 (AT3G54320), AtWRI2 (AT2G41710), AtWRI3 (AT1G16060), and AtWRI4 (AT1G79700) were retrieved from The Arabidopsis Information Resource (TAIR; accessed on 10 May 2023) [45] and used as queries to identify homologous sequences through BLASTP analysis (default parameters) against the genomes of the following species: *A. hypogaea* (accessed on 10 May 2023) [46,47,48,49], *A. duranensis* (accessed on 10 May 2023) [52], *A. ipaensis* (accessed on 10 May 2023) [52], *A. monticola* (accessed on 18 May 2023) [51], *G. max* (accessed on 21 September 2024) [131], *G. soja* (accessed on 21 September 2024) [131], *M. truncatula* (accessed on 21 September 2024) [132], *H. annuus* (accessed on 21 September 2024) [133], *Gossypium hirsutum* (accessed on 23 September 2024) [134], *R. communis* (accessed on 23 September 2024) [135], and *T. cacao* (accessed on 23 September 2024) [136] (see Appendix A). The CDSs of putative *AhLEC1s*, *AhFUS3s*, and *AhWRIs* (Appendix A) were cloned from HY917 using primers (Appendix A) designed based on their predicted nucleotide sequences from *A. hypogaea* genome databases. The CDS, genomic DNA and protein sequences of putative LEC1, FUS3, and WRI homologs from other plant species were downloaded from their respective genome databases (Appendix A).

### 4.3. Construction of Plant Expression Vectors and Genetic Transformation

For subcellular localization analysis, *AhLEC1b*, *AhFUS3b*, *AhWRI1a*, and *AhWRI1d* were individually cloned into the plant expression vector pAN580 (*P_35S_::MCS::GFP*). The empty vector pAN580 was used as a control. Each construct was co-transformed with a nuclear-localized marker (*P35S::NLS::mCherry*) into *Arabidopsis* protoplasts as previously described [137]. Images were captured under bright field, GFP, and mCherry fluorescence channels using a confocal laser-scanning microscopy (Zeiss LSM 510 META; Carl Zeiss Microscopy GmbH, Jena, Germany).

A tandem gene cassette, *P_35S_::AhLEC1b::T_NOS_::P_MAS_::AhFUS3b::CaMV poly(A) signal::P_NOS_::AhWRI1a::CaMV poly(A) signal::P_35S_::AhWRI1d::T_PE9_*, was designed, synthesized, and inserted into the binary expression vector pCAMBIA2300 by Shaanxi Jiyingjia Biotechnology Co., Ltd., Xi’an, China (Figure 8A). The resulting construct was introduced into *A. tumefaciens* C58, and then transformed into *A. thaliana* (wild-type Col-0) via the floral dip method [138]. Putative transgenic seeds were surface-sterilized with 10% NaClO and 0.5% Triton X100 for 5 min, rinsed four times with sterile distilled water, and plated on half-strength MS medium containing 50 mg L^−1^ Kanamycin. Resistant plants were then transferred to square pots filled with a 1:1 (*v*/*v*) mixture of peat-based compost and vermiculite and grown under the conditions as described above.

### 4.4. Analysis of Phylogenetic Tree, Conserved Domain, Physicochemical Properties, Sequence Alignments, Gene Structures, Identity, and PEST Motifs

Protein sequences were used to construct phylogenetic trees with MEGA11 software using the Maximum Likelihood method and the Jones Taylor Thornton (JTT) model with 2000 bootstrap replicates [53], to identify conserved domains using Batch CD-Search in the Conserved Domain Database (CDD) of NCBI (https://www.ncbi.nlm.nih.gov/Structure/bwrpsb/bwrpsb.cgi, accessed on 22 October 2024) [55] under automatic search mode, and to predict the theoretical molecular weight (MW) and isoelectric point (pI) using the ExPASy ProtParam tool (https://web.expasy.org/protparam/, accessed on 22 October 2024). Multiple sequence alignments were generated using CLC Sequence Viewer 6.8.1. Identity analyses of both conserved regions and full-length protein sequences were conducted with MegAlign software (Sequence Distances algorithm) in DNASTAR (Lasergene.v7.1 suite). Gene structures were illustrated based on genomic DNAs and CDSs using the Gene Structure Display Serve (GSDS2.0; https://gsds.gao-lab.org/, accessed on 20 October 2024) [54]. Subcellular location was predicted using ProtComp v.9.0 in softberry (http://www.softberry.com/; accessed on 20 December 2024). Nuclear localization signals (NLSs) were identified with cNLS Mapper (https://nls-mapper.iab.keio.ac.jp/cgi-bin/NLS_Mapper_form.cgi, accessed on 22 January 2025) [83]. Putative PEST motifs were detected using the epestfind tool (https://emboss.bioinformatics.nl/cgi-bin/emboss/epestfind, accessed on 22 May 2025).

### 4.5. Chromosomal Location and Gene Synteny Analysis

Chromosome locations of *LEC1s*, *FUS3s*, and *WRIs* in *A. hypogaea*, *A. duranensis*, *A. ipaensis*, and *A. monticola* were acquired from the GFF genome files downloaded from their respective genome databases (Appendix A). The positions were visualized on chromosomes using on the chromosomes using Gene Location Visualized from GTF/GFF in TBtools (v2.142) [139]. Gene synteny analyses were performed using the MCScanX algorithm implemented TBtools (v2.142) [139].

### 4.6. Analysis of Expression Patterns

The transcriptional patterns of *AhLEC1s*, *AhFUS3s*, and *AhWRIs* in different tissues across various development stages of peanut were obtained from the Peanut Genome Resource (PGR, accessed on 11 June 2024) and PeanutBase (accessed on 11 June 2024) [73,74]. Expression values were normalized by log_2_(FPKM + 1) for PGR and log_2_(TPM + 1) for PeanutBase, respectively, and visualized as HeatMaps using TBtools (v2.142) [139]. The transcriptional profiles of these genes under various stresses and phytohormone treatments were also retrieved from both databases.

Expression levels of *AhLEC1s*, *AhFUS3s*, and *AhWRIs* were analyzed across six stages of seed development of four peanut sister lines (P19-19, P19-61, P19-57, and HY917) based on our RNA-seq data (unpublished). Total RNA was extracted from 72 tissue samples using E.Z.N.A.^®^ Plant RNA Kit (Cat#R6827; Omega Bio-tek, Norcross, GA, USA) following the manufacturer’s protocol. Contaminating genomic DNA was removed using RNase-Free DNase I Set (Cat#E1091; Omega Bio-tek, Norcross, GA, USA). cDNA libraries were constructed with the NEBNext^®^ Ultra^TM^ RNA II Library Prep Kit for Illumina^®^ (NEB #E7770; New England Biolabs (NEB), Ipswich, MA, USA) and sequenced on an Illumina NovaSeq 6000 platform (2 × 150 bp paired-end reads) at Wuhan Benagen Technology Company Limited. (Wuhan, China), according to the manufacturer’s instructions. Raw paired-end reads were processed with fastp [140], and data quality was evaluated with FastQC (https://www.bioinformatics.babraham.ac.uk/projects/fastqc/; accessed on 12 October 2023). Clean reads were aligned to the peanut reference genome (Tifrunner.gnm2.ann1.4K0L; https://data.legumeinfo.org/Arachis/hypogaea/annotations/Tifrunner.gnm2.ann1.4K0L/; accessed on 18 October 2023) using STAR [141]. The resulting alignments were sorted indexed, and quantified using RSEM [142] to obtain unique read counts per gene. Gene expression levels were normalized and reported as FPKM (Fragments Per Kilobase per Million mapped fragments). The FPKM values or *AhLEC1s*, *AhFUS3s*, and *AhWRIs* are provided in Appendix A.

Expression data of *AtLEC1*, *AtL1L*, *AtFUS3*, *AtWRI1*, *AtWRI2*, *AtWRI3*, and *AtWRI4* were obtained from publicly available *Arabidopsis* microarray datasets through the *Arabidopsis* eFP browser (https://bar.utoronto.ca/efp/cgi-bin/efpWeb.cgi; accessed on 12 June 2024) [143].

### 4.7. Analyses of Cis-Acting Elements and miRNAs Targeting AhLEC1s, AhFUS3s, and AhWRIs

*Cis*-acting elements were predicted using the putative promoter regions (2000 bp upstream of the start-codon) of *LEC1s*, *FUS3s*, and *WRIs* from *Arachis hypogaeaLine8* v1.3 and and *A. thaliana* Araport11 using PlantCARE (https://bioinformatics.psb.ugent.be/webtools/plantcare/html/; accessed on 12 October 2024) [76], and the results were then visualized with TBtools (v2.142) [139]. The CDSs of *LEC1s*, *FUS3s*, and *WRIs* from HY917 and *A. thaliana* were submitted to psRNATarget (https://www.zhaolab.org/psRNATarget/, accessed on 22 March 2025) [77] to identify potential targeting miRNAs under default parameters.

### 4.8. Analysis of the Content and Composition of Fatty Acids in Seeds

The mature seeds used for FA analysis were harvested from the lower part of the main stem of plants grown in different pots arranged randomly in one of three blocks. Seed FAs were extracted and analyzed as previously reported [144]. In detail, the total FAs were converted to FA methyl esters (FAMEs) in 1 M HCl-methanol at 80 °C for 2 h. FAMEs were analyzed via by GC–MS (GC-QQQ, 7890A-7001B, Agilent Technologies, Santa Clara, CA, USA) equipped with an HP-FFAP capillary column (30 m × 0.25 mm ID, 0.25 μm film thickness). Helium was used as the carrier gas at a flow rate of 1.0 mL/min with a split ratio of 5:1. The injection volume was 1.0 μL, and the injection temperatures were held at 250 °C. The temperature of the column oven was programmed from 40 °C (held for 5 min) to 250 °C (held for 5 min) at 8 °C/min. Mass spectrometry conditions were as follows: the energy of EI ionization source was 70 eV, the ion source temperature was 230 °C, the quadrupole temperature was 150 °C, the interface temperature was 250 °C, and the scan range was 30–400 *m*/*z*. Heptadecanoic acid (C17:0; Sigma-Aldrich, Saint Louis, MO, USA) was added as internal standard prior to extraction. Analyses were performed in biological triplicates.

### 4.9. Measurement of Thousand-Seed Weight, Plant Height, and Silique Length, and Warm Germination Test

Mature seeds used for thousand-seed weight analysis were collected from the lower portion of the main stems. The seeds were dried in open tubes in an oven for one week and then weighed using an electronic microbalance. Plant height and the silique length of main stem were measured on mature plants using 50 cm and 10 cm rulers, respectively.

For the warm germination test, mature seeds were evenly arranged in a row on a moist filter paper, covered with another moist filter paper, and then rolled up. The roll was wrapped in waxed paper and placed in a large beaker containing 2.5 cm (1 inch) of water at the bottom. The beaker was incubated in a growth chamber at 25 °C. Germination rates were assessed seven days after seeding.

### 4.10. Quantitative RT–PCR (qRT–PCR) Analysis of Gene Expression

The total RNA was extracted from *Arabidopsis* siliques at 15 days after flowering (DAF) using the RNAprep Pure Plant Kit (Cat#DP441, TIANGEN BIOTECH Co., Ltd., Beijing, China). First-strand cDNA was synthesized from 1 μg of total RNA using the FastKing RT Kit (with gDNase) (Cat#KR116, TIANGEN BIOTECH Co., Ltd., Beijing, China), according to the manufacturer’s instructions. Each cDNA sample was diluted 10–15-fold in sterile water prior to qRT–PCR analysis.

qRT–PCR was carried out on a CFX Connect^TM^ Real-Time PCR system (Bio–Rad, Hercules, CA, USA) with EvaGreen 2× qPCR MasterMix (MasterMix-S, Abm, Vancouver, BC, Canada), following the manufacturer’s protocol. The expression levels of the target genes were normalized using *Arabidopsis AtActin* as an internal control. Relative expression level was calculated by 2^−ΔΔCt^. The primers used are listed in Appendix A. All reactions were performed in biological triplicates.

### 4.11. Statistical Analysis

All the data are presented as the mean ± standard deviation at least three replicates. Statistical analyses were conducted in Excel 2016 and GraphPad Prism 6.02. Significant differences were determined by one-way analysis of variance (ANOVA) followed by Dunnett’s post hoc test. All bar charts were generated using GraphPad Prism 6.02.

## 5. Conclusions

In this study, two *AhLEC1s*, two *AhFUS3s*, nine *AhWRI1s*, two *AhWRI2s,* and four *AhWRI3s* were cloned from *A. hypogaea* HY917 and comprehensively analyzed using bioinformatics and expression profiling. The results revealed that *AhLEC1a/b*, *AhFUS3a/b*, and *AhWRI1a/b/c*, which are specifically expressed in developing peanut seeds, likely play critical roles in seed development, while *AhWRI1d*, *AhWRI1f*, and *AhWRI1g*, highly expressed in root nodules, may be involved in nodulation. Co-overexpression of *AhLEC1b*, *AhFUS3b*, *AhWRI1a*, and *AhWRI1d* in *Arabidopsis* significantly increased seed fatty acid content and thousand-seed weight. However, these improvements were accompanied by adverse agronomic traits, including reduced plant height, shorter siliques, and lower germination rates, highlighting a trade-off between lipid metabolism and plant growth. In summary, *AhLEC1*, *AhFUS3*, and *AhWRI* genes may play crucial roles in regulating oil synthesis and seed development in peanuts. Future work should prioritize using seed-specific promoters to drive the expression of these genes, thereby minimizing negative effects on plant growth. Further functional characterization in both peanut and *Arabidopsis* will be necessary to fully exploit their potential for improving oil traits in crops.

## Figures and Tables

**Figure 1 plants-14-02910-f001:**
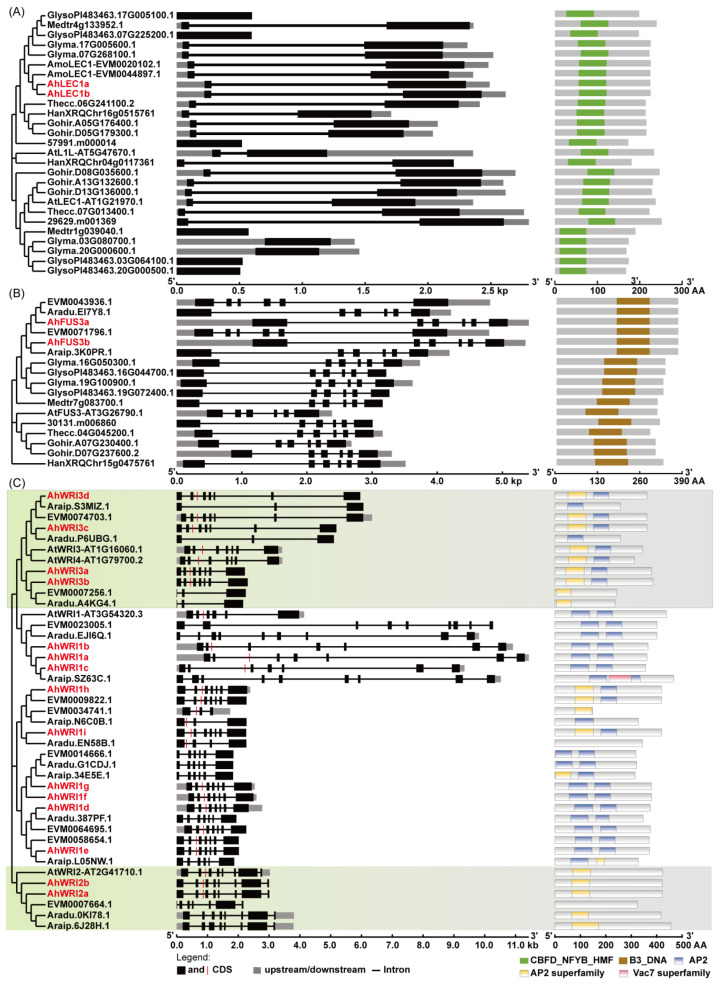
Phylogenetic relationships, gene structures, and conserved domains of the LEC1 (**A**), FUS3 (**B**), and WRI (**C**) family members in *A. hypogaea* and other plant species. Phylogenetic trees were constructed based on protein sequences using the Maximum Likelihood method with the Jones–Taylor–Thornton (JTT) model of MEGA11 [53]. Proteins from peanut are highlighted in red. Gene structures were illustrated using the Gene Structure Display Serve (GSDS2.0; https://gsds.gao-lab.org/; accessed on 20 October 2024) based on genomic DNA and coding sequences [54]. Black boxes denote exons within coding regions, lines represent introns, and thin red boxes mark the conserved 9-bp micro-exons (encoding VYL; Figure 2C,D and Appendix A) specific to *WRIs*. Gray boxes indicate untranslated upstream or downstream regions. The lengths of boxes and lines correspond to the size of exons and introns, respectively. Conserved domains were identified by searching full-length protein sequences using Batch CD-Search with default parameters in the Conserved Domain Database (CDD) in NCBI (https://www.ncbi.nlm.nih.gov/Structure/bwrpsb/bwrpsb.cgi; accessed on 22 October 2024) [55]. The CDSs of *A. hypogaea LEC1*, *FUS3*, and *WRI* family members were cloned from HY917 (Appendix A), while the genomic DNA sequences were retrieved from the PGR and PeanutBase databases (Appendix A). Corresponding sequences in other plant species were obtained from their respective genome databases (Appendix A). All LEC1s contain one CBFD_NFYB_HMF domain (pfam00808). All FUS3 homologs share one B3 DNA-binding domain (B3_DNA, cd10017). All AhWRI1s and AhWRI3s have two AP2 domains (smart00380 and cl00033 for AP2 and AP2 superfamily, respectively), consistent with WRI1 and WRI3 homologs in other plants (Appendix A), excepting Araip.S3MIZ.1, Aradu.P6UBG.1, EVM0007256.1, Aradu.A4KG4.1, EVM0034741.1, Araip.N6C0B.1, and Aradu.EN58B.1 (only one or no AP2 domain). AhWRI2a and AhWRI2b contain one AP2 domain (smart00380), similarly to other WRI2s proteins (Appendix A), except for EVM0007664.1 (no AP2 domain).

**Figure 2 plants-14-02910-f002:**
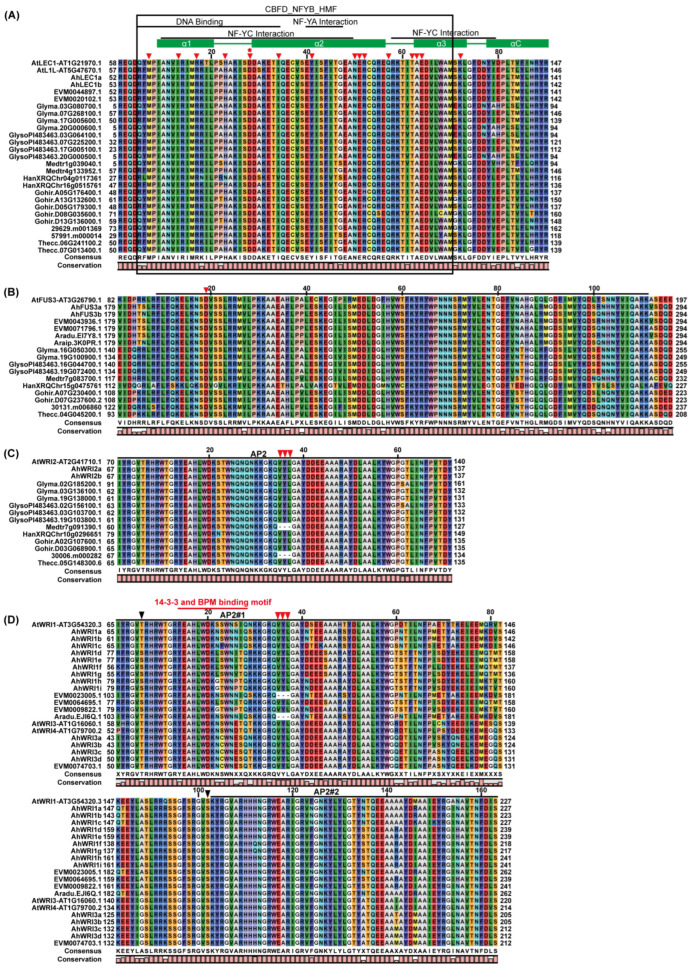
Alignments of the conserved domains of LEC1, FUS3, and WRI family members in *A. hypogaea* and other plant species. (**A**) Alignment of the conserved B domains of LEC1s. Black lines indicate the DNA-binding and subunit interaction domains [56]. Putative CBFD_NFYB_HMFs are boxed. Secondary structures [57], α-helices (solid green rectangles) and coils (green lines), are shown above the alignment. Red triangles mark amino acids conserved in the B-domains of AtLEC1 and AtL1L but divergent in other AtNF-YBs [11,12]. A conserved residue Asp (**D**) residue essential for AtLEC1 function [12] is indicated by a red star. (**B**) Alignment of the conserved B3 domains of FUS3 orthologs. The B domains of AhFUS3a and AhFUS3b show high similarity to those of other plant FUS3s. Red triangles indicate aspartic acid residues equivalent to D75 in HvFUS3; a D75N mutation in HvFUS3 was shown to impair DNA-binding ability [58]. (**C**) Alignment of the conserved AP2 domains of WRI2s. The only AP2 conserved domains of two AhWRI2s demonstrate strong similarity to those of their orthologs from other plants. The VYL motif within the AP2 domain is indicated by three red triangles. (**D**) Alignment of the two conserved AP2 domains of WRI1s and WRI3s. The two AP2 conserved domains, designated AP2#1 and AP2#2, are indicated with black lines. The VYL motif within the AP2#1 is marked with three red triangles. Numbers on the left and right indicate the start and end positions of the conserved domains within each protein; numbers above the alignment serve as reference points. In the consensus line, uppercase letters and “X” indicate residues conserved in more and less than 50% identity in the sequences, respectively. Different amino acid residues are distinguished by color.

**Figure 3 plants-14-02910-f003:**
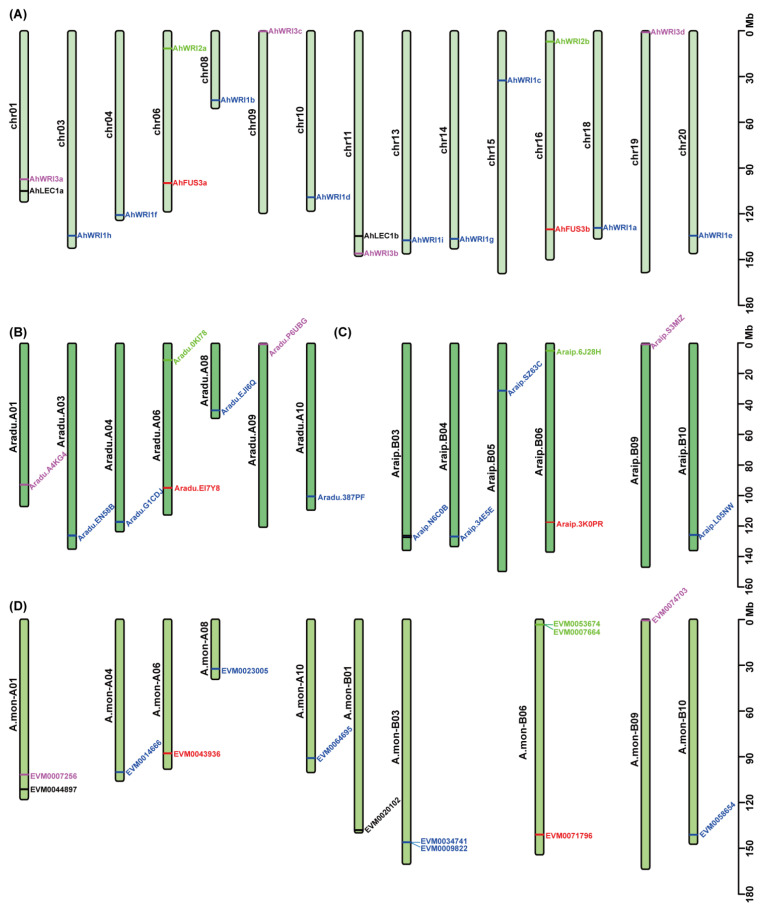
Chromosomal distribution of *LEC1s*, *FUS3s*, and *WRIs* of *A. hypogaea* (**A**), *A. duranensis* (**B**), *A. ipaensis* (**C**), and *A. monticola* (**D**). *LEC1s*, *FUS3s*, *WRI1s*, *WRI2s*, and *WRI3s* are indicated by black, red, blue, green, and purple, respectively.

**Figure 4 plants-14-02910-f004:**
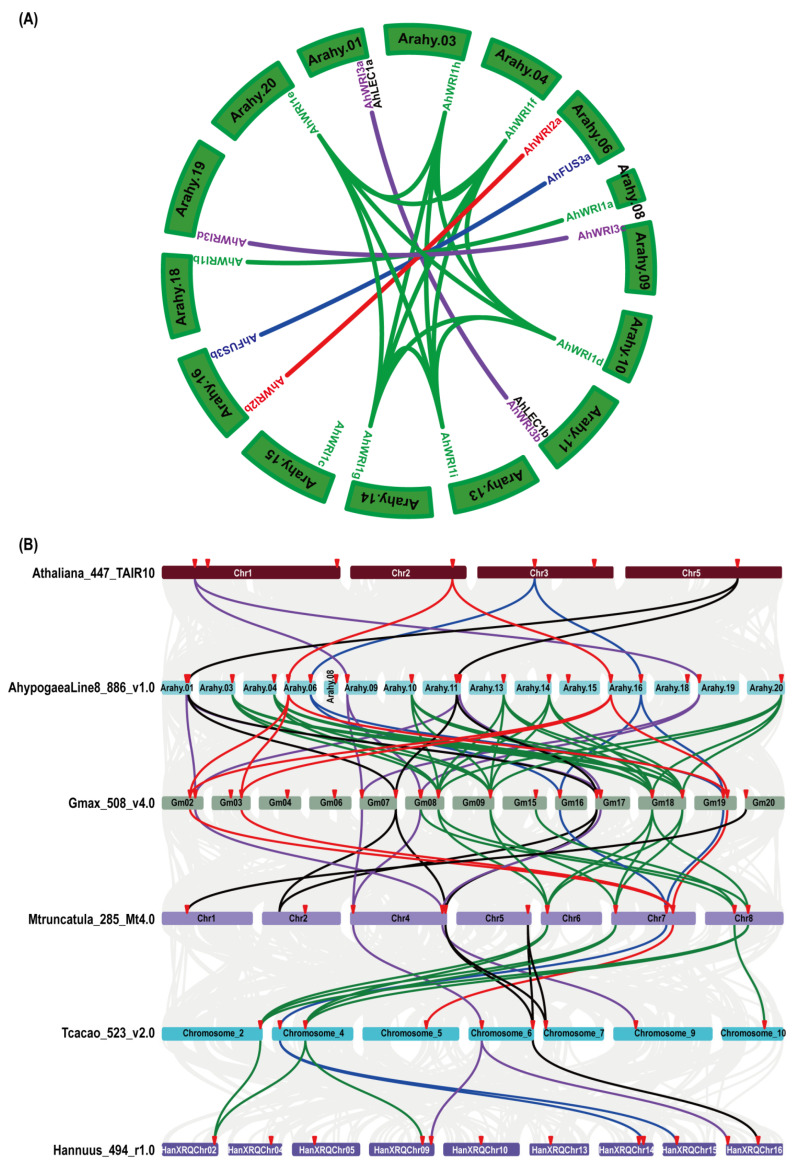
Synteny analysis of *LEC1s*, *FUS3s*, and *WRIs* in *A. hypogaea* and other plants. (**A**) Synteny analysis of genes in *A. hypogaea*. Blue-, green-, red-, and purple-colored lines indicate *AhFUS3s*, *AhWRI1s*, *AhWRI2s*, and *AhWRI3s*, respectively. (**B**) Synteny analysis of genes in *A. hypogaea* and other plant species. Black-, blue-, green-, red-, and purple-colored lines indicate the syntenic gene pairs of *LEC1s*, *FUS3s*, *WRI1s*, *WRI2s*, and *WRI3s* between the denoted species, respectively. The gray lines represent collinear blocks. Red triangles indicated the positions of *LEC1s*, *FUS3s*, *WRI1s*, *WRI2s*, and *WRI3s* on chromosomes.

**Figure 5 plants-14-02910-f005:**
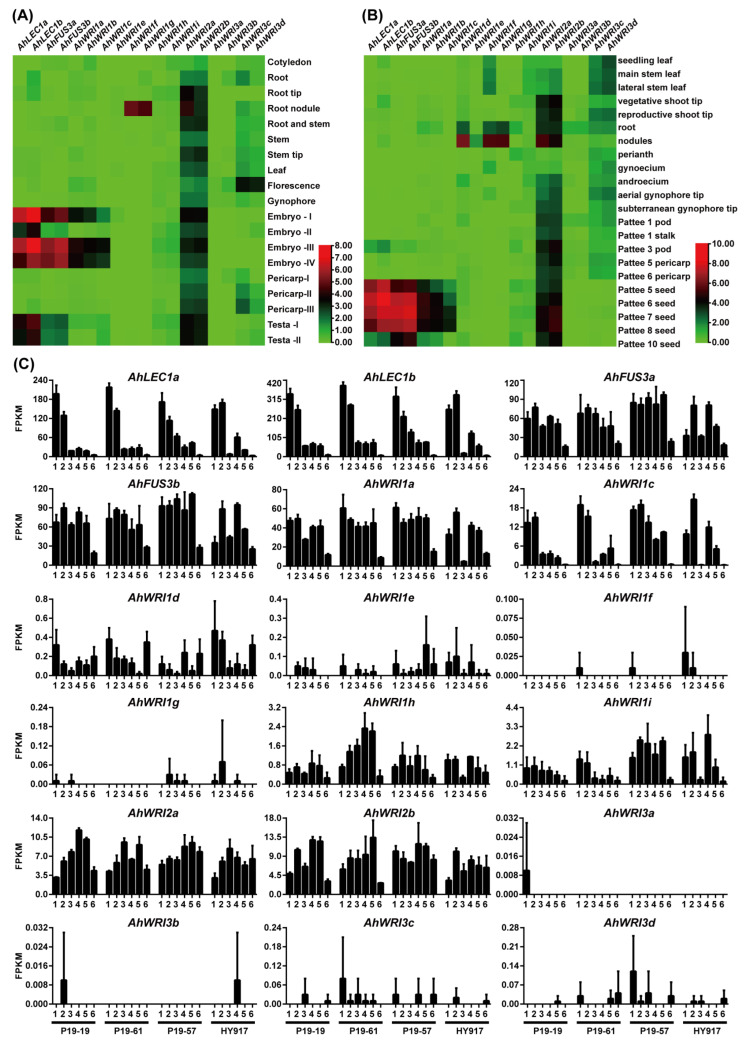
Expression patterns of *AhLEC1s*, *AhFUS3s*, and *AhWRIs* in the different tissues of *A. hypogaea*. Expression data were obtained from (**A**) PGR (FPKM, Fragments Per Kilobase of transcript per Million mapped reads) and (**B**) PeanutBase, using the Tifrunner.gnm2.ann2.PVFB annotation (TPM, Transcripts Per Million). Expression data for *AhWRI1d* was not available in the PGR database. For genes with multiple accessions in PeanutBase (*AhWRI1f* and *AhWRI1h*), expression values represent the normalized sum of TPM for all constituent models. The extracted FPKM and TPM values were transformed using log_2_(FPKM + 1) and log_2_(TPM + 1) normalization, respectively. The gynophore is a unique specialized organ that forms after pollination, which elongates and drives the developing ovary into the ground for fruit and seed development. “Pattee #” refers to seed samples categorized based on the Pattee maturity scale [75], which is a standard system for defining peanut pod and kernel maturity based on morphological and color characteristics. (**C**) Expression patterns of *AhLEC1s*, *AhFUS3s*, and *AhWRIs* across six seed developmental stages in four peanut sister lines (P19-19, P19-61, P19-57, and HY917; see Appendix A for developmental staging). The RNA-seq data were initially mapped to the Tifrunner.gnm1.ann1.4K0L genome annotation. In this annotation, a model for *AhWRI1b* was not availabe. Two partial gene models (*Arahy.4D3TQE.1* and *Arahy.RC4286.1*) were annotated as *AhWRI1h* based on sequence homology of their CDSs and encoded proteins; the expression value shown for *AhWRI1h* represents the normalized sum of FPKM from these two accessions. Data are presented as mean ± SD (*n* = 3 biological replicates).

**Figure 6 plants-14-02910-f006:**
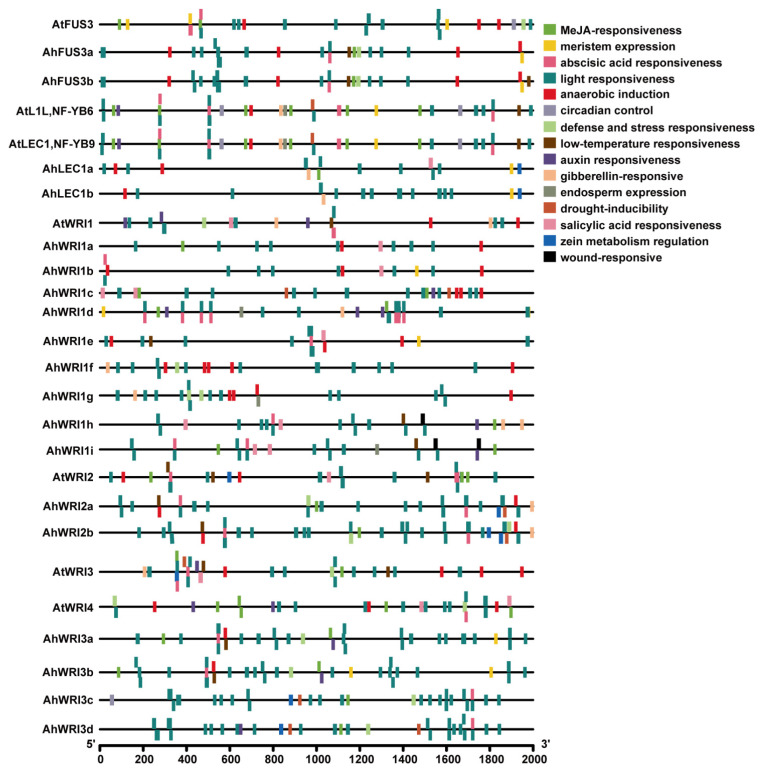
Analysis of *cis*-acting elements in the putative promoters of *AhLEC1s*, *AhFUS3s*, and *AhWRIs*. The 2000 bp upstream sequences of their start codons extracted from *Arachis hypogaeaLine8* v1.3 (accessed on 10 October 2024) and TAIR were used to analyze the *cis*-regulatory elements using plantCARE (https://bioinformatics.psb.ugent.be/webtools/plantcare/html/, accessed on 12 October 2024) [76].

**Figure 7 plants-14-02910-f007:**
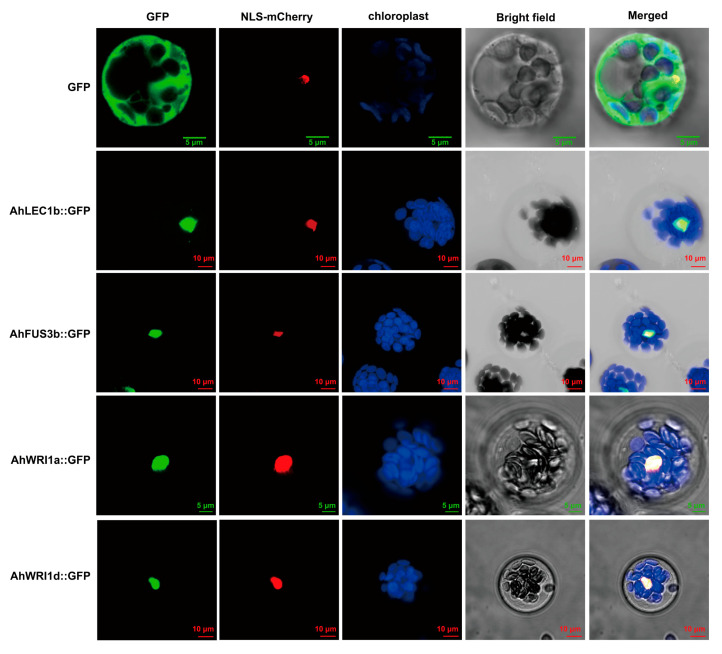
Subcellular localization of AhLEC1b, AhFUS3b, AhWRI1a, and AhWRI1d. Confocal images are from *Arabidopsis* protoplast cells transiently co-expressing *NLS-mCherry* with *GFP*, *AhLEC1b::GFP*, *AhFUS3b::GFP*, *AhWRI1a::GFP*, and *AhWRI1d::GFP*, respectively. GFP: green fluorescence; NLS-mCherry: mCherry red fluorescence; chloroplast: chloroplast fluorescence, indicated by blue. Green scale bars denote 5 μm and red scale bars denote 10 μm.

**Figure 8 plants-14-02910-f008:**
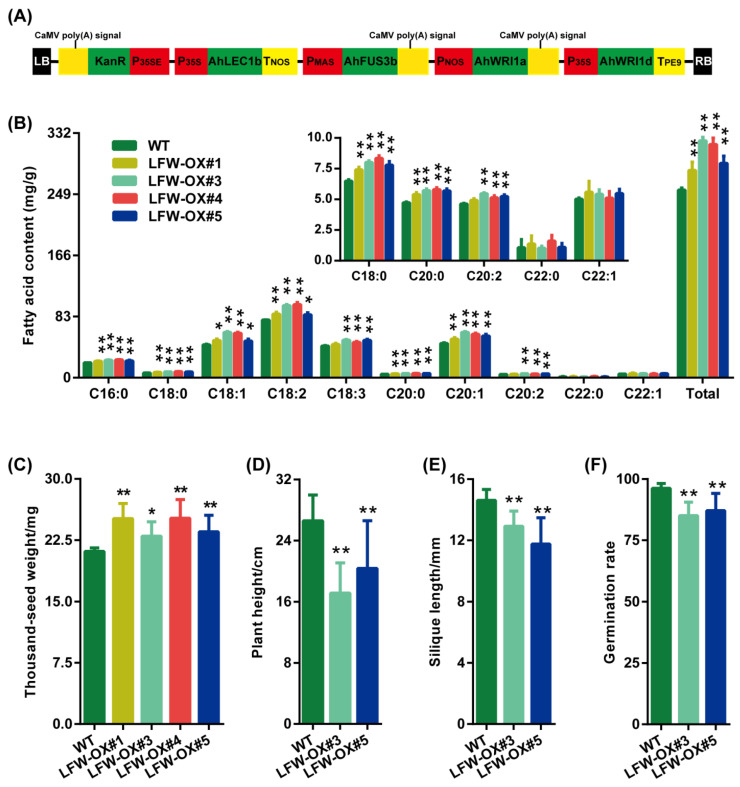
Analysis of different fatty acid contents, thousand-seed weight, plant height, silique length, and seed germination rate. (**A**), Schematic diagram of the constitutive expression cassette of tandem genes *AhLEC1b*, *AhFUS3b*, *AhWRI1a*, and *AhWRI1d*. LB, left border; KanR, *aphA-3* coding aminoglycoside phosphotransferase and conferring resistance to kanamycin; P_35SE_, CaMV 35S promoter (enhanced); P_35S_, CaMV 35S promoter; T_NOS_, nopaline synthase terminator; P_MAS_, mannopine synthase promoter; P_NOS_, nopaline synthase promoter; T_PE9_, pea rbcS E9 terminator; RB, right border. Comparison of different fatty acid contents (*n* = 3) (**B**), thousand-seed weight (*n* ≥ 12 (**C**), plant height (*n* ≥ 31) (**D**), silique length (*n* ≥ 26) (**E**), and seed germination rate (*n* ≥ 8) (**F**), between wild type and transgenic plants. Asterisks indicate significant differences between the wild type and transgenic plants. Data are presented as mean ± SD. * *p* < 0.05, ** *p* < 0.01 vs. WT (one-way ANOVA followed by Dunnett’s test).

**Table 1 plants-14-02910-t001:** Members of LEC1, FUS3, and WRIs in *A. hypogaea*, *A. duranensis*, *A. ipaensis*, and *A. monticola*.

*A. thaliana*	*A. hypogaea*					*A. monticola*	*A. duranensis*	*A. ipaensis*
	Gene Name	PGR	Tifrunner	BaileyII	Line8 v1.3			
AtLEC1 AtL1L	*AhLEC1a*	AH01G29730	Ah01g388600	chr01G4145	AhLine8.01G229500	EVM0044897	—	—
	*AhLEC1b*	AH11G27150	Ah11g380300	chr11G3925	AhLine8.11G179600	EVM0020102	—	—
AtFUS3	*AhFUS3a*	AH06G23930	Ah06g316900	chr06G3370	AhLine8.06G179300	EVM0043936	Aradu.EI7Y8	—
	*AhFUS3b*	AH16G29570	Ah16g401200	chr16G3980	AhLine8.16G197300	EVM0071796	—	Araip.3K0PR
AtWRI1	*AhWRI1a*	AH08G25830	Ah08g309800	chr08G3455	AhLine8.08G210300	EVM0023005	Aradu.EJI6Q	—
	*AhWRI1b*	AH18G30790	Ah18g415600	chr18G4478	AhLine8.18G216000	—	—	—
	*AhWRI1c*	AH15G12290	Ah15g212800	chr15G2176	AhLine8.15G128400	—	—	Araip.SZ63C
	*AhWRI1d*	—	Ah10g334400	IDmodified-mrna-2114	AhLine8.10G185300	EVM0064695	Aradu.387PF	—
	*AhWRI1e*	AH20G31430	Ah20g425400	chr20G4248	AhLine8.20G202100	EVM0058654	—	Araip.L05NW
	*AhWRI1f*	AH04G29530	Ah04g393900	chr04G4066	AhLine8.04G202100	EVM0014666	Aradu.G1CDJ	—
			Ah04g396000					
			Ah04g394100					
	*AhWRI1g*	AH14G34420	Ah14g462800	IDmodified-mrna-77	AhLine8.14G213200	—	—	Araip.34E5E
	*AhWRI1h*	AH03G41970	Ah03g520600	chr03G5733	AhLine8.03G308300	EVM0009822	—	Araip.N6C0B
			Ah03g520700					
	*AhWRI1i*	AH13G44600	Ah13g549900	chr13G6111	AhLine8.13G320200	EVM0034741	Aradu.EN58B	—
AtWRI2	*AhWRI2a*	AH06G02150	Ah06g106000	chr06G1256	AhLine8.06G083900	—	Aradu.0KI78	—
					AhLine8.06G084200			
	*AhWRI2b*	AH16G04650	Ah16g055500	chr16G566	AhLine8.16G038500	EVM0053674	—	Araip.6J28H
						EVM0007664		
AtWRI3 AtWRI4	*AhWRI3a*	AH01G23420	Ah01g316300	IDmodified-mrna-7040	AhLine8.01G177800	EVM0007256	Aradu.A4KG4	—
	*AhWRI3b*	AH11G34510	Ah11g474800	chr11G4916	AhLine8.11G236600	—	—	—
	*AhWRI3c*	AH09G00410	Ah09g004900	IDmodified-mrna-2659	AhLine8.09G003900	—	Aradu.P6UBG	—
	*AhWRI3d*	AH19G01070	Ah19g015300	chr19G149	AhLine8.19G012100	EVM0074703	—	Araip.S3MIZ

The accession numbers of LEC1, FUS3, and WRI members in *A. hypogaea*, *A. duranensis*, *A. ipaensis*, and *A. monticola* were derived from the PGR [46], PeanutBase [47,48,52], *Arachis hypogaeaLine8* v1.3 [49,50], and an allotetraploid wild peanut *A. monticola* genome [51], respectively.

## Data Availability

Data are contained within the article and Appendix A.

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
