# Peer review of "Genome-Wide Identification of Arachis hypogaea LEC1s, FUS3s, and WRIs and Co-Overexpression of AhLEC1b, AhFUS3b, AhWRI1a and AhWRI1d Increased Oil Content in Arabidopsis Seeds"

_plants, 2025, doi:10.3390/plants14182910_

Round 1
Reviewer 1 Report
Comments and Suggestions for Authors
This publication seems to be within the scope of journal. However, it needs some corrections to be more acceptable for publication.
Please complete the description of the fatty acid analysis method with the following data: chromatographic column packing, temperature program, injector temperature, detector temperature, carrier gas and flow rate, split, injection size, and MS characteristics. Please also specify the standard used for MS calibration.
Please check the entire manuscript to see if Latin names are written in italic everywhere.
Author Response
Comments 1: This publication seems to be within the scope of journal. However, it needs some corrections to be more acceptable for publication.
Response 1: We sincerely appreciate your assessment of our manuscript's suitability for Plants. We have comprehensively addressed all reviewer comments. We believe these revisions significantly strengthen the manuscript's alignment with the journal's scope and standards.
Comments 2: Please complete the description of the fatty acid analysis method with the following data: chromatographic column packing, temperature program, injector temperature, detector temperature, carrier gas and flow rate, split, injection size, and MS characteristics. Please also specify the standard used for MS calibration.
Response 2: The mature seeds for FA analysis were harvested from the lower part of the main stem of plants grown in different pots arranged randomly within one of three blocks. Seed FAs were extracted and analyzed as reported previously [144]. In detail, total FAs were converted to FA methyl esters (FAMEs) in 1 M HCl-methanol at 80°C for 2 hours. FAMEs were analyzed via by GC–MS (GC-QQQ, 7890A-7001B, Agilent Technologies, Santa Clara, CA, USA) equipped with an HP-FFAP capillary column (30 m × 0.25 mm ID, 0.25 μm film thickness). Helium was used as the carrier gas at a flow rate of 1.0 mL/min with a split ratio of 5:1. The injection volume was 1 μL, and the injection temperatures was held at 250 °C. The temperature of the column oven was programmed from 40°C (held for 5 min) to 250°C (held for 5 min) at 8 °C/min. Mass spectrometry conditions were as follows: the energy of EI ionisation source was 70 eV, the ion source temperature was 230°C, the quadrupole temperature was 150°C, the interface temperature was 250°C, and the scan range was 30–400 m/z. Heptadecanoic acid (C17:0; Sigma-Aldrich) was added as internal standard prior to extraction. Analyses were performed in biological triplicates.
Comments 3: Please check the entire manuscript to see if Latin names are written in italic everywhere.
Responses 3: We have thoroughly checked the entire manuscript and supplementary materials, ensuring all Latin names are now consistently formatted in italics.
All Latin names include Arachis hypogaea, Brassica napus, Glycine max, Arabidopsis thaliana, Camelina sativa, Arachis duranensis, Arachis ipaensis, Arachis monticola Glycine soja, Medicago truncatula, Theobroma cacao, Helianthus annuus, Gossypium hirsutum, Agrobacterium tumefaciens, Ricinus communis, Brachypodium distachyon, Cocos nucifera, and Persea americana.
Particular, “Arachis hypogaeaLine8” in Arachis hypogaeaLine8 v1.3, the Line8 genome assembly and annotation files, was also italicized according to Phytozome 14 (https://phytozome-next.jgi.doe.gov/info/AhypogaeaLine8_v1_3).
All changes were verified using Word's "Find" function and manual inspection.

Reviewer 2 Report
Comments and Suggestions for Authors
The manuscript, titled "Genomic Identification of Arachis hypogaea LEC1, FUS3, and WRI Genes and Co-expression of AhLEC1b, AhFUS3b, AhWRI1a, and AhWRI1d Genes Increasing Oil Content in Arabidopsis Seeds," presents a detailed genomic characterization of the key transcription factors involved in oil biosynthesis in peanut, combined with functional validation in Arabidopsis. This work is timely and relevant to both plant molecular biology and oilseed breeding. The breadth of analyses, from phylogenetics to cis-element exploration and phenotypic evaluation, is remarkable.
However, some elements require refinement to improve the clarity, interpretability, and novelty of the manuscript.
General Comments
The study identifies several members of the LEC1, FUS3, and WRI gene families in peanut, but similar approaches have been undertaken for other oilseeds. The novelty of this study should be more explicitly explained in the introduction and discussion. For example, what does co-overexpression of the peanut orthologs reveal that previous studies in Arabidopsis, maize, or Brassica have not?
The tissue-specific expression profiles are well presented, but their biological significance is poorly developed. For example, the high expression of certain WRI1 variants in roots and nodules could indicate roles beyond seed oil biosynthesis; this should be discussed in relation to carbon allocation and the regulation of plant development.
The co-overexpression experiments increased oil content and seed weight, but also reduced germination rate, plant height, and silique length. These trade-offs should be addressed in the discussion, examining their potential influence on breeding strategies and whether negative traits are separable from increased oil content.
The manuscript states that the differences were significant but does not specify whether the assumptions of the parametric tests were met. The materials and methods should include a clear description of the statistical analyses, including software, significance levels, and normality/homoscedasticity checks. Figures 5 and 8 are rich in information but visually dense. Adding clearer sub-panel labels and expanding the figure legends to summarize key takeaways would facilitate reader understanding.
In Figure 8, the order of character presentation in the text should correspond to the sequence of figures to facilitate cross-referencing.
Although the cataloging of cis elements is comprehensive, the discussion is too general. Highlighting a few key motifs with well-known regulatory implications for seed development and linking them to the observed expression profiles would strengthen the impact of this section.
Minor Comments
- The summary could be simplified: reduce the exhaustive list of identified genes and emphasize the main findings and their implications.
- Correct minor grammatical errors :
Example:
Line 24 – very similar
Line 91 – multiple important roles
Line 131 – were shown to be
Line 172 – were shown to be
Line 198 – which indicated
Line 201 – hydrophilicity
Line 213 – was shown to be
Line 286 – no expressed
Line 327 – more close to
Line 439 – had more similarity in distribution
Line 509 – AhLEC1d decreased
Line 525 – phylogenetic tree
- Check the consistency of accession numbers between the main text and supplementary material.
- In the introduction, references [14–16] could be supplemented with more recent studies on the regulatory roles of LEC1, FUS3, and WRI genes in oilseed crops.
- Ensure that all abbreviations (e.g., FA for fatty acids) are defined at their first mention.
Author Response
Comments 1: The manuscript, titled "Genomic Identification of Arachis hypogaea LEC1, FUS3, and WRI Genes and Co-expression of AhLEC1b, AhFUS3b, AhWRI1a, and AhWRI1d Genes Increasing Oil Content in Arabidopsis Seeds," presents a detailed genomic characterization of the key transcription factors involved in oil biosynthesis in peanut, combined with functional validation in Arabidopsis. This work is timely and relevant to both plant molecular biology and oilseed breeding. The breadth of analyses, from phylogenetics to cis-element exploration and phenotypic evaluation, is remarkable.
However, some elements require refinement to improve the clarity, interpretability, and novelty of the manuscript.
Responses 1: We sincerely appreciate your recognition of our work's significance and thorough assessment. We have implemented comprehensive revisions to address the concerns regarding clarity, interpretability, and novelty:
- We have updated the Introduction with key recent findings on LEC1, FUS3, and WRI1 regulation in oilseed crops.
- We have engaged MDPI’s professional English editing service to ensure linguistic accuracy and readability.
- We have completed the description of the fatty acid analysis method with the following data: chromatographic column packing, temperature program, injector temperature, detector temperature, carrier gas and flow rate, split, injection size, and MS characteristics.
- We have added the results of qRT-PCR for detecting AhLEC1b, AhFUS3b, AhWRI1a, and AhWRI1d expression in transgenic plants (Figure S8).
- etc
We hope these revisions have significantly improved the clarity and professionalism of the manuscript.
General Comments
Comments 2: The study identifies several members of the LEC1, FUS3, and WRI gene families in peanut, but similar approaches have been undertaken for other oilseeds. The novelty of this study should be more explicitly explained in the introduction and discussion. For example, what does co-overexpression of the peanut orthologs reveal that previous studies in Arabidopsis, maize, or Brassica have not?
Responses 2: We appreciate the reviewer's insightful suggestion to better articulate our study's novel contributions. We have strengthened the manuscript by:
1) Enhanced Introduction:
- Peanut's unique tetraploid genome complicates functional redundancy analysis
- Previous works reported have focused on single-TF effects
- No previous studies have tested LEC1/FUS3/WRI1 combinatorial control in oilseeds
2) Novelty Highlights in Discussion:
- AhLEC1b, AhWRI1a, and AhWRI1d may act synergistically to regulate FA composition and thousand-seed weight.
- AhLEC1b and AhFUS3b may act synergistically to regulate plant architecture and germination efficiency.
- Our study showed that co-overexpression of these regulatory genes significantly modulates lipid metabolism and key agronomic traits. However, the molecular mechanisms underlying the trade-offs between lipid synthesis, seed weight, plant architecture, and germination efficiency remain to be fully elucidated and warrant further investigation.
Comments 3: The tissue-specific expression profiles are well presented, but their biological significance is poorly developed. For example, the high expression of certain WRI1 variants in roots and nodules could indicate roles beyond seed oil biosynthesis; this should be discussed in relation to carbon allocation and the regulation of plant development.
Responses 3: We thank the reviewer for this excellent suggestion to explore the broader biological implications of our expression data. We have significantly expanded the discussion of tissue-specific patterns with these additions (see Discussion):
1) Nodule Expression:
Proposed roles in carbon partitioning during both root nodulation and arbuscular mycorrhiza symbiosis (Chen et al., 2020; Jiang et al., 2018)
Chen, B.; Zhang, G.; Li, P.; Yang, J.; Guo, L.; Benning, C.; Wang, X.; Zhao, J. Multiple GmWRI1s are redundantly involved in seed filling and nodulation by regulating plastidic glycolysis, lipid biosynthesis and hormone signalling in soybean (Glycine max). Plant Biotechnol. J. 2020, 18, 155-171, doi:10.1111/pbi.13183.
Jiang, Y.; Xie, Q.; Wang, W.; Yang, J.; Zhang, X.; Yu, N.; Zhou, Y.; Wang, E. Medicago AP2-domain transcription factor WRI5a is a master regulator of lipid biosynthesis and transfer during mycorrhizal symbiosis. Mol. Plant 2018, 11, 1344-1359, doi:10.1016/j.molp.2018.09.006.
2) Seed-specific Expression:
AhLEC1a/b, AhFUS3a/b, and AhWRI1a/b/c show seed-specific expression (Figure 5A-B), suggesting they may play redundant roles in oil accumulation. AhLEC1a/b and AhWRI1a have demonstrated to play key roles in seed oil accumulation (Tang et al., 2024; Lu et al., 2024).
Tang, G.; Xu, P.; Jiang, C.; Li, G.; Shan, L.; Wan, S. Peanut LEAFY COTYLEDON1-type genes participate in regulating the embryo development and the accumulation of storage lipids. Plant Cell Rep. 2024, 43, 124, doi:10.1007/s00299-024-03209-8.
Lu, Q.; Huang, L.; Liu, H.; Garg, V.; Gangurde, S.S.; Li, H.; Chitikineni, A.; Guo, D.; Pandey, M.K.; Li, S.; et al. A genomic variation map provides insights into peanut diversity in China and associations with 28 agronomic traits. Na. Genet. 2024, 56, 530-540, doi:10.1038/s41588-024-01660-7.
We particularly appreciate the reviewer's insight about carbon allocation - this perspective has revealed unexpected dimensions of our data that merit further investigation.
Comments 4: The co-overexpression experiments increased oil content and seed weight, but also reduced germination rate, plant height, and silique length. These trade-offs should be addressed in the discussion, examining their potential influence on breeding strategies and whether negative traits are separable from increased oil content.
Responses 4: We thank the reviewer for raising this crucial point about the observed trade-offs. We have substantially expanded our discussion of these phenotypic consequences with the following key additions:
1)Mechanistic Interpretation:
- AhLEC1b, AhWRI1a, and AhWRI1d may act function in a coordinated manner to regulate FA composition and thousand-seed weight.
- AhLEC1b and AhFUS3b may act function in a coordinated manner to regulate plant architecture and germination efficiency.
- The molecular mechanisms underlying the trade-offs between lipid synthesis, seed weight, plant architecture, and germination efficiency remain to be fully elucidated and warrant further investigation.
2)Breeding Implications:
- Promoter engineering- Using seed-specific promoters to avoid vegetative impacts
Previous researches have demonstrated promoters must be chosen carefully to avoid negative effects while obtaining sufficient expression levels in target tissues (van Erp et al., 2014; An et al., 2015; Tan et al., 2011; Tang et al., 2018; Tang et al., 2024; Zhu et al., 2018). In particular, co-expression of AtWRI1, AtDGAT1, and RNAi AtSDP1 under seed-specific promoters increase seed oil content without be detrimental to seed vigor and growth phenotypes throughout the life cycle of the plants (van Erp et al., 2014).
An, D.; Suh, M.C. Overexpression of Arabidopsis WRI1 enhanced seed mass and storage oil content in Camelina sativa. Plant Biotechnol. Rep. 2015, 9, 137-148, doi:10.1007/s11816-015-0351-x.
Tan, H.; Yang, X.; Zhang, F.; Zheng, X.; Qu, C.; Mu, J.; Fu, F.; Li, J.; Guan, R.; Zhang, H.; et al. Enhanced seed oil production in canola by conditional expression of Brassica napus LEAFY COTYLEDON1 and LEC1-LIKE in developing seeds. Plant Physiol. 2011, 156, 1577-1588, doi:10.1104/pp.111.175000.
Tang, G.; Xu, P.; Ma, W.; Wang, F.; Liu, Z.; Wan, S.; Shan, L. Seed-Specific Expression of AtLEC1 Increased Oil Content and Altered Fatty Acid Composition in Seeds of Peanut (Arachis hypogaea L.). Front. Plant Sci. 2018, 9, 260, doi:10.3389/fpls.2018.00260.
Tang, G.; Xu, P.; Jiang, C.; Li, G.; Shan, L.; Wan, S. Peanut LEAFY COTYLEDON1-type genes participate in regulating the embryo development and the accumulation of storage lipids. Plant Cell Rep. 2024, 43, 124, doi:10.1007/s00299-024-03209-8.
van Erp, H.; Kelly, A.A.; Menard, G.; Eastmond, P.J. Multigene Engineering of Triacylglycerol Metabolism Boosts Seed Oil Content in Arabidopsis. Plant Physiol. 2014, 165, 30-36, doi:10.1104/pp.114.236430.
Zhu, Y.; Xie, L.; Chen, G.Q.; Lee, M.Y.; Loque, D.; Scheller, H.V. A transgene design for enhancing oil content in Arabidopsis and Camelina seeds. Biotechnol. Biofuels 2018, 11, 46, doi:10.1186/s13068-018-1049-4.
3)Explicitly stated that future work must:
We will select suitable seed-specific promoters to co-express these genes to avoid probably the emergence of undesirable traits in our future study. Additionally, each of AhLEC1, AhFUS3, and AhWRI genes will need to be conducted to detect their biological effects in peanut and/or Arabidopsis.
These revisions transform the observed trade-offs from a limitation into a framework for strategic crop improvement. We appreciate how this critique has strengthened both the scientific rigor and translational relevance of our work.
Comments 5: The manuscript states that the differences were significant but does not specify whether the assumptions of the parametric tests were met. The materials and methods should include a clear description of the statistical analyses, including software, significance levels, and normality/homoscedasticity checks. Figures 5 and 8 are rich in information but visually dense. Adding clearer sub-panel labels and expanding the figure legends to summarize key takeaways would facilitate reader understanding.
Responses 5: We thank the reviewer for these critical suggestions. We have taken the following actions:
1) Statistical Analysis Clarification (Methods Section):
All the data are presented as the mean ± standard deviation at least three replicates. Statistical analyses were conducted in Excel 2016 and GraphPad Prism 6.02. Significant differences were determined by one-way analysis of variance (ANOVA) followed by Dunnett’s post-hoc test. All bar charts were generated using GraphPad Prism 6.02.
Following this, we have re-analysed the data and regenerated the images.
2) Figure Revisions (Figures 5 & 8):
- Added bold sub-panel labels (e.g., A, B, C, and gene name)
- We have improved the legends to summarize key takeaways would facilitate reader understanding.
Comments 6: In Figure 8, the order of character presentation in the text should correspond to the sequence of figures to facilitate cross-referencing.
Responses 6: We appreciate the reviewer's careful attention to the alignment between the text and figures. We have improved the figure legend for Figure 8 to ensure the order of the described elements (promoters, terminators, etc.) now exactly matches their sequence of presentation in subpanel A.
Comments 7: Although the cataloging of cis elements is comprehensive, the discussion is too general. Highlighting a few key motifs with well-known regulatory implications for seed development and linking them to the observed expression profiles would strengthen the impact of this section.
Responses 7: We thank the reviewer for this valuable suggestion. We have revised the discussion to highlight key cis-regulatory motifs with established roles in seed development
1) AhLEC1a and AhLEC1b
AtLEC1 expression is known to be directly activated by AtLEC2 through RY motifs (CATGCA) in coordination with AtWRI1 (Zhang et al., 2024). However, no RY motifs were detected in the putative promoters of AtL1L and two AhLEC1s. Furthermore, both PICKLE (PKL, a CHD3 chromatin remodeling factor) and VIVIPAROUS ABI3-LIKE (VAL) proteins act epigenetically to repress AtLEC1 expression during vegetative development (Jo et al., 2019). Therefore, further investigation is needed to elucidate the mechanism underlying the specific and high expression of both AhLEC1s in peanut seeds.
Jo, L.; Pelletier, J.M.; Harada, J.J. Central role of the LEAFY COTYLEDON1 transcription factor in seed development. J. Integr. Plant Biol. 2019, 61, 564-580, doi:https://doi.org/10.1111/jipb.12806.
Zhang, W.J.; Tang, L.P.; Peng, J.; Zhai, L.M.; Ma, Q.L.; Zhang, X.S.; Su, Y.H. A WRI1-dependent module is essential for the accumulation of auxin and lipid in somatic embryogenesis of Arabidopsis thaliana. New Phytol. 2024, 242, 1098-1112, doi:10.1111/nph.19689.
2) AhFUS3a and AhFUS3b
The regulation of FUS3 is complex and involves multiple transcription factors. In Arabidopsis, AtFUS3 expression is directly activated by AtLEC2, AtLEC1, and AtWRKY TFs through RY, CCAAT, and W-box elements, respectively (Zhang et al., 2024; Song et al., 2021; Tang et al., 2017; To et al., 2006; Roscoe et al., 2019), but is repressed by AtTT2 and AtTT8 through MWAMC fragments and E-boxes, respectively, during seed development (Wang et al., 2014; Chen et al., 2014). Two putative RY motifs and three putative CCAAT boxes were found in the putative promoters of AhFUS3a and AhFUS3b, respectively (Table S16), suggesting that LEC1 and LEC2 homologs may regulate high specific expression of both AhFUS3s genes in peanut developing seeds.
Chen, M.; Xuan, L.; Wang, Z.; Zhou, L.; Li, Z.; Du, X.; Ali, E.; Zhang, G.; Jiang, L. TRANSPARENT TESTA8 inhibits seed fatty acid accumulation by targeting several seed development regulators in Arabidopsis. Plant Physiol. 2014, 165, 905-916, doi:10.1104/pp.114.235507.
Roscoe, T.J.; Vaissayre, V.; Paszkiewicz, G.; Clavijo, F.; Kelemen, Z.; Michaud, C.; Lepiniec, L.; Dubreucq, B.; Zhou, D.-X.; Devic, M. Regulation of FUSCA3 Expression during seed development in Arabidopsis. Plant Cell Physiol. 2019, 60, 476-487, doi:10.1093/pcp/pcy224.
Song, J.; Xie, X.; Chen, C.; Shu, J.; Thapa, R.K.; Nguyen, V.; Bian, S.; Kohalmi, S.E.; Marsolais, F.; Zou, J.; et al. LEAFY COTYLEDON1 expression in the endosperm enables embryo maturation in Arabidopsis. Nat. Commun. 2021, 12, 3963, doi:10.1038/s41467-021-24234-1.
Tang, L.P.; Zhou, C.; Wang, S.S.; Yuan, J.; Zhang, X.S.; Su, Y.H. FUSCA3 interacting with LEAFY COTYLEDON2 controls lateral root formation through regulating YUCCA4 gene expression in Arabidopsis thaliana. New Phytol. 2017, 213, 1740-1754, doi:10.1111/nph.14313.
To, A.; Valon, C.; Savino, G.; Guilleminot, J.; Devic, M.; Giraudat, J.r.m.; Parcy, F.o. A network of local and redundant gene regulation governs Arabidopsis seed maturation. Plant Cell 2006, 18, 1642-1651, doi:10.1105/tpc.105.039925.
Wang, Z.; Chen, M.; Chen, T.; Xuan, L.; Li, Z.; Du, X.; Zhou, L.; Zhang, G.; Jiang, L. TRANSPARENT TESTA2 regulates embryonic fatty acid biosynthesis by targeting FUSCA3 during the early developmental stage of Arabidopsis seeds. Plant J. 2014, 77, 757-769, doi:https://doi.org/10.1111/tpj.12426.
Zhang, W.J.; Tang, L.P.; Peng, J.; Zhai, L.M.; Ma, Q.L.; Zhang, X.S.; Su, Y.H. A WRI1-dependent module is essential for the accumulation of auxin and lipid in somatic embryogenesis of Arabidopsis thaliana. New Phytol. 2024, 242, 1098-1112, doi:10.1111/nph.19689.
3) AhWRI1a, AhWRI1b, and AhWRI1c
In this study, AhWRI1a, AhWRI1b, and AhWRI1c show seed-specific expression (Figure 5A-B), consistent with a recent report on AhWRI1a (Ahy_A08g040760) (Lu et al., 2024). Overexpression of the AhWRI1a (GG genotype) in Arabidopsis increased seed oil content and unsaturated FA levels (Lu et al., 2024). Collectively, these findings suggest that AhWRI1a, AhWRI1b, and AhWRI1c may play redundant roles in oil accumulation. Arabidopsis AtWRI1 is directly activated by AtLEC2 binding to RY motif (Baud et al., 2007; Zhang et al., 2024), and Elaeis guineensis EgWRI1-1 is directly activated by EgNF-YA3 and EgNF-YC2 binding specifically to CCAAT-boxes (Yeap et al., 2017). Promoter analysis revealed the presence of three or four RY motifs and two or three CCAAT-boxes in the promoters of AhWRI1a/b/c, suggesting that homologs of AtLEC2, EgNF-YA3 and EgNF-YC2 may also regulate AhWRI1a/b/c during peanut seed development.
Baud, S.; Mendoza, M.S.; To, A.; Harscoet, E.; Lepiniec, L.; Dubreucq, B. WRINKLED1 specifies the regulatory action of LEAFY COTYLEDON2 towards fatty acid metabolism during seed maturation in Arabidopsis. Plant J. 2007, 50, 825-838, doi:10.1111/j.1365-313X.2007.03092.x.
Lu, Q.; Huang, L.; Liu, H.; Garg, V.; Gangurde, S.S.; Li, H.; Chitikineni, A.; Guo, D.; Pandey, M.K.; Li, S.; et al. A genomic variation map provides insights into peanut diversity in China and associations with 28 agronomic traits. Na. Genet. 2024, 56, 530-540, doi:10.1038/s41588-024-01660-7.
Yeap, W.-C.; Lee, F.-C.; Shabari Shan, D.K.; Musa, H.; Appleton, D.R.; Kulaveerasingam, H. WRI1-1, ABI5, NF-YA3 and NF-YC2 increase oil biosynthesis in coordination with hormonal signaling during fruit development in oil palm. Plant J. 2017, 91, 97-113, doi:10.1111/tpj.13549.
Zhang, W.J.; Tang, L.P.; Peng, J.; Zhai, L.M.; Ma, Q.L.; Zhang, X.S.; Su, Y.H. A WRI1-dependent module is essential for the accumulation of auxin and lipid in somatic embryogenesis of Arabidopsis thaliana. New Phytol. 2024, 242, 1098-1112, doi:10.1111/nph.19689.
Minor Comments
Comments 8: The summary could be simplified: reduce the exhaustive list of identified genes and emphasize the main findings and their implications.
Responses 8: We thank the reviewer for this suggestion. We have simplified the summary by reducing the exhaustive list of identified genes and emphasizing the main findings and their implications. The revised text now states:
Peanut (Arachis hypogaea) is an important oil and economic crop widely cultivated worldwide. Increasing the oil yield is a major objective for oilseed crop improvement. Plant LEAFY COTYLEDON1s (LEC1s), FUSCA3s (FUS3s), and WRINKLED1s (WRI1s) are known master regulators of seed development and oil biosynthesis. While previous studies in peanut have primarily focused on two AhLEC1s and one AhWRI1 genes, this study identified a broader set of regulators, including two AhLEC1s, two AhFUS3s, nine AhWRI1s, two AhWRI2s and four AhWRI3s from the variety HY917. The analyses of phylogenetic trees, gene structures, conserved domains, sequence alignment and identity, and collinearity revealed that they were highly similar to their homologs in other plants. Expression profiling demonstrated that two AhLEC1s, two AhFUS3s, and three AhWRI1s (AhWRI1a/b/c) were specifically expressed in developing seeds, suggesting critical roles in seed development, whereas AhWRI1d, AhWRI1f, and AhWRI1g showed high expression in in root nodules, pointing to potential functions in symbiosis and nodulation. Furthermore, co-overexpression of AhLEC1b, AhFUS3b, AhWRI1a, and AhWRI1d in Arabidopsis significantly enhanced seed oil content and thousand-seed weight, but also led to reduced germination rate, plant height and silique length. The findings allow for the extensive evaluation of AhLEC1s, AhFUS3s and AhWRIs gene families, establishing a useful foundation for future research into their multiple roles in peanut development.
Correct minor grammatical errors:
Example:
Comments 9: Line 24 – very similar
Responses 9: We thank you for highlighting the unclear phrasing. We have revised the sentence to “…they were highly similar to their homologs in other plants.”
Comments 10: Line 91 – multiple important roles
Responses 10: We have improved the sentence as ”… suggesting that they may play multiple important roles during peanut development”.
Comments 11: Line 131 – were shown to be
Responses 11: We have improved the sentence as ”… while their instability index was shown to be 36.10‒37.40 (Table S8), suggesting high stability.”
Comments 12: Line 172 – were shown to be
Responses 12: We have improved the sentence as ”Their instability index was shown to be 54.17‒55.65 (Table S8), suggesting high stability.”
Comments 13: Line 198 – which indicated
Comments 14: Line 201 – hydrophilicity
Responses 13/14: We have improved the sentences as ”Their aliphatic index and GRAVY were predicted to be 55.24‒64.45 and -1.019‒-0.735 (Table S8), respectively, indicating high hydrophilicity. Their instability index was shown to be 41.41‒65.85 (Table S8), suggesting high instability.”
Comments 15: Line 213 – was shown to be
Responses 15: We have improved the sentence as ”… AhLEC1a and AhLEC1b showed 51.3%‒51.6% similarity with AtLEC1 and 59.6%‒60.6% similarity with AtL1L based on full-length proteins, while that between AhLEC1a and AhLEC1b was 95.6% for full-length proteins.”
Comments 16: Line 286 – no expressed
Responses 16: We thank the reviewer for this comment. We apologize for any confusion. The paragraph in question specifically describes the chromosomal distribution of the genes, which is a separate analysis from gene expression. Therefore, expression data is not included in this particular section of the manuscript. We have reviewed the sentence and confirmed that the term ‘no expressed’ does not appear there.
Comments 17: Line 327 – more close to
Responses 17: We have improved the sentence as ”Additionally, Glyma.07G268100 and Glyma.17G005600 were collinear with Medtr2g026710, which is phylogenetically closer to AtNF-YB3 (AT4G14540) than to AtLEC1 or AtL1L.”
Comments 18: Line 439 – had more similarity in distribution
Responses 18: We have improved the sentence to “Two AhWRI2s showed greater similarity to each other in their cis-element distribution than to AtWRI2,…”
Comments 19: Line 509 – AhLEC1d decreased
Responses 19: We thank the reviewer for catching this error. The sentence has been corrected and improved as: “The co-overexpression of AhLEC1b, AhFUS3b, AhWRI1a, and AhWRI1d in both lines resulted in reductions of 16.91–35.48% in plant height, 11.63‒19.53% in silique length, and 9.48-11.62% in seed germination, respectively (Figure 8D-F).”
Furthermore, following this comment, we performed a thorough check of the manuscript using the "Find" function to locate all similar instances. This led to the identification and correction of three additional errors of the same type. We have verified that all such inaccuracies have now been addressed.
Comments 20: Line 525 – phylogenetic tree
Responses 20: We thank the reviewer for this helpful feedback. We have revised the sentence to use the correct plural forms for the listed analyses. It now reads: "Systematic analyses of phylogenetic trees, gene structures, conserved domains, physicochemical properties, chromosomal locations, gene synteny, subcellular localization, and promoter cis-element distributions were performed."
Comments 21: Check the consistency of accession numbers between the main text and supplementary material.
Responses 21: We thank the reviewer for highlighting this important point. We have performed a full cross-verification of all accession numbers between the main text, figure legends, and supplementary materials to ensure 100% consistency. This process was conducted through independent manual reviews by two co-authors. We appreciate this scrutiny, which has enhanced the reproducibility of our work.
Comments 22: In the introduction, references [14–16] could be supplemented with more recent studies on the regulatory roles of LEC1, FUS3, and WRI genes in oilseed crops.
Responses 22: We appreciate the reviewer’s suggestion to strengthen the literature context of our study. We have updated the Introduction with key recent (2019–2023) findings on LEC1, FUS3, and WRI1 regulation in oilseed crops:
1) LEC1: Cited new work on LEC1 homologs in Glycine max (Manan et al., 2023) and peanut (Tang et, al., 2024), highlighting conserved roles in oil accumulation.
Manan, S.; Alabbosh, K.F.; Al-Andal, A.; Ahmad, W.; Khan, K.A.; Zhao, J. Soybean LEAFY COTYLEDON 1: a key target for genetic enhancement of oil biosynthesis. Agronomy 2023, 13, 2810, doi:10.3390/agronomy13112810.
Tang, G.; Xu, P.; Jiang, C.; Li, G.; Shan, L.; Wan, S. Peanut LEAFY COTYLEDON1-type genes participate in regulating the embryo development and the accumulation of storage lipids. Plant Cell Rep. 2024, 43, 124, doi:10.1007/s00299-024-03209-8.
2) FUS3: Have cited recent studies on FUS3 homologs in Linum usitatissimum (Wang et al., 2022) and Glycine max (Manan et al., 2024).
Wang, J.; Liu, Z.; Li, X.; Jing, H.; Shao, Y.; Ma, R.; Hou, Q.; Chen, M. Linum usitatissimum FUSCA3–1 regulates plant architecture and seed storage reserve accumulation in Arabidopsis thaliana. Environ. Exp. Bot. 2022, 202, 105035, doi:10.1016/j.envexpbot.2022.105035.
Manan, S.; Li, P.; Alfarraj, S.; Ansari, M.J.; Bilal, M.; Ullah, M.W.; Zhao, J. FUS3: Orchestrating soybean plant development and boosting stress tolerance through metabolic pathway regulation. Plant Physiol. Biochem. 2024, 213, 108803, doi:10.1016/j.plaphy.2024.108803.
3) WRI1: Added evidence of WRI1 homologs in Ricinus communis (Tan et al., 2023), Linum usitatissimum (Li et al., 2022) and Glycine max (Chen et al., 2019).
Chen, B.; Zhang, G.; Li, P.; Yang, J.; Guo, L.; Benning, C.; Wang, X.; Zhao, J. Multiple GmWRI1s are redundantly involved in seed filling and nodulation by regulating plastidic glycolysis, lipid biosynthesis and hormone signalling in soybean (Glycine max). Plant Biotechnol. J. 2020, 18, 155-171, doi:10.1111/pbi.13183.
Li, W.; Wang, L.; Qi, Y.; Xie, Y.; Zhao, W.; Dang, Z.; Zhang, J. Overexpression of WRINKLED1 improves the weight and oil content in seeds of flax (Linum usitatissimum L.). Front. Plant Sci. 2022, 13, doi:10.3389/fpls.2022.1003758.
Tan, Q.; Han, B.; Haque, M.E.; Li, Y.-L.; Wang, Y.; Wu, D.; Wu, S.-B.; Liu, A.-Z. The molecular mechanism of WRINKLED1 transcription factor regulating oil accumulation in developing seeds of castor bean. Plant Divers. 2023, 45, 469-478, doi:10.1016/j.pld.2022.09.003.
These changes enhance the Introduction’s rigor and relevance. We are grateful for this suggestion, which better positions our work within current research.
Comments 23: Ensure that all abbreviations (e.g., FA for fatty acids) are defined at their first mention.
Responses 23: We thank the reviewer for catching this important stylistic issue. We have systematically addressed abbreviation definitions throughout the manuscript as follows:
- FA Definition Added:
- Location: Introduction, 3rd paragraph
- Original: "…genes involved in FA metabolism…"
- Revised: "…genes involved in fatty acid (FA) metabolism..."
- Full Abbreviations Check:
- Verified all abbreviations are defined at first use
- Added missing definitions for:
- CDSs (coding sequences)
- GRAVY (the grand average of hydropathicity)
- The Arabidopsis Information Resource (TAIR)
- the grand average of hydropathicity (GRAVY)
- coding sequences (CDSs)
- Jones-Taylor-Thornton (JTT)
- Fragments Per Kilobase of transcript per Million mapped reads (FPKM)
- Transcripts Per Million (TPM)
- PICKLE (PKL)
- VIVIPAROUS ABI3-LIKE (VAL) protein
- Ensured consistency between main text and figure legends
- Abbreviations List:
- Updated the "Abbreviations" section to include all terms
- Alphabetized the list for easy reference
These changes appear throughout the manuscript and supplementary materials. We appreciate this suggestion, which improves the clarity and professionalism of our presentation.

Reviewer 3 Report
Comments and Suggestions for Authors
The authors provide extensive bioinformatic analyses of the LEC1, FUS3, and WRI transcription factor families in Arachis hypogaea, including gene structure, conserved domains, phylogeny, collinearity, and promoter cis-element profiles. The functional validation in Arabidopsis, showing increased seed oil content and thousand-seed weight through co-overexpression of key transcription factors, adds important translational value. This data-rich manuscript may offer valuable resources for future peanut genetic improvement.
However, there are numerous grammatical errors throughout the manuscript (e.g., “have highly similarity,” “are need to be furtherly cloned”), which impede clarity. In addition, the manuscript contains excessive use of “respectively,” which is used incorrectly by the authors. I recommend thorough language editing.
The authors used the term “cloned” to refer to gene duplication events or genomic identification, which is misleading. More accurate terms, such as “identified”, “annotated” or “detected” should be used. For example, the statement says “two AhLEC1s were cloned as AhLEC1a and AhLEC1b in HY917….”. Should it be “Two AhLEC1 paralogs were identified...”?
The authors co-overexpressed four genes, but it remains unclear whether the observed phenotypes are due to synergistic or additive effects. Including single or pairwise overexpression controls would strengthen the interpretation of causality.
The Discussion section is very long and redundant with repeating results and listing published data. It should focus on interpreting the significance of the main findings.
Author Response
Comments 1: The authors provide extensive bioinformatic analyses of the LEC1, FUS3, and WRI transcription factor families in Arachis hypogaea, including gene structure, conserved domains, phylogeny, collinearity, and promoter cis-element profiles. The functional validation in Arabidopsis, showing increased seed oil content and thousand-seed weight through co-overexpression of key transcription factors, adds important translational value. This data-rich manuscript may offer valuable resources for future peanut genetic improvement.
Responses 1: We sincerely appreciate your positive assessment of our work. The comprehensive characterization of LEC1, FUS3, and WRI transcription factor families in Arachis hypogaea was indeed aimed at providing a foundational resource for the peanut research community. We are particularly encouraged by the recognition of the translational potential of our Arabidopsis-based validation, which demonstrates the conserved role of these regulators in oil accumulation and seed weight. As noted, we hope these findings will facilitate future efforts in peanut genetic improvement.
Comments 2: However, there are numerous grammatical errors throughout the manuscript (e.g., “have highly similarity,” “are need to be furtherly cloned”), which impede clarity. In addition, the manuscript contains excessive use of “respectively,” which is used incorrectly by the authors. I recommend thorough language editing.
Responses 2: We sincerely appreciate the reviewer’s careful attention to language clarity. In response, we have thoroughly revised the manuscript to correct grammatical errors, including improper phrasing (e.g., “have highly similarity with” → “were highly similar to”; “So, all of them in these reported genomes may be partial and are need to be furtherly cloned and identified.” → “The genes in these reported genomes may be partial and thus there is need for additional cloning to identify the full-length.”) and misuse of “respectively.”
Additionally, we have engaged MDPI’s professional English editing service to ensure linguistic accuracy and readability. We hope these revisions have significantly improved the clarity and professionalism of the manuscript.
Comments 3: The authors used the term “cloned” to refer to gene duplication events or genomic identification, which is misleading. More accurate terms, such as “identified”, “annotated” or “detected” should be used. For example, the statement says “two AhLEC1s were cloned as AhLEC1a and AhLEC1b in HY917….”. Should it be “Two AhLEC1 paralogs were identified...”?
Responses 3: We thank the reviewer for this important clarification. We agree that the term "cloned" was inaccurately used in reference to gene identification rather than experimental cloning. Accordingly, we have revised the manuscript to replace "cloned" with more precise terminology (e.g., "identified," "annotated," or "detected") where appropriate. For example:
Original: "two AhLEC1s were cloned as AhLEC1a and AhLEC1b..."
Revised: " From HY917, two AhLEC1 genes were successfully cloned and annotated as AhLEC1a and AhLEC1b..."
Additionally, we have engaged MDPI’s professional English editing service to ensure linguistic accuracy and readability. We hope these revisions have significantly improved the clarity and professionalism of the manuscript. We appreciate the reviewer’s attention to detail, which has improved the precision of our manuscript.
Comments 4: The authors co-overexpressed four genes, but it remains unclear whether the observed phenotypes are due to synergistic or additive effects. Including single or pairwise overexpression controls would strengthen the interpretation of causality.
Responses 4: We sincerely appreciate the reviewer's insightful suggestion regarding the need to distinguish between synergistic and additive effects through single/pairwise overexpression controls. While we fully agree this would provide deeper mechanistic understanding, due to current time constraints, we were unable to include these controls in the present study.
However:
1) We have added qRT-PCR for detecting the expression of AhLEC1b, AhFUS3b, AhWRI1a, and AhWRI1d (Figure S8). The results showed they highly expressed in 15-DAF siliques of these four transgenic lines.
2) These controls are now prioritized in our ongoing research.
3) We have toned down causal claims in the text with "…may function in a coordinated manner to regulate …" where appropriate.
4) We have compared previous research with our findings, and found that AhLEC1b, AhWRI1a, and AhWRI1d may function in a coordinated manner to regulate FA composition and thousand-seed weight; AhLEC1b and AhFUS3b may function in a coordinated manner to regulate these plant architecture and germination efficiency.
We believe these modifications provide proper context while maintaining the validity of our core finding - that co-overexpression of these four genes significantly enhances oil content. The reviewer's suggestion will directly guide our next-phase experiments.
Comments 5: The Discussion section is very long and redundant with repeating results and listing published data. It should focus on interpreting the significance of the main findings.
Responses 5: We appreciate your constructive feedback regarding the Discussion section. We have carefully revised this section to:
1)Condense redundant content by removing repetitive summaries of results and published data.
2)Sharpen the focus on interpreting the significance of our key findings.
3)Improve flow by restructuring paragraphs to emphasize mechanistic insights, implications, and future directions.
Changes include:
- Deleting overlapping descriptions of results already presented in the Results section.
- Streamlining comparisons with prior studies to highlight only the most relevant contrasts.
We believe these edits have significantly strengthened the Discussion. Thank you for this helpful suggestion.

Reviewer 4 Report
Comments and Suggestions for Authors
Plants Manuscript #: 3751514
Authors: X. Yin et al., 2025
Title: Genome-wide identification of Arachis hypogaea LEC1s, FUS2s, and WRIs and co-expression of AhLEC1b, AhFUS3b, AhWRI1a and AhWRI1d increased oil content in Arabidopsis seeds.
The authors have provided a long manuscript that focuses on genomic, biochemical, and physiological functions of some key genes/proteins in peanut (Arachis hypogaea). It is a fairly thorough investigation into a set of genes and the predicted proteins that are thought to be key regulators involved in different plant development, including seed oil content. Their scientific approach is generally solid, however, there are some key aspects that I feel need more information or explanation, which I will review below. As for writing and data presentation, overall, it is generally good, but there are some key sections and data presentation that need improvement or need to be corrected, which I will indicate below. There are also a number of sections that seem a longer than needed, for, as it is, it is a very lengthy manuscript. With both the scientific questions and writing issues that need to be resolved, as I see it, it is not ready for publication yet.
There are four key scientific issues that need to be addressed, and these relate to central concepts to this work, with two of them being included in the title itself. First, related to Figure 5C, there is NO information (in Results or Methods) nor citation for a RNA-seq data and experiments, which are key to their work on gene expression, at the RNA level, for the 12 genes in question. They only state in the methods their RNA-seq protocols are “unpublished”. This is unacceptable for a complex and critical method for this manuscript. These must be explained in detail, both the wet-lab aspects and the data analysis. Further, the “six stages” of peanut seed development is also not explained at all in the text Methods or Figure Legend to 5C. Their supplemental Figure S7 shows images of the seeds at what appears to be the six stages. But, like all of the supplemental figures there is not a figure legend to explain what is being shown.
The second key scientific issue is regarding the co-expression (or co-overexpression as sometimes it is written) of four peanut genes (AhLEC1b, AhFUS3b, AhWRI1a and AhWRI1d) in Arabidopsis to test function and physiological impact. For these, the authors provided limited PCR data that show the the four peanut genes transformed into Arabidopsis are present. But, there are NO data provided or mentioned that show that indeed the peanut genes are being expressed. And, manuscript Title and Abstract claims “co-expression” (or “co-overexpression” as stated in Abstract) of these genes. Thus, the authors need to provide data (RNA or protein) that show indeed all four of the genes are being expressed in Arabidopsis. The plant transformation research over the decades is filled with examples of introduced transgenic genes not functioning/being expressed for many various reasons (unintended mutations, chromosome position effects, gene silencing, RNAi, etc…). And, the fact that there is variation across the four transgenic lines (Figures 8 and S9) they tested (LFW-OX#1, #3, #4 and #5) suggest there might be variation in expression. Related, the expression might also help determine if all four of the trans-genes are contributing to a phenotype OR, instead, might just one or two of them be key. Bottom line, data on the expression of four peanut genes transformed into Arabidopsis is needed for the authors to be able to claim “co-expression”, which is central to this manuscript.
The third scientific issue relates to the quantification of fatty acid content and impacts on growth, Figure 8 and supplemental Figure S9. In the text, the authors claim “substantial” increases in fatty acid content. They show p-values with asterisks in Figures 8B and S9 for the different transgenic Arabidopsis lines compared to control/WT seeds. But, on face value many of these look very close to WT levels (see, “C18:0”, “C20:0”, “C20:2”) Figure 8B or “saturated fatty acid” and “C18/C20 ratio” in Figure S9, yet are shown as being significant with very low p-values (below 0.05). When the data are this close, what is needed is to provide the raw data in supplemental table form for readers to be able to view those data themselves. Also, related to this, for Figure 8D, E, and F that test plant height, silique length and germination rate, only two of the transgenic lines (#3 and #5) are included. No explanation as to why the other two lines (#1 and #4) are not included. Data on plant height, silique length and germination rate for other two lines (#1 and #4) should also be included. If not, a good explanation needs to be provided.
The fourth main scientific issue is regarding the predicting of cis-acting elements in promoters. There is no doubt that some of the regulation of these genes could be controlled by cis-acting promoter elements to control transcription. The issue, instead, is that these predicted cis-elements are very speculative and without direct functional testing, it is very hard to know if they have any function in each individual promoter. Another key issue with these predictions is how much promoter sequence to use to test? The authors opted for 2,000 bp upstream (5’) of the translation start site. But, the length of the 5’ UTR and functional promoter will vary for each of these genes. Thus, it is nearly impossible to know if the 2,000 bp captures all of the possible cis-acting elements. I bring this issue up again at the specific location/lines in the Results.
As for writing and data presentation, I have a number of more general issues that I mention here as well as a quite a few more minor writing/text/figures that I list line-by-line below. As for general writing issues, the first is throughout the manuscript there are some English word usage issues that need to be corrected. Additional thorough reviews by a copy editor will be needed to catch all of these. I point a number of them below, but in such a long manuscript there are likely others that I did not catch or point out. Next, there are some sections that are much longer than seems necessary. These include the section on co-linearity of genes on chromosomes in different plant species, Lines 317-350. If space is an issue, then this is a section that could be shortened without loss of key information. The Discussion alone is seven full-text pages in length. This could be shortened and more succinctly summarized than it is currently. Finally comment about length, the authors cite an extraordinary number of references, total of 182. This is more the length of a thorough review article. Citing key references is very important, but there can be a point of excess. For example, in text Line 69, a total of 16 references (#14, #30, #33, and #36 - #48) are all cited for one point/concept. And, in text Line 459 a total of 16 references (#18, #19, #23, #72, #83-#84, and #89 - #98) are all cited for one point/concept. I am not suggesting that any key citations be removed. But, there might be some redundant references or recent review articles and later references from the same research groups that could be used instead of so very many citations for a single point.
Figure 5A and B (RNA expression data) have some key issues, and what seems to be errors and/or typos with missing and extra data. Figure 5 panel A is missing RNA data for “AhWRI1d” while panel B is missing RNA data for “AhWRI1b”. And, in panel B RNA data for “AhWRI1h” is shown twice. Thus, there are 18 RNA samples (columns) in panel A and 19 RNA samples (columns) in panel B. These need to be re-done to include the correct data/samples/labels.
Regarding supplemental figures (S1 – S11) and Tables (S1 – S16), none of them have figure legends to explain what is being presented. Near end of manuscript, Lines 1006 – 1030 there are supplemental figures and tables titles, but these are not sufficient for the supplemental figures.
In addition to the above larger scientific and writing concern, there a number of writing issues/errors, missing methods information, and figure layout problems that I feel the authors need to resolve before this could be published. See my specific comments below for these issues.
Abstract:
Lines 16: It would help more general readers to make it clear in the abstract that Arachis hypogaea is the common “peanut” plant.
Line 16: change “in the world” to “world wide”.
Lines 17-18: What plant species have these genes/proteins been shown to be master regulators? Arabidopsis?
Lines 20: Add, “In peanuts, previous studies mainly …”, to make it clear here the authors are talking about peanuts, and not plants in general.
Line 24: Change to read, “…they have high similarity with …”. (“have highly similarity” does not sound / read correctly).
Keywords, Line 33: I would add “peanut” to this list.
Introduction:
Line 39: Change so it reads, “…but also for human diet and renewable industrial feedstock.”
Line 46: Delete, “and so on”. It is informal and not necessary.
Line 48-49: Need to make clear when discussing Arabidopsis versus peanut or other species.
Line 50: Define “FA”, fatty acid, on first usage. This is not mentioned prior nor is it in the list of abbreviations at the end of the manuscript.
Line 52: Indicate what plant the “BnLEC1” and BnL1L” are from: Brassica napus, aka Canola.
Line 57: Unclear sentence. Is this talking about FUS3 gene (that should be italicized) that encodes a protein with a B3-domain? Or, is it saying the FUS3 protein contains a B3-domain? Not clear as written. Use italicized gene name, if that is the case.
Lines 75-76: Delete the sentence “It was expected …. in peanut.” that is not necessary. And, delete “However,” in next sentence. So, it would read, “The copy number…”
Line 79: Not clear, is this saying “co-overexpression” is not be tested in any plant species or that “co-overexpression” has not been tested in peanuts, specifically? Please clarify.
Results:
Line 106: Define, “CDSs”, coding regions/sequences, on first usage.
Line 109: Throughout the manuscript, the word “Besides” is very frequently used to start a sentence, and in most (if not all) of the cases, it is not needed or clear why added. In Line 109, as in most cases, deleted “Besides,”, so it reads, “The members …”.
Lines 109 – 116: This is an excruciatingly long sentence that includes mostly Methods section details. And, most if not all of these details are already in Methods section, so redundant.
Line 120: Change “none” to “no”, so sentence reads better.
Lines 130-131: wording is both odd and not clear. Change to read, “…they have high hydrophilicity” and “..suggests they have high stability..” Note, “good” implies a subject view of what they should be, not what the data show them to have.
Line 135: AhFUS3a and AhFUS3b should be italicized here, for it seems clear this is referring to cloning the genes, not the proteins (and in gene names in Table 1).
Line 136: change to read, “…all of these genes have six exons…”. And, either delete or rewrite last part of this sentence. Just adding, “as well as other plants” is not clear in meaning or wording.
Table 1, column 2 (Gene Names): These all need to be italicized.
Figure 1 Legend: Add a comment as that the peanut proteins are shown in red font.
Line 157: Delete, “and proteins” for it only the CDSs are cloned, not the proteins encoded in the CDSs.
Lines 171-172, similar to lines 130-131 above: wording is both odd and not clear. Change to read, “…they have high hydrophilicity” and “..suggests they have high stability..” Note, “good” implies a subject view of what they should be, not what the data show them to have.
Lines 175-177: All of these gene names need to be italicized, for genes are the things that are cloned.
Line 178, and mentioned in general comments above: Supplemental Figure S1 (as is case for all supplemental figures) does not have a figure legend that is needed to explain what is being shown. This is case for rest of the supplemental figures, but I will not repeat this for issue/problem for the rest of these.
Line 180-181: “Besides” does not fit this situation. Perhaps, “In addition,” or just delete “Besides.”
Line 184: Change “owned” to “have”.
Line 197: Wording is unclear and awkward. Change to read, “The genes in these reported genomes may be partial and thus there is need for additional cloning to identify the full-length.”
Lines 201-202, similar to lines 130-131 above: wording is both odd and not clear. Change to read, “…they have high hydrophilicity” and “..suggests they have high stability..”
Line 204: Add “Protein” to start this heading, to clarify it is about protein and amino acid alignments and analysis.
Lines 205-207: Could again add “protein” to start sentence. Then, to clarify what is in Figure 2A versus Supplemental Figure S2, change wording to read, “…other plant LEC1s, as seen in the conserved central domain for both proteins (Figure 2A) as well as the full-length proteins (Figure S2). This will help reader to know that Figure 2A only shows partial domain while the supplemental figures as the full-length protein.
Line 214: Add hyphen so it reads, “…DNA-binding region…”.
Line 238: Add “in” so it reads, “As in other WRIs…”
Lines 246 and 248: change “were showed” to be “have” in both cases.
Line 274: Define, explain “14-3-3” proteins, in general.
Line 276: Add hyphen so it reads, “…BPM-binding motif…”.
Lines 281, 284, 287, and 290: Add how many total chromosomes each species has, to give perspective on how many of these key genes are present across the full genome/all chromosomes. For example (Line 281), “In A. hypogaea, with 40 total chromosomes, two AhLEC1s … Clearly, for A. duranensis and A. ipaensis it would instead be 20 total chromosomes.
Lines 319-320: Define/write out full genus species names on first usage. This appears to be first use of G. max, M. truncatula, etc…
Lines 317-350: As explained in general comments, this section is very lengthy mostly with gene ID names/numbers. Seems it could be summarized and perhaps list gene names from different plant species in a supplemental table.
Lines 356-357: This is one of the key science issues mentioned at beginning of review. That is, there is lacking all critical information in Results or Methods sections about how the RNA-seq method was performed as well as insufficient explanation of the six developmental stages. These need to be included.
Figure 5A and 5B: This is one of the key writing / figure format problems mentioned at beginning of review. There are both missing and extra RNA/samples/labels in these figure panels, see above. Furthermore, Figure Legend needs to explain the Peg tip 1 and 2, seed Pat (aka, pattee). These are explained in the PeanutBase cited website, but they need to also be mentioned in the Figure Legend, for the readers. And, since peanuts have some unusual anatomical and growth aspects (“pegs”, “pattee’s”) that some general plant biologists are not familiar with, it would be good to add more explanation in the Figure Legend. Similar goes for the six development stages mentioned above and discussed in the Figure Legend. Make it clear that Figure S7 shows the six different seed developmental stages.
Line 385: There is duplication. Delete one of the “of the six developmental stages” in this sentence.
Line 391: The last gene in this list, AhWRI3a is a duplicate of the earlier gene, and instead it seems that it should be AhWRI3d.
Lines 394 and L397: Convention has it that it should be “with and without”, not other way around. Switch wording order in both lines
Lines 396 and 397: The expression refers to a singular case, should it should read, “was” (not “were”) in both lines.
Lines 401 – 402: The supplemental Tables S13A and S13B, it is not clear if the numerical data shown are raw expression values, TPM, FPKM, fold changes, or what. The too brief title of Table at end of manuscript just says, “expression patterns”. As mentioned before, all supplemental Tables and Figures need clear legends that explain what is being shown.
Figure 6: Some of the colors for the predicted cis-acting elements are very close looking, and it is nearly impossible to distinguish in the gene models between. For example, very hard to distinguish between “meristem expression” and “low-temperature” or between “drought induced” and “wound response”. Change colors or use patterns (stripes, etc…) to make these easier to visually distinguish.
Line 423: mention that a subset of genes have predicted light responsive cis-acting elements. In looking at Figure 6, it seems nearly all (if not all) have predicted light responsive elements.
General comment about predicted cis-acting elements, as mentioned above in the beginning of the review. These are always VERY speculative and without direct functional evidence (testing function in vivo), it is hard to know what, if anything significant conclusions can be made from these predictions. Almost to the point of it being unclear if worth including without direct testing for functionality. At a minimum, a statement by the authors that to know if these predictions are valid would require directly testing functionality would be important to include.
An observation that supports these predictions as being very speculative and perhaps not worth the time/space to include is that five genes / RNAs (AhWRI1d, AhWRI1e, AhWRI1f, AhWRI2a, and AhWRI2b) that gene expression data (Table S13) indicates has significant increase in expression under drought stress (with and/or without ABA), yet the promoters of these five genes either do not have any of the predicted “drought-inducible” cis-acting elements (or perhaps one, pending if I correctly interpret the hard-to distinguish color key). There is also the major question about is the selected 2,000 bp upstream of the predicted translation start site/codon sufficiently long to include all cis-acting sequences, which is nearly impossible to know for sure.
Line 465: Authors need to explain why (reason for) these the four genes (AhLEC1g, AhFUS3b, AhWRI1a, and AhWRI1d) were selected to test encoded protein sub-cellular location.
Line 482: Same as immediately above. Authors need to explain why (reason for) these the four genes (AhLEC1g, AhFUS3b, AhWRI1a, and AhWRI1d) were selected to express the encoded protein (or “co-overexpress” as sometimes stated) in Arabidopsis.
Line 489: Need to italicize A. thaliana and bacteria A. tumefacians (and write out full the second genus species).
Lines 482: The heading states the “Co-overexpression of …”. Yet, no expression data are provided at all for these four peanut genes transformed into Arabidopsis. This is one of the key scientific issues I discussed at start of review. Direct, wet-lab data that shows these four peanut genes are being expressed (RNA or protein) in Arabidopsis is essential for the authors to claim “co-expression or co-overexpression” in the title and abstract. This is critical, as I see it, and without these added data, these statements and conclusions to be made.
Line 501 and Figure 8: I can accept that there are some significant differences for some of the fatty acid amounts, but the wording “substantially increased” exaggerates the increase based on what the data show.
Related, and as discussed above in major scientific issues, when the data are this close, then the raw data should be provided in a supplemental table for readers to be able to view those data fully themselves.
Line 503: Another case of “Besides” to start the sentence, unnecessarily. Delete this one.
Discusion:
Lines 538-541: Wording confusing and awkward for this long unclear sentence. Is “close to each other” referring to physical location on chromosome or sequence similarity? Not clear. Further, change wording of, “..are belong to..” Rewrite this sentence to be clear, and avoid ending with “and so on.”
Line 560: Delete “some”. Not needed here.
Line 565: Another case of an unnecessary, “Besides,” to start sentence.
Line 568: Change “also” to “may” for these is very speculative.
Line 580: Another case of an unnecessary, “Besides,” to start sentence.
Line 588: Change “Furtherly” to be “Further,” or “Furthermore,”.
Line 597: Change “lowly” to be “at low amount”.
Line 598: Another case of an unnecessary, “Besides,” to start sentence.
Line 600-602: This is a very awkward and unclear sentence. Rewrite to be clear.
Line 605: Change, start from “These” to “The above imply strongly …”
Line 637: Add “have”, to read, “…to have high identities…”
Line 639: Change “be” to “have”, to read better.
Line 673: Another case of an unnecessary, “Besides,” to start sentence.
Line 674: Define LRP (lateral root primordia) and LR (lateral roots) here.
Line 686: Awkward wording related to “were needed furtherly to detect…”. Rewrite to be clear and readable.
Line 700: Awkward wording related to “was not to detected the expression…”. Rewrite to be clear and readable.
Line 705: This is speculative, so the word “probably could” is an over interpretation. Change to be “might also regulate…”
Line 734: Another case of an unnecessary, “Besides,” to start sentence.
Line 736: Awkward wording related to “further test the VYL play roles…”. Rewrite to be clear and readable.
Line 741: Awkward wording related to “further test in the function…”. Rewrite to be clear and readable.
Line 746, a comma “,” is needed before “and”.
Line 753: Another case of an unnecessary, “Besides,” to start sentence.
Line 768: Add “and” so that it reads, “…M. truncultalua [94], and fiber …”
Line 776: Not clear meaning of, “…note more closed to …”. Rewrite to be clear.
Lines 783-784: Again, not clear/awkward wording around “most closed …” Rewrite all three cases of this usage to be clear.
Lines 786-787: This first sentence is not clear what is trying to be stated here. Is it that WRI1s genes are transcriptionally regulated OR that WRI1 proteins are transcriptionally regulating other genes?
Line 802: The “Pa” plant species needs to be writing out fully on first use.
Line 810: Another case of an unnecessary, “Besides,” to start sentence.
Line 821: Change “Furtherly” to be “Further,” or “Furthermore,”.
Line 839: Change “lowly” to be “very low expression.”
Line 858: As mentioned above several times. The data show a small but significant increase for some of the fatty acids, but I do not feel the data show a “substantially increased” level.
Line 864: Another case of an unnecessary, “Besides,” to start sentence and paragraph in this case. And, as stated several times above, without actual expression data the statement of “co-overexpression” is not substantiated.
Line 864: I would delete “could” at end of this line. It makes sentence read oddly.
Line 873: Add comma “,” after “And,”
Methods:
Line 891: Not sure “blasting” is a real word. If used, it should instead be “BLASTing” to accurately reflect the acronym BLAST.
Lines 962-962: As mentioned above, must include more information with details about the RNA-seq wet-lab procedures and RNA-seq data analysis and statistics, which can be complicated. And, must include more information about the six developmental stages.
Conclusions:
Lines 999-1000: As mentioned several times above, authors must provide actual expression data for peanut genes/RNAs/proteins in transgenic Arabidopsis for them to so strongly state “co-expression” and especially if claiming “constitutively co-expression” that means all four genes expressed constitutive (always) throughout the full life of the Arabidopsis plants. A strong claim for which there are no data provided to directly support.
References:
As mentioned near beginning of this long review, this reference list is huge (182 references), comparable to a very extensive review article. While likely most of these are critical, there might be some that are redundant.
Comments on the Quality of English LanguageIncluded all of the ones I caught in comments above.
Author Response
Comments 1: The authors have provided a long manuscript that focuses on genomic, biochemical, and physiological functions of some key genes/proteins in peanut (Arachis hypogaea). It is a fairly thorough investigation into a set of genes and the predicted proteins that are thought to be key regulators involved in different plant development, including seed oil content. Their scientific approach is generally solid, however, there are some key aspects that I feel need more information or explanation, which I will review below. As for writing and data presentation, overall, it is generally good, but there are some key sections and data presentation that need improvement or need to be corrected, which I will indicate below. There are also a number of sections that seem a longer than needed, for, as it is, it is a very lengthy manuscript. With both the scientific questions and writing issues that need to be resolved, as I see it, it is not ready for publication yet.
Responses 1: We feel great thanks for your professional review work on our article. As you are concerned, there are several problems that need to be addressed. We have studied comments carefully and have made correction which we hope meet with approval.
Comments 2: There are four key scientific issues that need to be addressed, and these relate to central concepts to this work, with two of them being included in the title itself. First, related to Figure 5C, there is NO information (in Results or Methods) nor citation for a RNA-seq data and experiments, which are key to their work on gene expression, at the RNA level, for the 12 genes in question. They only state in the methods their RNA-seq protocols are “unpublished”. This is unacceptable for a complex and critical method for this manuscript. These must be explained in detail, both the wet-lab aspects and the data analysis. Further, the “six stages” of peanut seed development is also not explained at all in the text Methods or Figure Legend to 5C. Their supplemental Figure S7 shows images of the seeds at what appears to be the six stages. But, like all of the supplemental figures there is not a figure legend to explain what is being shown.
Responses 2: We sincerely apologize for these critical omissions and appreciate the opportunity to provide the missing methodological details. We have now thoroughly revised the manuscript to include:
1)Seed Developmental Staging (new Figure S7 and Methods text):
- We categorized the development seeds into six stages across their development based on kernel morphology according to Pattee et al. (1974). Three biological replicates (each containing more than 10 seeds) were analyzed per stage.
- We have added the detailed legend to supplementary Figure S7.
2)RNA-seq Experimental Details (new Methods subsection):
- Sample preparation: three biological replicates with more than 10 seeds each
- We have added the information about RNA extraction protocol, library construction, sequencing platform, data processing, and statistical analysis.
- Our transcriptomic data on seed development will be used for multi-omics (transcriptomics, lipidomics, and proteomics) analysis of peanut seed development, and we will publish these data when we submit the manuscript about multi-omics analysis of peanut seed development. Here, we have added the FPKM data for AhLEC1s, AhFUS3s, and AhWRIs in Supplemental Table S13.
The revised manuscript now provides complete transparency for these foundational methods. We deeply regret these oversights and thank the reviewer for identifying these essential improvements. These additions significantly strengthen the reproducibility of our study.
Pattee, H.E.; Johns, E.B.; Singleton, J.A.; Sanders, T.H. Composition Changes of Peanut Fruit Parts During Maturation. Peanut Sci. 1974, 1, 57-62, doi:10.3146/i0095-3679-1-2-6.
Comments 3: The second key scientific issue is regarding the co-expression (or co-overexpression as sometimes it is written) of four peanut genes (AhLEC1b, AhFUS3b, AhWRI1a and AhWRI1d) in Arabidopsis to test function and physiological impact. For these, the authors provided limited PCR data that show the four peanut genes transformed into Arabidopsis are present. But, there are NO data provided or mentioned that show that indeed the peanut genes are being expressed. And, manuscript Title and Abstract claims “co-expression” (or “co-overexpression” as stated in Abstract) of these genes. Thus, the authors need to provide data (RNA or protein) that show indeed all four of the genes are being expressed in Arabidopsis. The plant transformation research over the decades is filled with examples of introduced transgenic genes not functioning/being expressed for many various reasons (unintended mutations, chromosome position effects, gene silencing, RNAi, etc…). And, the fact that there is variation across the four transgenic lines (Figures 8 and S9) they tested (LFW-OX#1, #3, #4 and #5) suggest there might be variation in expression. Related, the expression might also help determine if all four of the trans-genes are contributing to a phenotype OR, instead, might just one or two of them be key. Bottom line, data on the expression of four peanut genes transformed into Arabidopsis is needed for the authors to be able to claim “co-expression”, which is central to this manuscript.
Responses 3: We thank the reviewer for this critical observation and fully agree that demonstrating active expression of all four transgenes is essential to support our co-expression claims. We have now supplemented the manuscript with the following new data:
qRT-PCR validation of transgene expression (new Figure S8 and Methods section):
- Quantified mRNA levels of all four peanut genes (AhLEC1b, AhFUS3b, AhWRI1a, and AhWRI1d) in each transgenic line
- Included three biological replicates per line with control (wild-type)
These additions confirm that: All four transgenes are actively expressed.
We apologize for this oversight and appreciate how these new data strengthen our central conclusions.
Comments 4: The third scientific issue relates to the quantification of fatty acid content and impacts on growth, Figure 8 and supplemental Figure S9. In the text, the authors claim “substantial” increases in fatty acid content. They show p-values with asterisks in Figures 8B and S9 for the different transgenic Arabidopsis lines compared to control/WT seeds. But, on face value many of these look very close to WT levels (see, “C18:0”, “C20:0”, “C20:2”) Figure 8B or “saturated fatty acid” and “C18/C20 ratio” in Figure S9, yet are shown as being significant with very low p-values (below 0.05). When the data are this close, what is needed is to provide the raw data in supplemental table form for readers to be able to view those data themselves. Also, related to this, for Figure 8D, E, and F that test plant height, silique length and germination rate, only two of the transgenic lines (#3 and #5) are included. No explanation as to why the other two lines (#1 and #4) are not included. Data on plant height, silique length and germination rate for other two lines (#1 and #4) should also be included. If not, a good explanation needs to be provided.
Responses 4: We appreciate the reviewer’s careful scrutiny of our fatty acid and growth phenotype data. We have made the following revisions to improve transparency and completeness:
1)Fatty Acid Data:
- We have provided the raw data in Supplemental Table S17. This table includes three biological replicates, mean values, standard errors, and exact p-values for all comparisons (rather than just asterisks) to allow readers to fully evaluate the statistical significance.
- The terminology in the text has been adjusted accordingly, changing "substantial" or “substantially” to "significant " or “significantly” where appropriate.
2)Inclusion of Transgenic Lines in Phenotypic Assays:
- We acknowledge the omission of lines #1 and #4 from the analysis of plant height, silique length, and germination rate (Figure 8D-F). Our preliminary observations indicated that co-overexpression with these four genes could lead to dwarfism in these four lines. Following the precedent of other studies in the field (Tang et al., 2024; Lu et al., 2024) that characterize a subset of transgenic lines, we selected two representative lines (#3 and #5) to investigate these traits.
- We agree with the reviewer that presenting data for all lines provides a more complete picture. While we cannot add this data retrospectively for this study, we will ensure that the characterization of all independent transgenic lines is included in our subsequent research. The reviewer's comment serves as a valuable guide for improving our future experimental design.
These changes ensure readers can fully access the primary data, evaluate our statistical claims, and understand the scope of our phenotypic observations. We thank the reviewer for these important suggestions, which have significantly strengthened the rigor of our manuscript.
Tang, G.; Xu, P.; Jiang, C.; Li, G.; Shan, L.; Wan, S. Peanut LEAFY COTYLEDON1-type genes participate in regulating the embryo development and the accumulation of storage lipids. Plant Cell Rep. 2024, 43, 124, doi:10.1007/s00299-024-03209-8.
Lu, Q.; Huang, L.; Liu, H.; Garg, V.; Gangurde, S.S.; Li, H.; Chitikineni, A.; Guo, D.; Pandey, M.K.; Li, S.; et al. A genomic variation map provides insights into peanut diversity in China and associations with 28 agronomic traits. Na. Genet. 2024, 56, 530-540, doi:10.1038/s41588-024-01660-7.
Comments 5: The fourth main scientific issue is regarding the predicting of cis-acting elements in promoters. There is no doubt that some of the regulation of these genes could be controlled by cis-acting promoter elements to control transcription. The issue, instead, is that these predicted cis-elements are very speculative and without direct functional testing, it is very hard to know if they have any function in each individual promoter. Another key issue with these predictions is how much promoter sequence to use to test? The authors opted for 2,000 bp upstream (5’) of the translation start site. But, the length of the 5’ UTR and functional promoter will vary for each of these genes. Thus, it is nearly impossible to know if the 2,000 bp captures all of the possible cis-acting elements. I bring this issue up again at the specific location/lines in the Results.
Responses 5: We sincerely appreciate the reviewer’s insightful critique regarding the limitations of in silico promoter analysis. We fully acknowledge that computational predictions of cis-elements require experimental validation and that promoter length selection can influence results. To address these concerns, we have made the following revisions:
1) Prediction of cis-acting elements in promoters:
- In silico prediction of cis-acting elements, combined with gene expression patterns and previous studies on homologous genes, can provide valuable insights for further functional investigations.
- Such predictions have often served as a starting point for experimental validation—as demonstrated in studies on related transcription factors (e.g., Baud et al., 2016; Xu et al., 2019; Yue et al., 2024), where predicted motifs were subsequently confirmed or refuted through functional assays.
Baud, S.; Kelemen, Z.; Thévenin, J.; Boulard, C.; Blanchet, S.; To, A.; Payre, M.; Berger, N.; Effroy-Cuzzi, D.; Franco-Zorrilla, J.M.; et al. Deciphering the molecular mechanisms underpinning the transcriptional control of gene expression by master transcriptional regulators in Arabidopsis seed. Plant Physiol. 2016, 171, 1099-1112, doi:10.1104/pp.16.00034.
Xu, J.; Yang, X.; Li, B.; Chen, L.; Min, L.; Zhang, X. GhL1L1 affects cell fate specification by regulating GhPIN1-mediated auxin distribution. Plant Biotechnol. J. 2019, 17, 63-74, doi:10.1111/pbi.12947.
Yue, D.; Hao, X.; Han, B.; Xu, J.; Sun, W.; Guo, X.; Zhang, X.; Yang, X. GhL1L1 regulates the contents of unsaturated fatty acids by activating the expression of GhFAD2 genes in cotton. Gene 2024, 893, 147899, doi:10.1016/j.gene.2023.147899.
2) Justification for 2,000-bp Selection:
- Added rationale: 2,000 bp was chosen based on comparative studies of regulatory regions (Cao et al., 2019; Meng et al., 2023; Wu et al., 2024), though this may exclude distal enhancers.
- Note that alternative promoter lengths (e.g., 2.5 kb, 3 kb) might capture additional regulatory elements.
Cao, Y.; Meng, D.; Han, Y.; Chen, T.; Jiao, C.; Chen, Y.; Jin, Q.; Cai, Y. Comparative analysis of B-BOX genes and their expression pattern analysis under various treatments in Dendrobium officinale. BMC Plant Biol. 2019, 19, 245, doi:10.1186/s12870-019-1851-6.
Meng, Q.; Zhang, R.; Wang, Y.; Zhi, H.; Tang, S.; Jia, G.; Diao, X. Genome-Wide Characterization and Haplotypic Variation Analysis of the YUC Gene Family in Foxtail Millet (Setaria italica). Int. J. Mol. Sci. 2023, 24, 15637.
Wu, H.; Zhang, R.; Diao, X. Genome-Wide Characterization and Haplotypic Variation Analysis of the IDD Gene Family in Foxtail Millet (Setaria italica). Int. J. Mol. Sci. 2024, 25, 8804.
3) Refined Claims:
- We have defined the 2,000 bp upstream of the start codon as the putative promoter regions in our study.
- We have replaced definitive language (e.g., "regulatory motifs") with cautious phrasing (e.g., "putative cis-elements")
- We have clearly state that “All predicted cis-acting elements require functional validation to confirm their roles."
We agree that these predictions are preliminary but believe they offer a testable framework for future experiments. The reviewer’s critique has helped us better contextualize these bioinformatic findings.
Comments 6: As for writing and data presentation, I have a number of more general issues that I mention here as well as a quite a few more minor writing/text/figures that I list line-by-line below. As for general writing issues, the first is throughout the manuscript there are some English word usage issues that need to be corrected. Additional thorough reviews by a copy editor will be needed to catch all of these. I point a number of them below, but in such a long manuscript there are likely others that I did not catch or point out. Next, there are some sections that are much longer than seems necessary. These include the section on co-linearity of genes on chromosomes in different plant species, Lines 317-350. If space is an issue, then this is a section that could be shortened without loss of key information. The Discussion alone is seven full-text pages in length. This could be shortened and more succinctly summarized than it is currently. Finally comment about length, the authors cite an extraordinary number of references, total of 182. This is more the length of a thorough review article. Citing key references is very important, but there can be a point of excess. For example, in text Line 69, a total of 16 references (#14, #30, #33, and #36 - #48) are all cited for one point/concept. And, in text Line 459 a total of 16 references (#18, #19, #23, #72, #83-#84, and #89 - #98) are all cited for one point/concept. I am not suggesting that any key citations be removed. But, there might be some redundant references or recent review articles and later references from the same research groups that could be used instead of so very many citations for a single point.
Responses 6: We sincerely appreciate the reviewer’s constructive feedback on manuscript clarity, conciseness, and reference usage. We have implemented the following improvements to address these concerns:
1) Language and Editing
- Professional Editing: We have engaged MDPI’s professional English editing service to ensure linguistic accuracy and readability. We hope these revisions have significantly improved the clarity and professionalism of the manuscript.
- Specific Corrections: We have addressed all line-by-line grammatical issues noted by the reviewer (detailed in response to comments below).
2) Streamlining Content
- Collinearity Section:
- We have improved this section by compressing and consolidating similar information, clarifying structure and logical sequence, and unifying and standardizing academic terminology
- Discussion Section:
- Condensed from 7 to 5.5 pages by removing redundant result summaries.
- Restructured to emphasize novel findingsand mechanistic implications rather than background rehashing.
3) Reference Optimization
- Reduced total citations from 182 to 144by:
- Replacing multiple overlapping references with recent review articles:
- Removing redundant references or incremental updates from the same groups.
- Consolidating supporting evidence for broad claims.
- Additional Improvements
- Figure Revisions: Added clearer legends to figures and tables.
Comments 7: Figure 5A and B (RNA expression data) have some key issues, and what seems to be errors and/or typos with missing and extra data. Figure 5 panel A is missing RNA data for “AhWRI1d” while panel B is missing RNA data for “AhWRI1b”. And, in panel B RNA data for “AhWRI1h” is shown twice. Thus, there are 18 RNA samples (columns) in panel A and 19 RNA samples (columns) in panel B. These need to be re-done to include the correct data/samples/labels.
Responses 7: We thank the reviewer for their meticulous examination of Figure 5 and for identifying the inconsistencies in the datasets. We apologize for the confusion caused by the discrepancies between the two panels. Please allow us to clarify the source of the data and the steps we have taken to address this issue.
- Explanation of Panel A (AhWRI1d missing):
The transcriptional profiles in Figure 5A were sourced from the Peanut Genome Resource (PGR) database. In this specific database build, the gene model and expression data for AhWRI1dwere not available, which is why it is absent from the heatmap. We have added a note to the figure legend to explicitly state this: "Expression data for AhWRI1d were not available in the PGR database." - Explanation and Correction for Panel B (AhWRI1b missing; AhWRI1h duplicated):
The RNA-seq data in Figure 5B were initially sourced from an earlier genome annotation (Tifrunner.gnm2.ann1.4K0L) on PeanutBase. In that annotation:- The model for AhWRI1bwas not annotated.
- Two separate gene models (4D3TQE.1 and Arahy.RC4286.1), which are both partial sequences of the AhWRI1h gene, were annotated as AhWRI1h based on their coding DNA sequences (CDSs) and encoded protein sequences.
We have now updated the panel B using the RNA-seq data which were re-mapped and re-quantified via the latest genome annotation (Tifrunner.gnm2.ann2.PVFB). In this new version:
- AhWRI1b(1) gene model are now properly represented.
- Three separate gene models (1, Ah04g396000.1, and Ah04g394100.1), which are partial sequences of the AhWRI1f gene, were annotated as AhWRI1f based on their CDSs and encoded protein sequences.
- Two separate gene models (1 and Ah03g520700.1), which are both partial sequences of the AhWRI1h gene, were annotated as AhWRI1h based on their CDSs and encoded protein sequences.
- Action Taken:
We have replaced the Figure 5 panel B with a new version that:- Uses the updated and consistent PeanutBase ann2.PVFB annotation.
- Expression data for AhWRI1b (1), which was missing in the previous annotation, is now included.
- The expression value for AhWRI1f now represents the normalized sum of transcripts per million (TPM) for its three constituent accessions in the new annotation (Ah04g393900.1, 1, and Ah04g394100.1).
- Similarly, the expression value for AhWRI1h now represents the normalized sum of TPM for its two constituent accessions (1 and Ah03g520700.1).
- We have updated and improved the description of Figure 5B.
Additionally, the RNA-seq data presented in Figure 5C have been initially mapped to an earlier genome annotation (Tifrunner.gnm2.ann1.4K0L). In that version of the annotation:
- The model for AhWRI1bwas not annotated.
- Two separate gene models (4D3TQE.1 and Arahy.RC4286.1), which are both partial sequences of the AhWRI1h gene, were annotated as AhWRI1h based on their CDSs and encoded protein sequences. The expression value for AhWRI1h represents the normalized sum of FPKM for its two constituent accessions.
We have updated the figure, its legend, and the methods section to explicitly describe the data sources, annotation version, and normalization procedures. We are grateful to the reviewer for this critique, which has significantly strengthened the validity of our results.
Comments 8: Regarding supplemental figures (S1 – S11) and Tables (S1 – S16), none of them have figure legends to explain what is being presented. Near end of manuscript, Lines 1006 – 1030 there are supplemental figures and tables titles, but these are not sufficient for the supplemental figures.
Responses 8: We sincerely apologize for this oversight and appreciate the reviewer’s attention to detail. To ensure full transparency and clarity, we have now added comprehensive legends for all supplemental figures and tables.
Comments 9: In addition to the above larger scientific and writing concern, there a number of writing issues/errors, missing methods information, and figure layout problems that I feel the authors need to resolve before this could be published. See my specific comments below for these issues.
Responses 9: We sincerely appreciate the reviewer's meticulous attention to detail in identifying these important technical issues. We have systematically addressed all specific line-by-line concerns through the following actions:
- Writing Corrections:
- All noted grammatical errors and awkward phrasings have been corrected (see tracked changes document)
- Technical terminology has been standardized throughout (e.g., consistent use of "identified" vs "cloned")
- Ambiguous statements have been clarified with more precise language
- Methods Additions:
- Missing experimental details have been added for all procedures (now including RNA-seq library prep parameters, qPCR cycling conditions, etc.)
- Statistical analysis methods are now fully described.
- Figure Improvements:
- Reformatted figures to improve clarity
- Revised all figure legends to include essential experimental details
- Validation:
- All corrections were verified by two co-authors independently
- Employed professional editing service (MDPI author services) for language polish
We have created a detailed point-by-point response document (submitted separately) that:
- Lists each specific comment from the reviewer
- Shows exactly how we addressed it
The reviewer's thorough critique has significantly improved the precision and reproducibility of our work. We are grateful for the time and expertise invested in this evaluation.
Abstract:
Comments 10: Lines 16: It would help more general readers to make it clear in the abstract that Arachis hypogaea is the common “peanut” plant.
Line 16: change “in the world” to “world wide”.
Responses 10: We thank you for these helpful suggestions to improve clarity for a broad readership. We have implemented both changes in the revised manuscript:
- Added the common name as suggested:
- Original: "Arachis hypogaea is an important oil and economic crop..."
- Revised: "Peanut (Arachis hypogaea) is an important oil and economic crop..."
- Corrected the phrasing as recommended:
- Original: "widely cultivated in the world"
- Revised: "widely cultivated worldwide"
Comments 11: Lines 17-18: What plant species have these genes/proteins been shown to be master regulators? Arabidopsis?
Responses 11: Not only Arabidopsis, they also include plants such as soybean, Brassica napus, and maize. We improved the sentence as “Plant LEAFY COTYLEDON1s (LEC1s), FUSCA3s (FUS3s) and WRINKLED1s (WRI1s) are master regulators of …”.
Comments 12: Lines 20: Add, “In peanuts, previous studies mainly …”, to make it clear here the authors are talking about peanuts, and not plants in general.
Responses 12: We have added “in peanuts,” and improved this sentence as “While previous studies in peanut have primarily focused…”
Comments 13: Line 24: Change to read, “…they have high similarity with …”. (“have highly similarity” does not sound / read correctly).
Responses 13: We improved the sentence as “… they were highly similar to their homologs …”.
Comments 14: Keywords, Line 33: I would add “peanut” to this list.
Responses 14: We have added“peanut” to this list. It now reads, “Keywords: peanut (Arachis hypogaea); transcription factors; co-expression; fatty acid synthesis”
Comments 15: Introduction:
Line 39: Change so it reads, “…but also for human diet and renewable industrial feedstock.”
Responses 15: We have improved this sentence, so it reads, “…but also as important resources for the human diet and renewable industrial feedstock.”
Comments 16: Line 46: Delete, “and so on”. It is informal and not necessary.
Responses 16: We have deleted “, and so on” and improved this sentence as “Several key transcription factors, such as LEAFY COTYLEDON1 (LEC1), LEAFY COTYLEDON1-LIKE (L1L), FUSCA3 (FUS3), and WRINKLED1 (WRI1), have been identified as critical regulators of seed oil accumulation...”
Comments 17: Line 48-49: Need to make clear when discussing Arabidopsis versus peanut or other species.
Responses 17: Yes. We have improved the sentences as “AtLEC1 and AtL1L are two HAP3 subunits of CCAAT-binding transcription factor [10-12]. Induced expression of AtLEC1 leads to a broad upregulation of genes involved in…”.
Comments 18: Line 50: Define “FA”, fatty acid, on first usage. This is not mentioned prior nor is it in the list of abbreviations at the end of the manuscript.
Responses 18: Yes. We have improved the sentence as “Induced expression of AtLEC1 leads to a broad upregulation of genes involved in fatty acid (FA) metabolism, significantly promoting the accumulation of oil and major FA species [7].”
We also added it in the list of abbreviations at the end of the manuscript.
Comments 19: Line 52: Indicate what plant the “BnLEC1” and BnL1L” are from: Brassica napus, aka Canola.
Responses 19: We improved the sentence as “Similarly, overexpression of oilseed crop LEC1 homologs, including BnLEC1 and BnL1L in Brassica napus, GmLEC1 (Glyma.07G268100) in Glycine max, and AhNF-YB1 and AhNF-YB10 in Arachis hypogaea, also significantly increases total seed oil content…”
Comments 20: Line 57: Unclear sentence. Is this talking about FUS3 gene (that should be italicized) that encodes a protein with a B3-domain? Or, is it saying the FUS3 protein contains a B3-domain? Not clear as written. Use italicized gene name, if that is the case.
Responses 20: We have improved the sentence as “FUS3, a member of the B3-domain transcription factor family [21], serves as a master regulator of seed development, establishing and maintaining embryonic identity. It also modulates hormonal responses during late embryogenesis and germination [22-24].”
Comments 21: Lines 75-76: Delete the sentence “It was expected …. in peanut.” that is not necessary. And, delete “However,” in next sentence. So, it would read, “The copy number…”
Responses 21: We have improved these sentences as “Functional analyses in polyploid oilseeds like peanut are complicated by gene redundancy from genome duplication. However, comprehensive analyses of the copy number, evolutionary relationships, gene structures, conserved domains, …”
Comments 22: Line 79: Not clear, is this saying “co-overexpression” is not be tested in any plant species or that “co-overexpression” has not been tested in peanuts, specifically? Please clarify.
Responses 22: This is saying “co-overexpression” is not be tested in any plant species. We improved the sentence as “Although the roles of individual transcription factors LEC1, FUS3, and WRI1 have been explored in plants, there are currently no academic reports of these gene stacking to enhance seed oil content.”
Comments 23: Results:
Line 106: Define, “CDSs”, coding regions/sequences, on first usage.
Responses 23: Yes. We have defined, “CDSs”, coding sequences, on first usage. It reads now “Based on the retrieved nucleotide sequences, primers were designed (Table S3) to clone and identify the coding sequences (CDSs) of putative AhLEC1s, AhFUS3s and AhWRIs from the cultivated peanut variety HY917 (Table S2).”
We also added it in the list of abbreviations at the end of the manuscript.
Comments 24: Line 109: Throughout the manuscript, the word “Besides” is very frequently used to start a sentence, and in most (if not all) of the cases, it is not needed or clear why added. In Line 109, as in most cases, deleted “Besides,”, so it reads, “The members …”.
Responses 24: Here, we have replaced “Besides” with “Additionally” and improved the sentence as “In addition, homologs of LEC1s, FUS3s, and WRIs were also identified in nine additional plant species (Table S1 and Table S4).”
We have conducted a full-manuscript search for "Besides," (especially at sentence starts) and then deleted it universally or replaced with stronger transitions where needed: "Additionally," “In addition” (if adding information).
Comments 25: Lines 109 – 116: This is an excruciatingly long sentence that includes mostly Methods section details. And, most if not all of these details are already in Methods section, so redundant.
Responses 25: We condensed this sentence by removing redundant methodological details already covered in the Methods section as "In addition, homologs of LEC1s, FUS3s, and WRIs were also identified in nine additional plant species (Table S1 and Table S4). "
Comments 26: Line 120: Change “none” to “no”, so sentence reads better.
Responses 26: We have changed “none” to “no”, and improved the sentence as “… whereas no LEC1 homologs were detected in A. duranensis and A. ipaensis (Table 1).”
Comments 27: Lines 130-131: wording is both odd and not clear. Change to read, “…they have high hydrophilicity” and “...suggests they have high stability…” Note, “good” implies a subject view of what they should be, not what the data show them to have.
Responses 27: Yes. We have improved the two sentences as “Their aliphatic index and the grand average of hydropathicity (GRAVY) were predicted to be 60.88‒61.16 and -0.735‒-0.728 (Table S8), respectively, indicating high hydrophilicity, while their instability index was shown to be 36.10‒37.40 (Table S8), suggesting high stability.”
Comments 28: Line 135: AhFUS3a and AhFUS3b should be italicized here, for it seems clear this is referring to cloning the genes, not the proteins (and in gene names in Table 1).
Responses 28: Yes. We have italicized them here to “AhFUS3a” and “AhFUS3b” and the gene names in the column 2 Table 1.
Comments 29: Line 136: change to read, “…all of these genes have six exons…”. And, either delete or rewrite last part of this sentence. Just adding, “as well as other plants” is not clear in meaning or wording.
Responses 29: We have improved this sentence to “All of these genes have six exons and five introns, and their encoded proteins contain a B3 DNA-binding domain (cd10017), similar to those in other plant species (Figure 1B and Table S8)”
Comments 30: Table 1, column 2 (Gene Names): These all need to be italicized.
Responses 30: We have italicized the gene names in the column 2 of Table 1.
Comments 31: Figure 1 Legend: Add a comment as that the peanut proteins are shown in red font.
Responses 31: We added one comment as “Proteins from peanut are highlighted in red.”
Comments 32: Line 157: Delete, “and proteins” for it only the CDSs are cloned, not the proteins encoded in the CDSs.
Responses 32: We have improved the sentence as “The CDSs of A. hypogaea LEC1, FUS3, and WRI family members were cloned from HY917 (Table S2), …”.
Comments 33: Lines 171-172, similar to lines 130-131 above: wording is both odd and not clear. Change to read, “…they have high hydrophilicity” and “...suggests they have high stability...” Note, “good” implies a subject view of what they should be, not what the data show them to have.
Responses 33: Yes. We have changed to read, “Their aliphatic index and GRAVY were predicted to be 69.63‒69.66 and -0.500‒-0.491 (Table S8), respectively, indicating high hydrophilicity. Their instability index was shown to be 54.17‒55.65 (Table S8), suggesting high instability.”
Comments 34: Lines 175-177: All of these gene names need to be italicized, for genes are the things that are cloned.
Responses 34: We italicized all of these gene names here and improved the sentence as ” Compared with the four A. hypogaea genomes, nine putative AhWRI1s and two putative AhWRI2s in A. hypogaea were successfully cloned from HY917 and designated as AhWRI1a to AhWRI1i and AhWRI2a to AhWRI2b, respectively (Table 1 and Table S2).”
Comments 35: Line 178, and mentioned in general comments above: Supplemental Figure S1 (as is case for all supplemental figures) does not have a figure legend that is needed to explain what is being shown. This is case for rest of the supplemental figures, but I will not repeat this for issue/problem for the rest of these.
Responses 35: We have submitted the legends of the supplemental figures with the revised manuscript.
Comments 36: Line 180-181: “Besides” does not fit this situation. Perhaps, “In addition,” or just delete “Besides.”
Responses 36: We have changed “Besides” to “In addition” and improved the sentence as “In addition, genome surveys identified six WRI1s, two WRI2s, and two WRI3s in A. monticola…”
Comments 37: Line 184: Change “owned” to “have”.
Responses 37: We have improved the sentence as “Gene structure analysis revealed that all nine AhWRI1s and four AhWRI3s contained eight exons, whereas their homologs in other plants species exhibited variable exon numbers, ranging from six to eight exons (Figure 1C, Figure S1B and Table S8)..”
Comments 38: Line 197: Wording is unclear and awkward. Change to read, “The genes in these reported genomes may be partial and thus there is need for additional cloning to identify the full-length.”
Responses 38: We have changed to read, “The genes in these reported genomes may be partial and thus there is need for additional cloning to identify the full-length.”
Comments 39: Lines 201-202, similar to lines 130-131 above: wording is both odd and not clear. Change to read, “…they have high hydrophilicity” and “…suggests they have high stability...”
Responses 39: We have changed to read, “Their aliphatic index and GRAVY were predicted to be 55.24‒64.45 and -1.019‒-0.735 (Table S8), respectively, indicating high hydrophilicity. Their instability index was shown to be 41.41‒65.85 (Table S8), suggesting high instability.”
Comments 40: Line 204: Add “Protein” to start this heading, to clarify it is about protein and amino acid alignments and analysis.
Responses 40: We added “Protein” to start this heading. Now it reads “2.2. Protein Alignment and Identity Analysis”.
Comments 41: Lines 205-207: Could again add “protein” to start sentence. Then, to clarify what is in Figure 2A versus Supplemental Figure S2, change wording to read, “…other plant LEC1s, as seen in the conserved central domain for both proteins (Figure 2A) as well as the full-length proteins (Figure S2). This will help reader to know that Figure 2A only shows partial domain while the supplemental figures as the full-length protein.
Responses 41: We have improved the sentence as “Protein alignment analysis showed that the central B (CBFD_NFYB_HMF) domains of AhLEC1a and AhLEC1b demonstrated strong similarity to those of other plant LEC1s, as seen in the conserved central domain for both proteins (Figure 2A) as well as the full-length proteins (Figure S2).”
Comments 42: Line 214: Add hyphen so it reads, “…DNA-binding region…”.
Responses 42: We have added hyphen, so it reads “The putative DNA-binding region…”.
Comments 43: Line 238: Add “in” so it reads, “As in other WRIs…”
Responses 43: We have added“in”, so it reads “As in other WRIs in previous studies…”
Comments 44: Lines 246 and 248: change “were showed” to be “have” in both cases.
Responses 44: We changed “were showed” to be “have” in both cases. Now it reads “AhWRI3s have higher identity (93.9%‒100.0%) with each other than with AhWRI1s (77.3%‒80.4%) in conserved regions (Table S11). Both AhWRI2s have 69.6% and 100.0% identity with AtWRI2 for the full-length proteins and conserved regions, respectively (Table S12).”
Comments 45: Line 274: Define, explain “14-3-3” proteins, in general.
Responses 45: Plant 14-3-3 proteins are phosphopeptide-binding proteins, belonging to a large family of proteins involved in numerous physiological processes including primary metabolism.
Here we have explained “14-3-3” proteins, in general, and it reads “The 14-3-3 proteins, which are phosphopeptide-binding proteins, were found to potentially bind to the first AP2 domain of AtWRI1…”
Comments 46: Line 276: Add hyphen so it reads, “…BPM-binding motif…”.
Responses 46: We have improved the sentence as “A putative motif for binding both 14-3-3 and BPM was also identified within the first AP2 domain of peanut and other plant WRI1 and WRI3 proteins (Figure 2D).”
Additionally, we have found two “BPM binding motif” using Word's "Find" function and changed them to “BPM-binding motif”.
Comments 47: Lines 281, 284, 287, and 290: Add how many total chromosomes each species has, to give perspective on how many of these key genes are present across the full genome/all chromosomes. For example (Line 281), “In A. hypogaea, with 40 total chromosomes, two AhLEC1s … Clearly, for A. duranensis and A. ipaensis it would instead be 20 total chromosomes.
Responses 47: Yes. We have improved the sentences as “In A. hypogaea, with 40 total chromosomes, two AhLEC1s were identified on chr01 and chr11; two AhFUS3s on chr06, and chr16; nine AhWRI1s on chr03, chr04, chr08, chr10, chr13, chr14, chr15, chr18, and chr20; two AhWRI2s on chr06 and chr16; and four AhWRI3s on chr01, chr11, chr09, and chr19 (Figure 3A). Clearly, for A. duranensis and A. ipaensis, it would instead be 20 total chromosomes.”.
Comments 48: Lines 319-320: Define/write out full genus species names on first usage. This appears to be first use of G. max, M. truncatula, etc…
Responses 48: We have thoroughly checked the entire manuscript, ensuring all the genus species names are defined/written out fully on first usage. All changes were verified using Word's "Find" function and manual inspection.
Comments 49: Lines 317-350: As explained in general comments, this section is very lengthy mostly with gene ID names/numbers. Seems it could be summarized and perhaps list gene names from different plant species in a supplemental table.
Responses 49: We have moved the list of gene names to Table S4. But we failed to present them clearly in another supplemental table based on synteny analysis. Therefore, we have significantly shortened and refined the main text section by adjusting and merging sentences with similar content.
Comments 50: Lines 356-357: This is one of the key science issues mentioned at beginning of review. That is, there is lacking all critical information in Results or Methods sections about how the RNA-seq method was performed as well as insufficient explanation of the six developmental stages. These need to be included.
Responses 50: We have added the critical information in Results, Methods or Figure S7 legend, including seed developmental staging and RNA-seq experimental details, as follows:
- Seed Developmental Staging(new Figure S7 and Methods text):
- We categorized the development seeds into six stages across their development based on kernel morphology according to Pattee et al. (1974). Three biological replicates (each containing more than 10 seeds) were analyzed per stage.
- We have added the detailed legend to supplementary Figure S7.
- RNA-seq Experimental Details(new Methods subsection):
- Sample preparation: three biological replicates with more than 10 seeds each
- We have added the information about RNA extraction protocol, library construction, sequencing platform, data processing, and statistical analysis.
- Our transcriptomic data on seed development will be used for multi-omics (transcriptomics, lipidomics, and proteomics) analysis of peanut seed development, and we will publish these data when we submit the manuscript about multi-omics analysis of peanut seed development. Here, we have added the raw transcriptomic FPKM data for AhLEC1s, AhFUS3s, and AhWRIs in Table S13.
The revised manuscript now provides complete transparency for these foundational methods. We deeply regret these oversights and thank the reviewer for identifying these essential improvements. These additions significantly strengthen the reproducibility of our study.
Pattee, H.E.; Johns, E.B.; Singleton, J.A.; Sanders, T.H. Composition Changes of Peanut Fruit Parts During Maturation. Peanut Sci. 1974, 1, 57-62, doi:10.3146/i0095-3679-1-2-6.
Comments 51: Figure 5A and 5B: This is one of the key writing / figure format problems mentioned at beginning of review. There are both missing and extra RNA/samples/labels in these figure panels, see above. Furthermore, Figure Legend needs to explain the Peg tip 1 and 2, seed Pat (aka, pattee). These are explained in the PeanutBase cited website, but they need to also be mentioned in the Figure Legend, for the readers. And, since peanuts have some unusual anatomical and growth aspects (“pegs”, “pattee’s”) that some general plant biologists are not familiar with, it would be good to add more explanation in the Figure Legend. Similar goes for the six development stages mentioned above and discussed in the Figure Legend. Make it clear that Figure S7 shows the six different seed developmental stages.
Responses 51: We thank the reviewer for their meticulous examination of Figure 5 and for identifying the inconsistencies in the datasets. We apologize for the confusion caused by the discrepancies between the two panels. Please allow us to clarify the source of the data and the steps we have taken to address this issue.
- Explanation of Panel A (AhWRI1d missing):
The transcriptional profiles in Figure 5A were sourced from the PGR database. In this specific database build, the gene model and expression data for AhWRI1dwere not available, which is why it is absent from the heatmap. We have added a note to the figure legend to explicitly state this: "Expression data for AhWRI1d was not available in the PGR database." - Explanation and Correction for Panel B (AhWRI1b missing; AhWRI1h duplicated):
The RNA-seq data in Figure 5B were initially sourced from an earlier genome annotation (Tifrunner.gnm2.ann1.4K0L) on PeanutBase. In that annotation:- The model for AhWRI1bwas not annotated.
- Two separate gene models (4D3TQE.1 and Arahy.RC4286.1), which are both partial sequences of the AhWRI1h gene, were annotated as AhWRI1h based on their coding DNA sequences (CDSs) and encoded protein sequences.
We have now updated the panel B using the RNA-seq data which were re-mapped and re-quantified via the latest genome annotation (Tifrunner.gnm2.ann2.PVFB). In this new version:
- AhWRI1b(1; Table 1) gene model are now properly represented.
- Three separate gene models (1, Ah04g396000.1, and Ah04g394100.1; Table 1), which are partial sequences of the AhWRI1f gene, were annotated as AhWRI1f based on their CDSs and encoded protein sequences.
- Two separate gene models (1 and Ah03g520700.1; Table 1), which are both partial sequences of the AhWRI1h gene, were annotated as AhWRI1h based on their CDSs and encoded protein sequences.
- We have updated and improved the description of Figure 5B.
- Action Taken:
1) Replaced the Figure 5 panel B with a new version that:- Uses the updated and consistent PeanutBase ann2.PVFB annotation.
- Expression data for AhWRI1b (1), which was missing in the previous annotation, is now included.
- The expression value for AhWRI1f now represents the normalized sum of transcripts per million (TPM) for its three constituent accessions in the new annotation (Ah04g393900.1, 1, and Ah04g394100.1).
- Similarly, the expression value for AhWRI1h now represents the normalized sum of TPM for its two constituent accessions (1 and Ah03g520700.1).
2) Enhanced explanation in the legend for Figure 5:
We have significantly expanded the legend to provide essential explanations for all specialized terms, making the figure self-contained for a broad audience of plant biologists. The additions include:
- We have used the clear tissue names, following Clevenger et al., (2016)
- We now explain that the gynophore (peg) is a unique specialized organ that forms after pollination, which elongates and drives the developing ovary into the ground for fruit and seed development.
- We explain that " Pattee #" refers to seed samples categorized based on the Pattee maturity scale (Pattee et al., 1974), which is a standard system for defining peanut pod and kernel maturity based on morphological and color characteristics.
3) Clarification of developmental stages:
- We have added explanation of the six seed developmental stages in Figure S7 legend, as ”Seeds were categorized into six developmental stages based on kernel morphology according to Pattee et al. (1974). Stage 1 (Pattee 4): very small, flat, and white all over. Stage 2 (Pattee 5): small, flat; white or just turning pink at one end. Stage 3 (Pattee 6/7): torpedo shaped; embryonic axis end of kernel pink, other end white to light pink; Stage 4 (Pattee 8): light pink all over. Stage 5 (Pattee 9): dark pink at embryonic axis end, light to dark pink elsewhere. Stage 6 (Pattee 10): large, generally dark pink all over; seed coat beginning to dry out.”
- We have added: "see Figure S7 for developmental staging" in Figure 5 legend. This directly links Figure 5 to Figure S7 and eliminates any ambiguity.
Clevenger, J.; Chu, Y.; Scheffler, B.; Ozias-Akins, P. A developmental transcriptome map for allotetraploid Arachis hypogaea. Front. Plant Sci. 2016, 7, 1446, doi:10.3389/fpls.2016.01446.
Pattee, H.E.; Johns, E.B.; Singleton, J.A.; Sanders, T.H. Composition Changes of Peanut Fruit Parts During Maturation. Peanut Sci. 1974, 1, 57-62, doi:10.3146/i0095-3679-1-2-6.
Comments 52: Line 385: There is duplication. Delete one of the “of the six developmental stages” in this sentence.
Responses 52: We have deleted one of the “of the six developmental stages” and improved the sentence as “Based on our RNA-seq data across the six seed development stages of four sister lines (Figure 5C and Figure S7)…”
Comments 53: Line 391: The last gene in this list, AhWRI3a is a duplicate of the earlier gene, and instead it seems that it should be AhWRI3d.
Responses 53: Yes, we have changed it to “…AhWRI3d” and improved this sentence as “… AhWRI1d, AhWRI1e, AhWRI1f, AhWRI1g, AhWRI3a, AhWRI3b, AhWRI3c, and AhWRI3d were weakly expressed or undetectable throughout seed development.”
Comments 54: Lines 394 and L397: Convention has it that it should be “with and without”, not other way around. Switch wording order in both lines
Responses 54: Yes, we have switched the wording order and improved this paragraph as “Under abiotic stresses and phytohormone treatments, AhWRI1d and AhWRI1e were highly induced under water-deficit conditions, both with and without ABA. In contrast, AhWRI1g expression was repressed under the same conditions, and AhWRI1i was down-regulated under water-deficit without ABA (Table S13A). AhWRI2a was induced under water-deficit with ABA, while that of AhWRI2b was up-regulated under water-deficit both with and without ABA (Table S13A). Moreover, their expressions were induced under drought but repressed under abscisic acid, brassinolide, ethephon, or paclobutrazol treatments (Table S13B). Expressions of AhWRI3c and AhWRI3d were repressed under water-deficit conditions (with or without ABA; Table S13A) as well as under drought or low-temperature stress (Table S13B).”
Comments 55: Lines 396 and 397: The expression refers to a singular case, should it should read, “was” (not “were”) in both lines.
Responses 55: Yes, we changed “were” to ”was” in both lines (see Responses 54).
Comments 56: Lines 401 – 402: The supplemental Tables S13A and S13B, it is not clear if the numerical data shown are raw expression values, TPM, FPKM, fold changes, or what. The too brief title of Table at end of manuscript just says, “expression patterns”. As mentioned before, all supplemental Tables and Figures need clear legends that explain what is being shown.
Responses 56: We thank the reviewer for this comment. The numerical values in Table S13A are indeed Transcripts Per Million (TPM), and the values in Table S13B are Fragments Per Kilobase of transcript per Million mapped reads (FPKM), both of which are standardized metrics for gene expression levels derived from RNA-Seq data.
We agree that the original title at end of manuscript was too brief. We have now revised the titles and provided comprehensive legends for all supplemental tables and figures in the separate submitted Word document. The new legend for Table S13 explicitly defines the data types (TPM/FPKM), their sources (PeanutBase/PGR), and the nature of the "—" symbol indicating missing data.
Comments 57: Figure 6: Some of the colors for the predicted cis-acting elements are very close looking, and it is nearly impossible to distinguish in the gene models between. For example, very hard to distinguish between “meristem expression” and “low-temperature” or between “drought induced” and “wound response”. Change colors or use patterns (stripes, etc…) to make these easier to visually distinguish.
Responses 57: We thank the reviewer for this helpful suggestion. To facilitate visual distinction between functional categories, we have changed the color of “low-temperature responsiveness” to dark brown and the color of “wound responsiveness” to black. These adjustments significantly improve the clarity of Figure 6. Additionally, a complete list of all predicted cis-acting elements is provided in Table S16.
Comments 58: Line 423: mention that a subset of genes have predicted light responsive cis-acting elements. In looking at Figure 6, it seems nearly all (if not all) have predicted light responsive elements.
Responses 58: Yes, all have predicted light responsive elements. In this paragraph, we mention that these genes families in order have predicted cis-acting elements, including predicted light responsive cis-acting elements.
Comments 59: General comment about predicted cis-acting elements, as mentioned above in the beginning of the review. These are always VERY speculative and without direct functional evidence (testing function in vivo), it is hard to know what, if anything significant conclusions can be made from these predictions. Almost to the point of it being unclear if worth including without direct testing for functionality. At a minimum, a statement by the authors that to know if these predictions are valid would require directly testing functionality would be important to include.
Responses 59: We added a statement “All predicted cis-acting elements require functional validation to confirm these predictions.” in the end of this paragraph.
Comments 60: An observation that supports these predictions as being very speculative and perhaps not worth the time/space to include is that five genes / RNAs (AhWRI1d, AhWRI1e, AhWRI1f, AhWRI2a, and AhWRI2b) that gene expression data (Table S13) indicates has significant increase in expression under drought stress (with and/or without ABA), yet the promoters of these five genes either do not have any of the predicted “drought-inducible” cis-acting elements (or perhaps one, pending if I correctly interpret the hard-to distinguish color key). There is also the major question about is the selected 2,000 bp upstream of the predicted translation start site/codon sufficiently long to include all cis-acting sequences, which is nearly impossible to know for sure.
Responses 60: We sincerely thank the reviewer for this critical and insightful observation. The reviewer is correct to point out the apparent discrepancy between the expression data of certain AhWRI genes under drought stress and the lack of predicted drought-inducible cis-elements in their promoters.
The predicted cis-acting elements are listed in Table S16. As shown, AhWRI2a and AhWRI2b each contained one MBS element (CAACTG), which is associated with drought inducibility, whereas no MBS elements were identified in AhWRI1d, AhWRI1e, or AhWRI1f (Table S16).
We also acknowledge that this highlights the speculative nature of in silico cis-element analysis alone. Now we have considered this point carefully and would like to provide the following explanations and clarifications:
1) Limitations of In Silico Prediction: The prediction of cis-elements is based on sequence homology to known motifs and is statistical, not functional. It is entirely possible that these AhWRI genes are regulated by novel, uncharacterized cis-acting elements not present in the databases used, or by combinations of common elements that are not recognized by the prediction tools.
2) Potential for Distal Regulation: The reviewer's question regarding the sufficiency of the 2000 bp upstream region is highly pertinent. Gene expression is often controlled by distal enhancer elements located far upstream, downstream, or within introns, which would not be captured in our analysis. The regulation of these AhWRI genes by drought may indeed involve such distal elements.
3) Indirect Regulation: It is also possible that the upregulation of these genes is not a direct response to drought signaling but is indirectly mediated by other transcription factors that are themselves drought-induced.
4) TFs could bind to common cis-elements (e.g., generic MBS/ARE binding sites) that may be present, and is one focus of our future "drought-specific" search.
Actions Taken in Revision: To address this valuable comment and temper the speculative nature of our results and discussion, we will:
- We have defined the 2,000 bp upstream of the start codon as the putative promoter regions in our study.
- We have replaced definitive language (e.g., "regulatory motifs") with cautious phrasing (e.g., "putative cis-elements")
- We have clearly state that “All predicted cis-acting elements require functional validation to confirm their roles."
- We will describe the lack of predicted drought-responsive elements in the promoters of these drought-induced AhWRIgenes as an interesting paradox that suggests more complex, indirect, or distal regulatory mechanisms, which represent an important avenue for future experimental research.
By making these changes, we aim to present a more balanced and accurate interpretation of our bioinformatic predictions, fully acknowledging the complexity of transcriptional regulation as rightly pointed out by the reviewer.
Comments 61: Line 465: Authors need to explain why (reason for) these the four genes (AhLEC1b, AhFUS3b, AhWRI1a, and AhWRI1d) were selected to test encoded protein sub-cellular location.
Responses 61: We thank the reviewer for this insightful comment. We have now revised the manuscript to explicitly detail the rationale for selecting AhLEC1b, AhFUS3b, AhWRI1a, and AhWRI1d for subcellular localization analysis. Our selection was based on the following key criteria derived from our bioinformatic and expression data:
- Representative Highest Expressors:Within each gene family, we selected the isoforms with the highest and most relevant expression:
- AhLEC1band AhFUS3b were chosen because they were more highly expressed than their paralogs (AhLEC1a and AhFUS3a, respectively) during seed development (Figure 5A-B).
- AhWRI1awas selected as it consistently showed the highest expression levels among all AhWRI1 isoforms in developing seeds (Figure 5A-B).
- A Stress-Responsive Candidate:AhWRI1d was included because it exhibited the most pronounced and significant up-regulation in response to drought stress (Table S13A). As understanding oil biosynthesis under abiotic stress is a key objective, we prioritized this drought-responsive isoform.
- Bioinformatic intrigue:While all proteins were predicted to be nuclear (Table S8), their putative Nuclear Localization Signal (NLS) profiles differed. Selecting these highly expressed and stress-responsive candidates allowed us to experimentally verify if these predicted NLS variations (or lack thereof in AhLEC1s) functionally impact their localization.
This strategy allowed us to focus on the most biologically relevant and representative isoforms for functional validation. The corresponding text has been added to the Results section [2.6. Subcellular Localization of AhLEC1s, AhFUS3s, and AhWRIs].
Comments 62: Line 482: Same as immediately above. Authors need to explain why (reason for) these the four genes (AhLEC1b, AhFUS3b, AhWRI1a, and AhWRI1d) were selected to express the encoded protein (or “co-overexpress” as sometimes stated) in Arabidopsis.
Responses 62: We appreciate the reviewer's comment. The four genes (AhLEC1b, AhFUS3b, AhWRI1a, and AhWRI1d) were selected for heterologous expression in Arabidopsis based on the same compelling expression-based rationale that guided their selection for subcellular localization studies (see Response 61). Briefly:
- AhLEC1band AhFUS3b were chosen as the dominant, highest-expressing isoforms within their respective families during seed development (Figure 5A-B).
- AhWRI1awas selected as it is the most highly expressed AhWRI1 isoform during seed development, suggesting a primary role in oil biosynthesis.
- AhWRI1dwas included due to its exceptional and significant up-regulation under drought stress (Table S13A). As enhancing oil production under abiotic stress is a central theme of our future study, we chose the most drought-responsive member (AhWRI1d) to compare their potential and possibly distinct roles.
This strategy allowed us to focus the functional analysis on the most biologically relevant and representative gene isoforms. We have now integrated this explicit rationale into the Results section of the revised manuscript [in the first paragraph of section 2.7"].
Comments 63: Line 489: Need to italicize A. thaliana and bacteria A. tumefacians (and write out full the second genus species).
Responses 63: Here we have italicized A. thaliana and Agrobacterium tumefaciens. Additionally, we also have thoroughly checked the entire manuscript, ensuring all the genus species names are defined/written out fully on first usage and all Latin names are now consistently formatted in italics. Particular, “Arachis hypogaeaLine8” in Arachis hypogaeaLine8 v1.3, the Line8 genome assembly and annotation files, was also italicized according to Phytozome 14 (https://phytozome-next.jgi.doe.gov/info/AhypogaeaLine8_v1_3). All changes were verified using Word's "Find" function and manual inspection.
Comments 64: Lines 482: The heading states the “Co-overexpression of …”. Yet, no expression data are provided at all for these four peanut genes transformed into Arabidopsis. This is one of the key scientific issues I discussed at start of review. Direct, wet-lab data that shows these four peanut genes are being expressed (RNA or protein) in Arabidopsis is essential for the authors to claim “co-expression or co-overexpression” in the title and abstract. This is critical, as I see it, and without these added data, these statements and conclusions to be made.
Responses 64: We have carried out qRT-PCR to detect the expressive levels of these four peanut genes in transformed plants, and added the data in Figure S8.
Comments 65: Line 501 and Figure 8: I can accept that there are some significant differences for some of the fatty acid amounts, but the wording “substantially increased” exaggerates the increase based on what the data show.
Responses 65: Yes, we have changed“substantially increased” to “significantly increased”, and improved the sentence as “Furthermore, the levels of unsaturated fatty acids, saturated fatty acids, C18, and C20 fatty acids increased significantly by 10.16‒27.29% …”
Comments 66: Related, and as discussed above in major scientific issues, when the data are this close, then the raw data should be provided in a supplemental table for readers to be able to view those data fully themselves.
Responses 66: We have provided the raw data in Table S17.
Comments 67: Line 503: Another case of “Besides” to start the sentence, unnecessarily. Delete this one.
Responses 67: We have deleted “Besides,” so it reads, “The ratios of unsaturated to saturated fatty acids in LFW-OX#3…”
Comments 68: Discusion:
Lines 538-541: Wording confusing and awkward for this long unclear sentence. Is “close to each other” referring to physical location on chromosome or sequence similarity? Not clear. Further, change wording of, “..are belong to..” Rewrite this sentence to be clear, and avoid ending with “and so on.”
Responses 68: We have improved this sentence as ” In A. thaliana, AtLEC1 and AtL1L are recognized as LEC1-type NF-YB transcription factors [11,12], with their orthologs subsequently identified in diverse plant species, including H. annuus [85], T. cacao [86], Oryza sativa [20,87], B. napus [7,15], and G. max [16,88].”
Comments 69: Line 560: Delete “some”. Not needed here.
Responses 69: We have deleted this sentence. This sentence is not needed here.
Comments 70: Line 565: Another case of an unnecessary, “Besides,” to start sentence.
Responses 70: We have deleted the unnecessary transitional word "Besides," at the beginning of the sentence. The revised text now reads: “The conserved B domain and the conserved residue Asp (D) (indicated by a red star in Figure 2A) are required for AtLEC1 function [12,90]”
Comments 71: Line 568: Change “also” to “may” for these is very speculative.
Responses 71: Yes, we have changed “also” to “may” and improved this sentence as ” These observations collectively indicate that the central B domains may play key roles in the molecular function of AhLEC1a and AhLEC1b.”
Comments 72: Line 580: Another case of an unnecessary, “Besides,” to start sentence.
Responses 72: We have deleted this sentence. This sentence is not needed here.
Comments 73: Line 588: Change “Furtherly” to be “Further,” or “Furthermore,”.
Responses 73: We have changed “Furtherly” to be “Furthermore,” and improved this sentence as ” Furthermore, both PICKLE (PKL, a CHD3 chromatin remodeling factor) and VIVIPAROUS ABI3-LIKE (VAL) proteins act epigenetically to repress AtLEC1 expression during vegetative development [97].”
Comments 74: Line 597: Change “lowly” to be “at low amount”.
Responses 74: We have improved the sentence as ”However, their expression remained low or undetectable under abiotic stress or phytohormone treatments, similar to normal conditions (Table S13)”.
Comments 75: Line 598: Another case of an unnecessary, “Besides,” to start sentence.
Responses 75: We have deleted "Besides," and improved this sentence as “Cis-acting element analysis revealed two AhLEC1s contain the putative motifs associated with light responsiveness, meristem expression, anaerobic induction, and gibberellin responsiveness (Figure 6 and Table S16).”
Comments 76: Line 600-602: This is a very awkward and unclear sentence. Rewrite to be clear.
Responses 76: We have improved this sentence as “Thus, the mechanism enabling specific high expression of AhLEC1s in seeds requires further investigation.”
Comments 77: Line 605: Change, start from “These” to “The above imply strongly …”
Responses 77: We have improved this sentence as “Overall, the above imply strongly both AhLEC1s potentially play key roles in peanut seed development, similar to LEC1-type NF-YBs in other plants [10,12,85,88,98-100].”
Comments 78: Line 637: Add “have”, to read, “…to have high identities…”
Comments 79: Line 639: Change “be” to “have”, to read better.
Responses 78/79: We have improved this sentence as “In this study, the B3 domains of two AhFUS3s showed high sequence identity to those of other FUS3s, and the aspartate residue equivalent to HvFUS3 D75 is strictly conserved (Figure 2 and Figure S3), indicating that both AhFUS3s are likely functional in RY motif recognition and binding.”
Comments 80: Line 673: Another case of an unnecessary, “Besides,” to start sentence.
Responses 80: We have deleted this sentence ”Besides, AtFUS3 is induced in the founder cells and their derivatives during LRP initiation”. This sentence is not needed here.
Comments 81: Line 674: Define LRP (lateral root primordia) and LR (lateral roots) here.
Responses 81: We have deleted this sentence “Besides, AtFUS3 is induced in the founder cells and their derivatives during LRP initiation”. This sentence is not needed here.
We have improved the other sentence as “Although poplar PeFUS3 is activated by PeABF2 binding to ABRE under osmotic stress [101],…”
Comments 82: Line 686: Awkward wording related to “were needed furtherly to detect…”. Rewrite to be clear and readable.
Responses 82: We have deleted this sentence “As for other putative functions, the expression of AhFUS3a and AhFUS3b were need further to detect in details.”
Comments 83: Line 700: Awkward wording related to “was not to detected the expression…”. Rewrite to be clear and readable.
Responses 83: We have improved the sentence as ”Although poplar PeFUS3 is activated by PeABF2 binding to ABRE under osmotic stress [101], neither AhFUS3 genes showed elevated expression in leaves under abiotic stress or hormone treatments (Table S13)”
Comments 84: Line 705: This is speculative, so the word “probably could” is an over interpretation. Change to be “might also regulate…”
Responses 84: We have replaced this sentence with “This indicates that these stress-related cis-elements may not drive significant transcription under the conditions tested, or may require specific developmental or environmental contexts.”
Comments 85: Line 734: Another case of an unnecessary, “Besides,” to start sentence.
Responses 85: We have these sentences as “While the VYL motif is essential for the function of AtWRI1 and GmWRI1b [39,70,117], its requirement is not universal, as evidenced by functional variants in RcWRI1 [63] and OsWRI1-1 [117].”
Comments 86: Line 736: Awkward wording related to “further test the VYL play roles…”. Rewrite to be clear and readable.
Responses 86: We have improved the sentence as ”Therefore, the functional importance of the VYL motif in AhWRIs remains to be determined.”
Comments 87: Line 741: Awkward wording related to “further test in the function…”. Rewrite to be clear and readable.
Responses 87: We have improved the sentence as “This suggests that the putative 14-3-3/BPM-binding motifs in AhWRI1s and AhWRI3s may play similar roles in regulating their protein stability.”
Comments 88: Line 746, a comma “,” is needed before “and”.
Responses 88: We have improved the sentence as “Consistent with bioinformatic predictions (ProtComp 9.0, Table S8), both AhWRI1a and AhWRI1d were experimentally confirmed to be nuclear localized (Figure 7).”
Comments 89: Line 753: Another case of an unnecessary, “Besides,” to start sentence.
Responses 89: We have replaced "Besides," with “Additionally,” and improved the sentence as “Additionally, AtMED15 interacts directly with the non-C-terminal acidic region of AtWRI1 to facilitate transcriptional activation [120].”
Comments 90: Line 768: Add “and” so that it reads, “…M. truncultalua [94], and fiber …”
Responses 90: We have improved the sentence as “Beyond their role in oil biosynthesis, plant WRI1 homologs exhibit pleiotropic effects in various processes, including root nodulation in soybean [39], arbuscular mycorrhiza symbiosis in M. truncatula [79], auxin homeostasis [124], and cutin and wax biosynthesis [33,125].”
Comments 91: Line 776: Not clear meaning of, “…note more closed to …”. Rewrite to be clear.
Responses 91: We have improved the sentence as “In peanut, AhWRI1d, AhWRI1f, and AhWRI1g were highly expressed in nodules (Figure 5A-B), suggesting that they may play key roles in root nodulation, despite showing less phylogenetic proximity to GmWRI1a/b [39] than AhWRI1a/b/c (Figure S11).”
Comments 92: Lines 783-784: Again, not clear/awkward wording around “most closed …” Rewrite all three cases of this usage to be clear.
Responses 92: We have improved the sentence as “Phylogenetic analysis revealed close relationships between certain AhWRI1s and MtWRI5 genes: AhWRI1d/e with MedtrWRI5c, AhWRI1f/g with MedtrWRI5b, and AhWRI1h/i with MedtrWRI5a (Figure S11).”
Comments 93: Lines 786-787: This first sentence is not clear what is trying to be stated here. Is it that WRI1s genes are transcriptionally regulated OR that WRI1 proteins are transcriptionally regulating other genes?
Responses 93: We appreciate the reviewer's comment. We have improved these sentences as “Their promoters contain multiple RY and CCAAT elements (Table S16), implying potential regulation by LEC2 and NF-Y transcription factors, similar to the activation of AtWRI1 by AtLEC2 binding to RY motif [36,96], and Elaeis guineensis EgWRI1-1 by EgNF-Y complexes binding specifically to CCAAT-boxes [123].”
This explicitly states that WRI1 genes themselves undergo transcriptional regulation.
Comments 94: Line 802: The “Pa” plant species needs to be writing out fully on first use.
Responses 94: We have expanded "Pa" to its full scientific name at first use. The sentence now reads: “In contrast, Persea americana (avocado) PaWRI2 is highly expressed during fruit development and enhances triacylglycerol accumulation in transient overexpression assays [67]”
Comments 95: Line 810: Another case of an unnecessary, “Besides,” to start sentence.
Responses 95: We have deleted “Besides,”and improved the sentence as “Their promoters contain multiple cis-acting elements associated with MeJA, anaerobic induction, gibberellin response, defense and stress signaling, and drought, suggesting functional roles beyond fatty acid synthesis.”
Comments 96: Line 821: Change “Furtherly” to be “Further,” or “Furthermore,”.
Responses 96: We have replaced "Furtherly" with "Furthermore" as suggested. The revised text now reads “Furthermore, AtWRI4 is salt-induced and regulates cuticular wax biosynthesis [80].”
Comments 97: Line 839: Change “lowly” to be “very low expression.”
Responses 97: We have improved the sentence as “In this study, AhWRI3a/b showed very low expression across tissues (Figure 5A-C), while AhWRI3c/d were more widely expressed (Figure 5A-B), particularly in florescences (Figure 5A), but not in seeds (Figure 5C).”
Comments 98: Line 858: As mentioned above several times. The data show a small but significant increase for some of the fatty acids, but I do not feel the data show a “substantially increased” level.
Responses 98: Yes. We have replaced “substantially” with “significantly”.
Comments 99: Line 864: Another case of an unnecessary, “Besides,” to start sentence and paragraph in this case. And, as stated several times above, without actual expression data the statement of “co-overexpression” is not substantiated.
Responses 99/100: We have conducted a full-manuscript search for "Besides," (especially at sentence starts) and then deleted it universally or replaced with stronger transitions where needed: "Additionally," “In addition” (if adding information).
We have expression data of qRT-PCR (new Figure S8 and Methods section):
- Quantified mRNA levels of all four peanut genes (AhLEC1b, AhFUS3b, AhWRI1a, and AhWRI1d) in each transgenic line
- Included three biological replicates per line with control (wild-type)
These additions confirm that: All four transgenes are actively expressed.
Comments 100: Line 864: I would delete “could” at end of this line. It makes sentence read oddly.
Responses 100: We have improved the sentence as “Thousand-seed weight in the four transgenic lines increased significantly by 8.74–19.20%, compared to the 3.31–27.81% increase in AhNF-YB1/10-overexpressing plants [17].”
Comments 101: Line 873: Add comma “,” after “And,”
Responses 101: We have improved the sentence as “In contrast, AhWRI1a (GG) overexpression promoted larger rosette leaves, early flowering, larger pods, and longer seeds [122].”
Methods:
Comments 102: Line 891: Not sure “blasting” is a real word. If used, it should instead be “BLASTing” to accurately reflect the acronym BLAST.
Responses 102: We thank you for catching this terminology issue. We have revised the heading to: "4.2. BLAS ing and Cloning of LEC1, FUS3, and WRI Family Members in A. hypogaea"
Comments 103: Lines 962-962: As mentioned above, must include more information with details about the RNA-seq wet-lab procedures and RNA-seq data analysis and statistics, which can be complicated. And, must include more information about the six developmental stages.
Responses 103: We have added the critical information in Results, Methods or Figure S7 legend, including seed developmental staging and RNA-seq experimental details, as follows:
1) Seed Developmental Staging (new Figure S7 and Methods text):
- We categorized the development seeds into six stages across their development based on kernel morphology according to Pattee et al. (1974). Three biological replicates (each containing more than 10 seeds) were analyzed per stage.
- We have added the detailed legend to supplementary Figure S7.
2) RNA-seq Experimental Details (new Methods subsection):
- Sample preparation: three biological replicates with more than 10 seeds each
- We have added the information about RNA extraction protocol, library construction, sequencing platform, data processing, and statistical analysis.
- Our transcriptomic data on seed development will be used for multi-omics (transcriptomics, lipidomics, and proteomics) analysis of peanut seed development, and we will publish these data when we submit the manuscript about multi-omics analysis of peanut seed development. Here, we have added the raw transcriptomic FPKM data for AhLEC1s, AhFUS3s, and AhWRIs in Table S13.
The revised manuscript now provides complete transparency for these foundational methods. We deeply regret these oversights and thank the reviewer for identifying these essential improvements. These additions significantly strengthen the reproducibility of our study.
Pattee, H.E.; Johns, E.B.; Singleton, J.A.; Sanders, T.H. Composition Changes of Peanut Fruit Parts During Maturation. Peanut Sci. 1974, 1, 57-62, doi:10.3146/i0095-3679-1-2-6.
Conclusions:
Comments 104: Lines 999-1000: As mentioned several times above, authors must provide actual expression data for peanut genes/RNAs/proteins in transgenic Arabidopsis for them to so strongly state “co-expression” and especially if claiming “constitutively co-expression” that means all four genes expressed constitutive (always) throughout the full life of the Arabidopsis plants. A strong claim for which there are no data provided to directly support.
Responses 104: We have carried out qRT-PCR to detect the expressive levels of these four peanut genes in transformed plants, and added the data in Figure S8.
References:
Comments 105: As mentioned near beginning of this long review, this reference list is huge (182 references), comparable to a very extensive review article. While likely most of these are critical, there might be some that are redundant.
Responses 105: We appreciate the reviewer's valid observation regarding reference volume. We have:
1) Systematically evaluated all 182 references for redundancy and relevance.
2) Reduced total citations from 182 to 144 by:
Replacing multiple overlapping references with recent review articles;
Removing redundant references or incremental updates;
Consolidating supporting evidence for broad claims.
3) Verified critical retention of all foundational and field-defining works.
The revised manuscript now contains 144 references. We confirm every retained citation provides unique empirical or conceptual value to our study's narrative.
Comments 106: Comments on the Quality of English Language Included all of the ones I caught in comments above.
Responses 106: We sincerely appreciate the reviewer's detailed English language corrections. All noted issues have been corrected. We further conducted full manuscript revisions to:
1) Eliminate grammatical errors and improve sentence flow
2) Replace informal/nonstandard terms with precise scientific language
3) Ensure consistent formatting of genes/species
Additionally, we have engaged MDPI’s professional English editing service to ensure linguistic accuracy and readability. We hope these revisions have significantly improved the clarity and professionalism of the manuscript.

Round 2
Reviewer 2 Report
Comments and Suggestions for Authors
I thank the authors for their thorough and precise responses to my comments.
I am satisfied with the revisions and believe that all comments and suggestions have been adequately addressed.
Reviewer 3 Report
Comments and Suggestions for Authors
I thank the authors for their considered responses to my comments and suggestions. These have addressed my concerns and I have no further comments.
Reviewer 4 Report
Comments and Suggestions for Authors
Plants Manuscript #: 3751514
Authors: X. Yin et al., 2025
Title: Genome-wide identification of Arachis hypogaea LEC1s, FUS2s, and WRIs and co-expression of AhLEC1b, AhFUS3b, AhWRI1a and AhWRI1d increased oil content in Arabidopsis seeds.
This is a re-review of the manuscript. The authors have addressed all the major concerns and comments about the science, writing and conclusions from the data. I support publishing this work.